# IGC-Net for conditional average potential outcome estimation over time

**Konstantin Hess[1,2,\*], Dennis Frauen[1,2], Valentyn Melnychuk[1,2], Stefan Feuerriegel[1,2]**

[1]LMU Munich    [2]Munich Center for Machine Learning
[\*]Corresponding author: `k.hess@lmu.de`

## Abstract

Estimating potential outcomes for treatments *over time* based on observational data is important for personalized decision-making in medicine. However, many existing methods for this task fail to properly adjust for time-varying confounding and thus yield biased estimates. There are only a few neural methods with proper adjustments, but these have inherent limitations (e.g., division by propensity scores that are often close to zero), which result in poor performance. As a remedy, we introduce the *iterative G-computation network* (IGC-Net). Our IGC-Net is a novel, neural end-to-end model which adjusts for time-varying confounding in order to estimate conditional average potential outcomes (CAPOs) over time. Specifically, our IGC-Net is the first neural model to perform fully regression-based iterative G-computation for CAPOs in the time-varying setting. We evaluate the effectiveness of our IGC-Net across various experiments. In sum, this work represents a significant step towards personalized decision-making from electronic health records.

## 1 Introduction

Causal machine learning has large potential to personalize treatment decisions in medicine (Feuerriegel et al., 2024). An important task for this is to estimate conditional average potential outcomes (CAPOs) from observational data *over time*. Recently, such data has become prominent in medicine due to the growing prevalence of electronic health records (EHRs) (Allam et al., 2021; Bica et al., 2021) and wearable devices (Battalio et al., 2021; Murray et al., 2016). However, estimating CAPOs over time is notoriously difficult due to *time-varying confounding*: for several-step-ahead predictions, future time-varying confounders are unobserved as they depend on both future treatments and outcomes that have not yet occurred, which forces inference to rely only on past information and model-based forecasts.

One stream of methods (i.e., **CRN** (Bica et al., 2020), **CT** (Melnychuk et al., 2022), and **TE-CDE** (Seedat et al., 2022)) fails to perform proper adjustments for time-varying confounding and, thus, do not target the correct estimand. Hence, methods from this stream have infinite-sample bias: i.e., irreducible estimation errors irrespective of the amount of available data, which renders these methods unsuitable for medical applications.

To the best of our knowledge, there are only two neural methods that perform proper adjustments for time-varying confounding. However, these have important limitations (see Table 1). On the one hand, **RMSNs** (Lim et al., 2018) perform inverse propensity weighting, which, in the time-varying setting, relies on products of inverse propensity scores and, hence, division by values close to zero. On the other hand, **G-Net** (Li et al., 2021) and **G-transformer** (Xiong et al., 2024) perform G-computation, yet by *estimating the distribution of all time-varying confounders at all time-steps the future*, which is inefficient due to two reasons: it needs to estimate all moments of a high-dimensional random variable, and it requires indirect inference via Monte Carlo sampling.

| Category | Method(s) | Issue |
|---|---|---|
| ① **w/o proper adjustments** | **CRN** (Bica et al., 2020), **CT** (Melnychuk et al., 2022), **TE-CDE** (Seedat et al., 2022) | ✗ No proper adjustment and thus infinite-data bias (i.e., irreducible estimation errors irrespective of the dataset size) |
| ② **w/** proper adjustments | **RMSNs** (Lim et al., 2018) | ✗ Product of inverse propensity scores; division close to zero |
| | **G-Net** (Li et al., 2021) **G-transformer** (Xiong et al., 2024) | ✗ Estimation of the entire distribution (i.e., all higher-order moments) of confounders via MC sampling |
| | **IGC-Net** (ours) | ✓ Neural end-to-end iterative regression algorithm |

Table 1: **Overview of key neural methods for estimating CAPOs over time.** Existing methods that perform *proper adjustments* for time-varying confounding have important limitations: RMSNs rely on products of inverse propensity scores and *unstable division by values close to zero*, and G-Net estimates the *entire distribution of all time-varying confounders in the future via MC sampling*.

To fill the above research gap, we propose a novel, neural method for estimating CAPOs over time, which we call ***iterative G-computation network* (IGC-Net)**.[1] Our method allows for proper adjustments for time-varying confounding by leveraging the idea of G-computation, but where we develop a novel, regression-based iterative approach to integrate G-computation into neural networks through an end-to-end training algorithm. As a result, our IGC-Net overcomes the limitations of existing methods. Unlike RMSNs, we avoid inverse propensity scores, which makes our method robust, especially for longer time horizons. Unlike G-Net, we avoid estimating any probability distribution (i.e., any higher-order moments), but rather estimate CAPOs directly through our iterative regression algorithm in an end-to-end manner. We demonstrate the effectiveness of our IGC-Net across various experiments. Our IGC-Net based on transformers achieves state-of-the-art performance.

## 2 RELATED WORK

**APOs over time:** Estimating average potential outcomes (APOs) over time has a long-ranging history in classical statistics and epidemiology (Lok, 2008; Robins, 1986; Rytgaard et al., 2022; van der Laan & Gruber, 2012). Popular approaches are the G-methods (Robins & Hernán, 2009), including marginal structural models (MSMs) (Robins & Hernán, 2009; Robins et al., 2000), structural nested models (Robins, 1994; Robins & Hernán, 2009), G-computation (Bang & Robins, 2005; Robins, 1999; Robins & Hernán, 2009), and TMLE (van der Laan & Gruber, 2012). Some of these have been instantiated by neural models (Frauen et al., 2023a; Shirakawa et al., 2024). However, these works do **not** focus estimating CAPOs. Instead, they **ignore** individual patient characteristics.

**CAPOs over time (Table 1):** Existing neural methods have ***important limitations***:

Limitation ① proper adjustments: A number of neural methods for estimating CAPOs have been proposed that *do not properly adjust* for time-varying confounders. As a result, these methods are *biased* as they do not target the correct estimand. Here, key examples are the counterfactual recurrent network (**CRN**) (Bica et al., 2020), the treatment effect neural controlled differential equation (**TE-CDE**) (Seedat et al., 2022), and the causal transformer (**CT**) (Melnychuk et al., 2022). These methods try to account for time-varying confounders through balanced representations. However, balancing was originally designed for reducing finite-sample estimation variance and *not* for mitigating confounding bias (Johansson et al., 2016; Shalit et al., 2017). Hence, this is a heuristic and may even introduce representation-induced confounding bias (Melnychuk et al., 2024). Unlike these methods, our IGC-Net performs *proper adjustments for time-varying confounders*.

Limitation ② adjustment strategy: Existing neural methods with proper causal adjustments employ adjustment strategies that are in other ways problematic in empirical applications. On the one hand, the recurrent marginal structural networks (**RMSNs**) (Lim et al., 2018) construct pseudo outcomes through inverse propensity weighting (IPW). However, IPW constructs pseudo-outcomes with large variance compared to G-computation ($\rightarrow$ we show this later in Proposition 3). Specifically, for several-step-ahead predictions, IPW relies on products of inverse propensity scores and, hence, *division by values close to zero*. In contrast, the **G-Net** (Li et al., 2021) and **G-transformer** (Xiong et al., 2024) use G-computation to adjust for time-varying confounding (see Supplement D), but it proceeds by estimating the *entire distribution of all confounders at several time-steps in the future* (i.e., all moments of a high-dimensional random variable), which leads to poor empirical performance (see Section 4.4 for a detailed discussion). Different from G-Net, we propose a regression-based

---

[1]Code is available at `https://github.com/konstantinhess/IGC_net`.

approach to G-computation. Hence, our IGC-Net has two advantages in that **(i)** we do **_not_** attempt to learn the full distribution of all future time-varying confounders (i.e., all higher-order moments) but only estimate the first moments of a much lower-dimensional random variable, and **(ii)** we do **_not_** need Monte Carlo sampling but can perform end-to-end regressions.

## 3 PROBLEM FORMULATION

**Setup:** We follow previous literature (Bica et al., 2020; Li et al., 2021; Melnychuk et al., 2022) and consider data that consist of realizations of the following random variables: (i) outcomes $Y_t \in \mathbb{R}^{d_y}$, (ii) covariates $X_t \in \mathbb{R}^{d_x}$, and (iii) treatments $A_t \in \{0, 1\}^{d_a}$ at time steps $t \in \{0, \ldots, T\} \subset \mathbb{N}_0$, where $T$ is the time window that follows some unknown counting process. We are interested in estimating CAPOs for $\tau$ steps in the future. For any random variable $U_t \in \{Y_t, X_t, A_t\}$, we write $U_{t:t+\tau} = (U_t, \ldots, U_{t+\tau})$ to refer to a specific subsequence of a random variable. We further write $\bar{U}_t = U_{0:t}$ to denote the full trajectory of $U$ including time $t$. Finally, we write $\bar{H}_{t+\delta}^t = (\bar{Y}_{t+\delta}, \bar{X}_{t+\delta}, \bar{A}_{t-1})$ for $\delta \geq 0$, and we let $\bar{H}_t = \bar{H}_t^t$ denote the collective history of (i)–(iii).

**Estimation task:** We are interested in estimating the *conditional* average potential outcome (CAPO) for a future, interventional sequence of treatments, given the observed history. For this, we build upon the potential outcomes framework (Neyman, 1923; Rubin, 1978) for the time-varying setting (Robins & Hernán, 2009; Robins et al., 2000). Hence, we aim to estimate the potential outcome $Y_{t+\tau}[a_{t:t+\tau-1}]$ at future time $t+\tau$, $\tau \in \mathbb{N}$, for an interventional sequence of treatments $\bar{a} = a_{t:t+\tau-1}$, *conditionally* on the observed history $\bar{H}_t = \bar{h}_t$. That is, our objective is to estimate

$$\mathbb{E}\left[Y_{t+\tau}[a_{t:t+\tau-1}] \mid \bar{H}_t = \bar{h}_t\right]. \tag{1}$$

**Identifiability:** In order to estimate the causal quantity in Equation 1 from observational data, we make the following identifiability assumptions (Robins & Hernán, 2009; Robins et al., 2000) that are *standard in the literature* (Bica et al., 2020; Li et al., 2021; Lim et al., 2018; Melnychuk et al., 2022; Seedat et al., 2022): (1) *Consistency:* For an observed sequence of treatments $\bar{A}_t = \bar{a}_t$, the observed outcome $Y_{t+1}$ equals the corresponding potential outcome $Y_{t+1}[\bar{a}_t]$. (2) *Positivity:* For any history $\bar{H}_t = \bar{h}_t$ that has non-zero probability $\mathbb{P}(\bar{H}_t = \bar{h}_t) > 0$, there is a positive probability $\mathbb{P}(A_t = a_t \mid \bar{H}_t = \bar{h}_t) > 0$ of receiving any treatment $A_t = a_t$, where $a_t \in \{0, 1\}^{d_a}$. (3) *Sequential ignorability:* Given a history $\bar{H}_t = \bar{h}_t$, the treatment $A_t$ is independent of the potential outcome $Y_{t+\delta}[a_{t:t+\delta-1}]$, that is, $A_t \perp Y_{t+\delta}[a_{t:t+\delta-1}] \mid \bar{H}_t = \bar{h}_t$ for all $a_{t:t+\delta-1} \in \{0, 1\}^{\delta \times d_a}$.

**Why dealing with confounding is non-trivial in time-varying settings:** Estimating CAPOs without confounding bias poses a non-trivial challenge in the time-varying setting. The issue lies in the complexity of handling future time-varying confounders. In particular, for $\tau \geq 2$ and $1 \leq \delta \leq \delta' \leq \tau - 1$, future covariates $X_{t+\delta}$ and outcomes $Y_{t+\delta}$ may affect the probability of receiving certain treatments $A_{t+\delta'}$. Importantly, the time-varying confounders are *unobserved* during inference time, which is generally known as *runtime confounding* (Coston et al., 2020). Therefore, in order to estimate the direct effect of an interventional treatment sequence, one needs to adjust for the time-varying confounders. That is, it is in general **insufficient** to only adjust for the history (Frauen et al., 2025) via

$$\mathbb{E}\left[Y_{t+\tau}[a_{t:t+\tau-1}] \mid \bar{H}_t = \bar{h}_t\right] \neq \mathbb{E}\left[Y_{t+\tau} \mid \bar{H}_t = \bar{h}_t, A_{t:t+\tau-1} = a_{t:t+\tau-1}\right]. \tag{2}$$

One way to adjust for time-varying confounders is inverse propensity weighting (IPW), which is leveraged by RMSNs (Lim et al., 2018). However, as we show in **Proposition 3**, IPW is subject to large variance.

**G-computation:** Instead, we leverage G-computation (Bang & Robins, 2005; Robins, 1999; Robins & Hernán, 2009), which provides a rigorous way to account for the time-varying confounders. Formally, G-computation identifies the causal quantity in Equation 1 via

$$\begin{aligned}
&\mathbb{E}[Y_{t+\tau}[a_{t:t+\tau-1}] \mid \bar{H}_t = \bar{h}_t] \\
=&\mathbb{E}\bigg\{\mathbb{E}\bigg[\ldots\mathbb{E}\big\{\mathbb{E}[Y_{t+\tau} \mid \bar{H}_{t+\tau-1}^t, A_{t:t+\tau-1} = a_{t:t+\tau-1}] \mid \bar{H}_{t+\tau-2}^t, A_{t:t+\tau-2} = a_{t:t+\tau-2}\big\} \\
&\ldots \big| \bar{H}_{t+1}^t, A_{t:t+1} = a_{t:t+1}\bigg] \big| \bar{H}_t = \bar{h}_t, A_t = a_t\bigg\}.
\end{aligned} \tag{3}$$

We provide derivation of the G-computation formula for CAPOs in **Supplement D**. Due to the nested structure of G-computation, estimating Equation 3 from data is challenging.

*Why the approach by G-Net is problematic:* So far, only G-Net (Li et al., 2021) and G-transformer (Xiong et al., 2024) have used G-computation for estimating CAPOs in a neural model. For this, they estimate Equation 3 through

$$\int y_{t+\tau} p(y_{t+\tau} \mid \bar{h}_{t+\tau-1}^t, a_{t:t+\tau-1}) \prod_{\delta=1}^{\tau-1} \mathrm{d}p(x_{t+\delta}, y_{t+\delta} \mid \bar{h}_t, x_{t+1:t+\delta-1}, y_{t+1:t+\delta-1}, a_{t:t+\delta-1}). \quad (4)$$

However, Equation 4 requires estimating the entire *distribution of all time-varying confounders at several time steps in the future*. This has two drawbacks: (i) the distribution must be approximated (e.g., through Monte Carlo sampling), which is inefficient; and (ii) *all moments* of a $(\tau-1) \times (d_x+d_y)$-dimensional random variable need to be estimated. Importantly, our IGC-Net addresses both (i) and (ii). We provide a detailed comparison to our IGC-Net in **Section 4.4**.

*Our approach to G-computation*: We propose a novel way to address the above challenges, and integrate G-computation into neural networks to offer better empirical performance. In contrast to G-Net and G-transformer, our IGC-Net does **not** rely on high-dimensional distribution estimation through Monte Carlo sampling. Further, our IGC-Net does **not** require estimating any probability distribution. Instead, it performs *regression-based iterative G-computation* in an end-to-end training algorithm. Thereby, we perform *proper adjustments for time-varying confounding* through Equation 3, while relying only on regressions of with *low-variance pseudo-outcomes*.

## 4 ITERATIVE G-COMPUTATION NETWORK

In the following, we present our iterative G-computation network. Inspired by (Bang & Robins, 2005; Robins, 1999; Robins & Hernán, 2009) for APOs, we reframe G-computation for CAPOs over time through recursive conditional expectations. Thereby, we precisely formulate the training objective of our IGC-Net through iterative regressions ($\rightarrow$Proposition 1). We proceed below by first extending regression-based iterative G-computation to account for the heterogeneous response to a treatment intervention. We then detail the architecture of our IGC-Net and provide details on the end-to-end training and inference, which guarantees that we target the correct estimand and adjust for time-varying confounding ($\rightarrow$Proposition 2).

### 4.1 REGRESSION-BASED ITERATIVE G-COMPUTATION FOR CAPOS

Our IGC-Net leverages G-computation as in Equation 3 and, therefore, properly adjusts for time-varying confounders in Equation 1. However, we do **not** attempt to integrate over the estimated distribution of all time-varying confounders. Instead, one of our main novelties is that our IGC-Net performs iterative regressions in a *neural end-to-end architecture*. This allows us to estimate Equation 1 *without approximating high-dimensional probability distributions*.

We reframe Equation 3 equivalently as a recursion of conditional expectations. Thereby, we can formulate the iterative regression objective of our IGC-Net. In particular, our approach resembles an *iterative pseudo-outcome regression*. For this, let

$$g_{t+\delta}^{\bar{a}}(\bar{h}_{t+\delta}^t) = \mathbb{E}[G_{t+\delta+1}^{\bar{a}} \mid \bar{H}_{t+\delta}^t = \bar{h}_{t+\delta}^t, A_{t:t+\delta} = a_{t:t+\delta}], \quad (5)$$

where the *pseudo-outcomes* are defined as

$$G_{t+\tau}^{\bar{a}} = Y_{t+\tau}, \quad (6)$$

$$G_{t+\delta}^{\bar{a}} = g_{t+\delta}^{\bar{a}}(\bar{H}_{t+\delta}^t) \quad (7)$$

for $\delta = 0, \ldots, \tau - 1$. By reformulating the G-computation formula through recursions, the nested expectations in Equation 3 are now given by

$$G_{t+\tau-1}^{\bar{a}} = \mathbb{E}[Y_{t+\tau} \mid \bar{H}_{t+\tau-1}^t, A_{t:t+\tau-1} = a_{t:t+\tau-1}], \quad (8)$$

$$G_{t+\tau-2}^{\bar{a}} = \mathbb{E}\Big[\mathbb{E}[Y_{t+\tau} \mid \bar{H}_{t+\tau-1}^t, A_{t:t+\tau-1} = a_{t:t+\tau-1}] \mid \bar{H}_{t+\tau-2}^t, A_{t:t+\tau-2} = a_{t:t+\tau-2}\Big], \quad (9)$$

and so on. Hence, the G-computation formula in Equation 3 can be rewritten as

$$g_t^{\bar{a}}(\bar{h}_t) = \mathbb{E}[Y_{t+\tau}[a_{t:t+\tau-1}] \mid \bar{H}_t = \bar{h}_t]. \quad (10)$$

To further illustrate our regression-based iterative G-computation, we provide **two examples** in **Supplement E**, where we show step-by-step how our approach adjusts for time-varying confounding.

We show in the following proposition that iterative pseudo-outcome regression recovers the CAPO and thus performs proper adjustments for time-varying confounding. We summarize the iterative pseudo-outcome regression in the following proposition.

**Proposition 1.** *The regression-based iterative G-computation yields the CAPO in Equation 1.*

*Proof.* See Supplement C.1. □

In order to correctly estimate Equation 10 for a given history $\bar{H}_t = \bar{h}_t$ and an interventional treatment sequence $a = a_{t:t+\tau-1}$, all subsequent pseudo-outcomes in Equation 7 are required. However, the ground-truth realizations of the pseudo-outcomes $G_{t+\delta}^{\bar{a}}$ are *not available in the data*. Instead, only realizations of $G_{t+\tau}^{\bar{a}} = Y_{t+\tau}$ in Equation 6 are observed during the training. Hence, when training our IGC-Net, it alternately generates predictions $\tilde{G}_{t+\delta}^{\bar{a}}$ of the pseudo-outcomes for $\delta = 1, \ldots \tau - 1$, which it then uses for learning the estimator of Equation 5.

Therefore, the training of our IGC-Net completes two steps in an iterative scheme: First, it runs a ⒜ *generation step*, where it generates predictions of the pseudo-outcomes Equation 7. Then, it runs a ⒝ *learning step*, where it regresses the predictions $\tilde{G}_{t+\delta}^{\bar{a}}$ for Equation 7 and the observed $G_{t+\tau}^{\bar{a}} = Y_{t+\tau}$ in Equation 6 on the history to update the estimator for Equation 5. Finally, the updated estimators are used again in the next ⒜ *generation step*. This procedure resembles an iterative pseudo-outcome regression. Thereby, our IGC-Net is designed to simultaneously ⒜ *generate* predictions and ⒝ *learn* during the training. Importantly, we propose an implementation where both steps are performed in an *end-to-end* architecture, ensuring that information is shared across time and data is used efficiently.

## 4.2 MODEL ARCHITECTURE

We first introduce the architecture of our IGC-Net. Then, we explain the iterative prediction and learning scheme inside our IGC-Net, which presents one of the main novelties. Finally, we introduce the inference procedure.

Our IGC-Net consists of two key components (see **Figure 1**): (i) a *neural backbone* $z^{\phi}(\cdot)$, which can be, for example, be an LSTM or a transformer, and (ii) several *G-computation heads* $\{g_{\delta}^{\phi}(\cdot)\}_{\delta=0}^{\tau-1}$, where $\phi$ denote the trainable weights. First, the neural back-

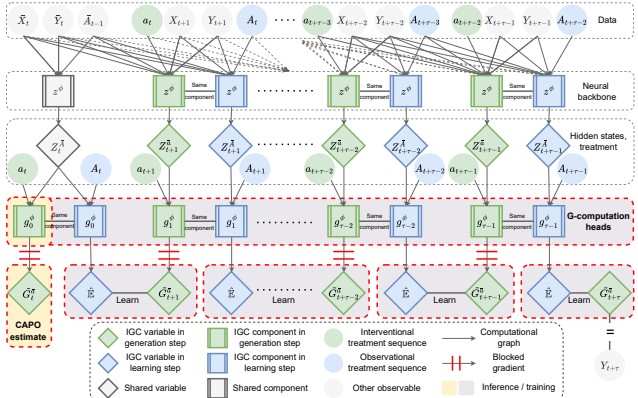

Figure 1: **Iterative G-computation network.** Neural end-to-end architecture of our iterative G-computation network.

bone encodes the entire observed history. Then, the G-computation heads take the encoded history and perform the iterative regressions according to Equation 5. For all $t = 1, \ldots, T - \tau$ and $\delta = 0, \ldots, \tau - 1$, the components are designed as follows:

• **Neural backbone:** For our main experiments in Section 5, we use a multi-input transformer $z^{\phi}(\cdot)$ as our neural backbone, which consists of three connected encoder-only sub-transformers $z^{\phi k}(\cdot)$, $k \in \{1, 2, 3\}$ and is inspired by (Melnychuk et al., 2022). We provide details on the architecture in **Supplement H** and provide additional *ablations with an alternative LSTM backbone* in **Supplement F.1**. At time $t$, the transformer $z^{\phi}(\cdot)$ receives data $\bar{H}_t = (\bar{Y}_t, \bar{X}_t, \bar{A}_{t-1})$ as input and passes them to one corresponding sub-transformer. In particular, each sub-transformer $z^{\phi k}(\cdot)$ is responsible to focus on one particular $\bar{U}_t^k \in \{\bar{Y}_t, \bar{X}_t, \bar{A}_{t-1}\}$ in order to effectively process the different types of inputs. Further, we ensure that information is shared between the sub-transformers. The output of the multi-input transformer are hidden states $Z_t^{\bar{A}}$, which are then passed to the (ii) G-computation heads.

Figure 2: **How our IGC-Net performs G-computation to adjust for time-varying confounding.**

• **G-computation heads:** The *G-computation heads* $\{g_\delta^\phi(\cdot)\}_{\delta=0}^{\tau-1}$ are the read-out component of our IGC-Net. As input at time $t + \delta$, the G-computation heads receive the hidden state $Z_{t+\delta}^{\bar{A}}$ from the above neural backbone. Recall that we seek to perform the iterative regressions in Equation 5 and Equation 10, respectively. For this, we require estimators of $\mathbb{E}[G_{t+\delta+1}^{\bar{a}} \mid \bar{H}_{t+\delta}, \bar{A}_{t+\delta}]$. Hence, the G-computation heads compute

$$\hat{\mathbb{E}}[G_{t+\delta+1}^{\bar{a}} \mid \bar{H}_{t+\delta}, A_{t+\delta}] = g_\delta^\phi(Z_{t+\delta}^{\bar{A}}, A_{t+\delta}), \quad \text{with} \quad Z_{t+\delta}^{\bar{A}} = z^\phi(\bar{H}_{t+\delta}) \tag{11}$$

for $\delta = 0, \ldots, \tau - 1$. As a result, the G-computation heads and the neural backbone together form the estimators that are required for the regression-based iterative G-computation. In particular, we thereby ensure that, for $\delta = 0$, the last G-computation head $g_0^\phi(\cdot)$ is trained as the estimator for the CAPO as given in Equation 10. That is, as illustrated in Fig. 2, for a fully trained neural backbone and G-computation heads, our IGC-Net estimates the CAPO via

$$\hat{\mathbb{E}}[Y_{t+\tau}[a_{t:t+\tau-1}] \mid \bar{H}_t = \bar{h}_t] = g_0^\phi(z^\phi(\bar{h}_t), a_t). \tag{12}$$

### 4.3 ITERATIVE TRAINING AND INFERENCE

We now introduce the iterative training of our IGC-Net, which consists of a Ⓐ *generation step* and a Ⓑ *learning step*. Then, we show how inference for a given history $\bar{H}_t = \bar{h}_t$ can be achieved. We summarize the iterative learning algorithm in **Algorithm 1**.

• **Iterative training:** Our IGC-Net is designed to estimate the CAPO $g_t^{\bar{a}}(\bar{h}_t)$ in Equation 10 for a given history $\bar{H}_t = \bar{h}_t$ and an interventional treatment sequence $a = a_{t:t+\tau-1}$ via Equation 12. Therefore, the G-computation heads in Equation 11 require the pseudo-outcomes $\{G_{t+\delta}^{\bar{a}}\}_{\delta=1}^\tau$ from Equation 7 during training. However, they are only available in the training data for $\delta = \tau$. That is, we only observe the factual outcomes $G_{t+\tau}^{\bar{a}} = Y_\tau$.

As a remedy, our IGC-Net first predicts the remaining pseudo-outcomes $\{G_{t+\delta}^{\bar{a}}\}_{\delta=1}^{\tau-1}$ in the Ⓐ *generation step*. Then, it can use these generated pseudo-outcomes and the observed $G_{t+\tau}^{\bar{a}}$ for learning the network weights $\phi$ in the Ⓑ *learning step*. In the following, we write $\{\tilde{G}_{t+\delta}^{\bar{a}}\}_{\delta=1}^{\tau-1}$ for the generated pseudo-outcomes. Note that, since $G_{t+\tau}^{\bar{a}} = Y_{t+\tau}$ is observed during training, we do not have to generate this target. Yet, for notational convenience, we write $\tilde{G}_{t+\tau}^{\bar{a}} = G_{t+\tau}^{\bar{a}}$.

Ⓐ *Generation step:* In this step, our IGC-Net generates $\tilde{G}_{t+\delta}^{\bar{a}} \approx g_{t+\delta}^{\bar{a}}(\bar{H}_{t+\delta}^t)$ as substitutes for Equation 7, which are the pseudo-outcomes in the iterative regression-based G-computation. Formally, our IGC-Net predicts these via

$$\tilde{G}_{t+\delta}^{\bar{a}} = g_\delta^\phi(Z_{t+\delta}^{\bar{a}}, a_{t+\delta}), \tag{13}$$

where

$$Z_{t+\delta}^{\bar{a}} = z^\phi(\bar{H}_{t+\delta}^t, a_{t:t+\delta-1}), \tag{14}$$

for $\delta = 0, \ldots, \tau - 1$. For this, all operations are *detached* from the computational graph. Hence, our IGC-Net now has pseudo-outcomes $\{\tilde{G}_{t+\delta}^{\bar{a}}\}_{\delta=0}^\tau$, which it can use in the following Ⓑ *learning step*. Of note, these generated pseudo-outcomes will be noisy for early training epochs. However, as training progresses, the G-computation heads perform increasingly more accurate predictions, as we explain below.

Ⓑ *Learning step:* This step is responsible for updating the weights $\phi$ of the neural backbone $z^\phi(\cdot)$ and the G-computation heads $\{g_\delta^\phi(\cdot)\}_{\delta=0}^{\tau-1}$. For this, our IGC-Net learns the estimator for Equation 5 via

$$\hat{\mathbb{E}}[G_{t+\delta+1}^{\bar{a}} \mid \bar{H}_{t+\delta}^t, A_{t:t+\delta}] = g_\delta^\phi(Z_{t+\delta}^{\bar{A}}, A_{t+\delta}), \tag{15}$$

where

$$Z_{t+\delta}^{\bar{A}} = z^\phi(\bar{H}_{t+\delta}) \tag{16}$$

for $\delta = 0, \ldots, \tau - 1$. In particular, the estimator is optimized by backpropagating the squared error loss $\mathcal{L}$ for all $\delta = 0, \ldots, \tau - 1$ and $t = 1, \ldots, T - \tau$ via

$$\mathcal{L} = \frac{1}{T-\tau} \sum_{t=1}^{T-\tau} \left( \frac{1}{\tau} \sum_{\delta=0}^{\tau-1} \left( g_\delta^\phi(Z_{t+\delta}^{\bar{A}}, A_{t+\delta}) - \tilde{G}_{t+\delta+1}^{\bar{a}} \right)^2 \right). \tag{17}$$

Then, after $\phi$ is updated, we can use the updated estimator in the next Ⓐ *generation step*.

Here, it is important that for $\delta = \tau$, the pseudo-outcome $\tilde{G}_{t+\tau}^{\bar{a}} = Y_{t+\tau}$ is *available in the data*. By learning $Y_{t+\tau}$ with

$$\hat{Y}_{t+\tau} = g_{\tau-1}^\phi(Z_{t+\tau-1}^{\bar{A}}, A_{t+\tau-1}), \tag{18}$$

it is ensured the last G-computation head $g_{\tau-1}^\phi(\cdot)$ is learned on a ground-truth quantity. Thereby, the weights of $g_{\tau-1}^\phi(\cdot)$ are gradually optimized during training. Hence, the predicted pseudo-outcome

$$\tilde{G}_{t+\tau-1}^{\bar{a}} = g_{\tau-1}^\phi(Z_{t+\tau-1}^{\bar{A}}, a_{t+\tau-1}) \tag{19}$$

in the next Ⓐ *generation step* become mores accurate. Therefore, the G-computation head $g_{\tau-2}^\phi(\cdot)$ is learned on a more accurate prediction in the following Ⓑ *learning step*, which thus leads to a better generated pseudo-outcome $\tilde{G}_{t+\tau-2}^{\bar{a}}$, and so on. As a result, the optimization of the G-computation heads gradually improves from $g_{\tau-1}^\phi(\cdot)$ up to $g_0^\phi(\cdot)$.

• **Inference:** Finally, we introduce how inference is achieved with our IGC-Net. Given a history $\bar{H}_t = \bar{h}_t$ and an interventional treatment sequence $a = a_{t:t+\tau-1}$, our IGC-Net is trained to estimate of Equation 1 through Equation 10. For this, our IGC-Net computes the CAPO via

$$\hat{g}_t^{\bar{a}}(\bar{h}_t) = \hat{\mathbb{E}}[G_{t+1}^{\bar{a}} \mid \bar{H}_t = \bar{h}_t, A_t = a_t] = g_0^\phi(z^\phi(\bar{h}_t), a_t). \tag{20}$$

We summarize this in the following proposition.

---

**Algorithm 1:** Training and inference

**Training:**

**Input** : Data $(\bar{H}_{T-1}, A_{T-1}, Y_T)$, treatment sequence $\bar{a} \in \{0, 1\}^{d_a \times \tau}$, learning rate $\eta$
**Output:** Trained IGC-Net $\{z^\phi, g_\delta^\phi\}_{\delta=0}^{\tau-1}$
**for** $t = 1, \ldots, T - \tau$ **do**
    // Initialize
    $a_{t:t+\tau-1} \leftarrow\!\mid \bar{a}$
    $\tilde{G}_{t+\tau}^{\bar{a}} \leftarrow\!\mid Y_{t+\tau}$
    // Ⓐ Generation step
    **for** $\delta = 1, \ldots, \tau - 1$ **do**
        $Z_{t+\delta}^{\bar{a}} \leftarrow\!\mid z^\phi(\bar{H}_{t+\delta}^t, a_{t:t+\delta-1})$
        $\tilde{G}_{t+\delta}^{\bar{a}} \leftarrow\!\mid g_\delta^\phi(Z_{t+\delta}^{\bar{a}}, a_{t+\delta})$
    **end**

    // Ⓑ Learning step
    **for** $\delta = 0, \ldots, \tau - 1$ **do**
        $Z_{t+\delta}^{\bar{A}} \leftarrow z^\phi(\bar{H}_{t+\delta})$
        $\mathcal{L}_t^\delta \leftarrow \left( g_\delta^\phi(Z_{t+\delta}^{\bar{A}}, A_{t+\delta}) - \tilde{G}_{t+\delta+1}^{\bar{a}} \right)^2$
    **end**
**end**
// Compute gradient and update IGC-Net
    parameters $\phi$
$\phi \leftarrow \phi - \eta \nabla_\phi \left( \frac{1}{T-\tau} \sum_{t=1}^{T-\tau} \left( \frac{1}{\tau} \sum_{\delta=0}^{\tau-1} \mathcal{L}_t^\delta \right) \right)$

**Inference:**

**Input** : Data $\bar{H}_t$, treatments $\bar{a} \in \{0, 1\}^{d_a \times \tau}$
**Output:** $\hat{g}_t^{\bar{a}}(\bar{H}_t) = \hat{\mathbb{E}}[G_{t+1}^{\bar{a}} \mid \bar{H}_t, A_t = a_t]$
// Initialize
$a_{t:t+\tau-1} \leftarrow\!\mid \bar{a}$
// Ⓐ Generation step
$\hat{g}_t^{\bar{a}}(\bar{H}_t) \leftarrow\!\mid g_0^\phi(z^\phi(\bar{H}_t), a_t)$

**Legend**: Operations with "←" are attached to the computational graph, while operations with "←|" are detached from it.

---

**Proposition 2.** *Our IGC-Net estimates the G-computation formula as in Equation 10 and, therefore, performs proper adjustments for time-varying confounding.*

*Proof.* We provide an intuition in Figure 2. The full proof is in Supplement C.2. □

### 4.4 ADVANTAGES OVER EXISTING NEURAL METHODS

• **CT, CRN, and TE-CDE:** Our IGC-Net is vastly different from CT (Melnychuk et al., 2022), CRN (Bica et al., 2020) and TE-CDE (Seedat et al., 2022). These methods do **not** perform proper adjustments for time-varying confounding. In particular, they estimate $\mathbb{E}[Y_{t+\tau} \mid H_t = h_t, A_{t:t+\tau} = a_{t:t+\tau}]$, which is **not** the CAPO (Frauen et al., 2025). Hence, they target an *incorrect estimand*, leading to irreducible *bias*, so deploying them to medical scenarios would be irresponsible.

| Method | Estimated moment | Moment dimension | Estimation strategy |
|---|---|---|---|
| G-Net (Li et al., 2021) & G-transformer (Xiong et al., 2024) | 1st | $(\tau - 1) \times (d_x + d_y) + d_y$ | Monte Carlo sampling ✗ |
| | 2nd | $(\tau - 1) \times (d_x + d_y)$ | |
| | 3rd | $(\tau - 1) \times (d_x + d_y)$ | |
| | ... | ... | |
| | ∞ | $(\tau - 1) \times (d_x + d_y)$ | |
| IGC-Net (*ours*) | 1st | $\tau \times d_y$ | End-to-end regressions ✓ |

Table 2: **Comparison: G-Net and G-transformer vs. our IGC-Net.** G-Net and G-transformer require estimating the *full distribution of all time-varying confounders in the future (i.e., estimating all moments).*

• **RMSNs:** RMSNs (Lim et al., 2018) rely on pseudo-outcome regressions in order to adjust for time-varying confounders. However, their pseudo-outcomes are constructed via inverse propensity weighting, which leads to pseudo-outcomes with larger variance than ours:

**Proposition 3.** *Pseudo-outcomes constructed via inverse propensity weighting have larger variance than pseudo-outcomes in our iterative G-computation network.*

*Proof.* See Supplement C.3. □

• **G-Net and G-transformer** In order to estimate a $\tau$-step-ahead CAPO, G-Net (Li et al., 2021) and G-transformer (Xiong et al., 2024) require (i) a $d_y$-dimensional regression as well as estimating the *entire distribution* of a $(\tau - 1) \times (d_y + d_x)$-dimensional confounding variable. That is, it needs to estimate *all moments* of a *high-dimensional* random variable, which is inefficient. In contrast, our IGC-Net only requires $\tau$ regressions of a $d_y$-dimensional outcome and, hence, only needs to estimate the *first moment* of a much *lower-dimensional* random variable. We provide a comparison in **Table 2**.

## 5 EXPERIMENTS

We show the performance of our IGC-Net against key neural methods for estimating CAPOs over time (see Table 1). Further details (e.g., implementation details, hyperparameter tuning, runtime) are given in **Supplement I**.

| Confounding strength | $\gamma = 10$ | $\gamma = 11$ | $\gamma = 12$ | $\gamma = 13$ | $\gamma = 14$ | $\gamma = 15$ | $\gamma = 16$ | $\gamma = 17$ | $\gamma = 18$ | $\gamma = 19$ | $\gamma = 20$ |
|---|---|---|---|---|---|---|---|---|---|---|---|
| CRN (Bica et al., 2020) | $4.05 \pm 0.55$ | $5.45 \pm 1.68$ | $6.17 \pm 1.27$ | $4.98 \pm 1.49$ | $5.24 \pm 0.33$ | $4.84 \pm 0.95$ | $5.41 \pm 1.20$ | $5.09 \pm 0.77$ | $5.08 \pm 0.87$ | $4.47 \pm 0.84$ | $4.80 \pm 0.70$ |
| TE-CDE (Seedat et al., 2022) | $4.08 \pm 0.54$ | $4.21 \pm 0.42$ | $4.33 \pm 0.11$ | $4.48 \pm 0.47$ | $4.39 \pm 0.38$ | $4.67 \pm 0.65$ | $4.84 \pm 0.46$ | $4.31 \pm 0.38$ | $4.44 \pm 0.53$ | $4.61 \pm 0.42$ | $4.72 \pm 0.45$ |
| CT (Melnychuk et al., 2022) | $3.44 \pm 0.73$ | $3.70 \pm 0.77$ | $3.60 \pm 0.62$ | $3.87 \pm 0.68$ | $3.88 \pm 0.75$ | $3.87 \pm 0.65$ | $5.26 \pm 1.67$ | $4.04 \pm 0.74$ | $4.13 \pm 0.90$ | $4.30 \pm 0.72$ | $4.49 \pm 0.94$ |
| RMSNs (Lim et al., 2018) | $3.34 \pm 0.20$ | $3.41 \pm 0.17$ | $3.61 \pm 0.25$ | $3.76 \pm 0.25$ | $3.92 \pm 0.26$ | $4.22 \pm 0.40$ | $4.30 \pm 0.52$ | $4.48 \pm 0.59$ | $4.60 \pm 0.46$ | $4.47 \pm 0.53$ | $4.62 \pm 0.51$ |
| G-transformer (Xiong et al., 2024) | $5.42 \pm 1.67$ | $5.50 \pm 1.76$ | $5.32 \pm 1.85$ | $5.65 \pm 2.01$ | $5.46 \pm 1.97$ | $5.81 \pm 1.88$ | $5.76 \pm 1.70$ | $5.76 \pm 1.63$ | $5.67 \pm 1.84$ | $6.09 \pm 1.85$ | $6.00 \pm 1.89$ |
| G-Net (Li et al., 2021) | $3.51 \pm 0.37$ | $3.71 \pm 0.33$ | $3.80 \pm 0.29$ | $3.89 \pm 0.27$ | $3.91 \pm 0.26$ | $3.94 \pm 0.26$ | $4.05 \pm 0.37$ | $4.09 \pm 0.41$ | $4.22 \pm 0.53$ | $4.21 \pm 0.55$ | $4.24 \pm 0.45$ |
| **IGC-Net** (ours) | $\mathbf{3.13 \pm 0.22}$ | $\mathbf{3.16 \pm 0.14}$ | $\mathbf{3.31 \pm 0.20}$ | $\mathbf{3.27 \pm 0.14}$ | $\mathbf{3.30 \pm 0.11}$ | $\mathbf{3.49 \pm 0.30}$ | $\mathbf{3.53 \pm 0.26}$ | $\mathbf{3.50 \pm 0.26}$ | $\mathbf{3.41 \pm 0.29}$ | $\mathbf{3.59 \pm 0.21}$ | $\mathbf{3.71 \pm 0.27}$ |
| Rel. improvement | 6.4% | 7.3% | 7.9% | 12.9% | 15.0% | 9.9% | 12.9% | 13.1% | 17.4% | 14.8% | 12.5% |

Table 3: **RMSE on synthetic data.** Based on the tumor data with $\tau = 2$. Our IGC-Net consistently outperforms all baselines. We highlight the relative improvement over the best-performing baseline.

• **Synthetic data:** First, we follow common practice in benchmarking for causal inference (Bica et al., 2020; Li et al., 2021; Lim et al., 2018; Melnychuk et al., 2022) and evaluate the performance of our IGC-Net against other baselines on fully synthetic data. The use of synthetic data is beneficial as it allows us to simulate the outcomes under a sequence of interventions, which are unknown in real-world datasets. Thereby, we are able to evaluate the performance of all methods for estimating CAPOs over time. Here, our main aim is to show that our IGC-Net is *robust against increasing levels of confounding*.

For this, we use data based on the pharmacokinetic-pharmacodynamic tumor growth model (Geng et al., 2017), which is a standard dataset for benchmarking causal inference methods in the time-varying setting (Bica et al., 2020; Li et al., 2021; Lim et al., 2018; Melnychuk et al., 2022), and allows for controlling the confounding strength with a parameter $\gamma$. Here, we are interested in the performance of our IGC-Net for increasing levels of confounding. We thus increase the confounding parameter $\gamma$ from $\gamma = 10$ to $\gamma = 20$, and the same parameterization as in (Melnychuk et al., 2022). We report details on the data-generating process in Supplement G.1.

Results: **Table 3** shows the average RMSE over five different runs for a prediction horizon of $\tau = 2$. Of note, we emphasize that our comparison is fair (see hyperparameter tuning in **Supplement I.2**). We make the following observations:

(i) Our **IGC-Net** outperforms all baselines by a significant margin. Importantly, as our IGC-Net performs proper adjustments for time-varying confounding, it is robust against increasing $\gamma$. In particular, our IGC-Net achieves a performance improvement over the best-performing baseline of up to 17.4%.

(ii) The ① baselines that do not perform proper adjustments (i.e., **CRN** (Bica et al., 2020), **TE-CDE** (Seedat et al., 2022), and **CT** (Melnychuk et al., 2022)) exhibit large variations in performance and are thus highly unstable. This is expected, as they do not target the correct causal estimand and, accordingly, suffer from the increasing confounding.

(iii) The baselines with ② problematic adjustment strategies (i.e., **RMSNs** (Lim et al., 2018), **G-Net** (Li et al., 2021)) are slightly more stable than the no-adjustment baselines. This can be attributed to that the tumor growth model has no time-varying covariates $X_t$ and to that we are only focusing on $\tau = 2$-step ahead predictions, both of which reduce the variance. However, the RMSNs and G-Net are still significantly worse than our IGC-Net.

| Training samples | N = 1000 | | | | | N = 2000 | | | | | N = 3000 | | | | |
|---|---|---|---|---|---|---|---|---|---|---|---|---|---|---|---|
| Prediction window | τ = 2 | τ = 3 | τ = 4 | τ = 5 | τ = 6 | τ = 2 | τ = 3 | τ = 4 | τ = 5 | τ = 6 | τ = 2 | τ = 3 | τ = 4 | τ = 5 | τ = 6 |
| CRN (Bica et al., 2020) | 0.42±0.11 | 0.58±0.21 | 0.74±0.31 | 0.84±0.42 | 0.95±0.51 | 0.39±0.12 | 0.50±0.14 | 0.58±0.15 | 0.64±0.16 | 0.70±0.17 | 0.37±0.10 | 0.46±0.11 | 0.56±0.13 | 0.65±0.16 | 0.75±0.24 |
| TE-CDE (Seedat et al., 2022) | 0.76±0.09 | 0.91±0.15 | 1.07±0.22 | 1.15±0.25 | 1.24±0.28 | 0.76±0.16 | 0.87±0.17 | 0.98±0.17 | 1.06±0.18 | 1.14±0.19 | 0.71±0.09 | 0.78±0.09 | 0.88±0.11 | 0.94±0.12 | 1.02±0.13 |
| CT (Melnychuk et al., 2022) | 0.33±0.14 | 0.44±0.18 | 0.53±0.21 | 0.57±0.19 | 0.60±0.19 | 0.31±0.11 | 0.41±0.13 | 0.49±0.15 | 0.55±0.15 | 0.60±0.15 | 0.32±0.10 | 0.40±0.11 | 0.49±0.12 | 0.55±0.13 | 0.61±0.15 |
| RMSNs (Lim et al., 2018) | 0.57±0.16 | 0.73±0.20 | 0.87±0.22 | 0.94±0.20 | 1.02±0.20 | 0.62±0.25 | 0.73±0.21 | 0.85±0.25 | 0.96±0.26 | 1.05±0.28 | 0.66±0.27 | 0.76±0.24 | 0.86±0.23 | 0.93±0.21 | 1.00±0.20 |
| G-transformer (Xiong et al., 2024) | 0.55±0.13 | 0.70±0.14 | 0.81±0.15 | 0.89±0.13 | 0.97±0.14 | 0.53±0.13 | 0.67±0.16 | 0.78±0.19 | 0.86±0.19 | 0.94±0.19 | 0.49±0.10 | 0.62±0.13 | 0.73±0.16 | 0.80±0.18 | 0.87±0.19 |
| G-Net (Li et al., 2021) | 0.56±0.14 | 0.73±0.17 | 0.86±0.18 | 0.95±0.20 | 1.03±0.21 | 0.55±0.12 | 0.73±0.14 | 0.87±0.14 | 1.00±0.22 | 1.12±0.26 | 0.54±0.11 | 0.72±0.16 | 0.88±0.21 | 1.00±0.26 | 1.11±0.32 |
| **IGC-Net (ours)** | **0.30±0.07** | **0.36±0.11** | **0.44±0.13** | **0.47±0.12** | **0.54±0.13** | **0.27±0.07** | **0.32±0.09** | **0.38±0.10** | **0.42±0.08** | **0.45±0.10** | **0.24±0.07** | **0.31±0.08** | **0.36±0.09** | **0.42±0.10** | **0.48±0.10** |
| Rel. improvement | 9.5% | 19.7% | 16.3% | 16.7% | 10.8% | 15.3% | 22.5% | 22.5% | 22.6% | 25.0% | 26.7% | 24.0% | 25.2% | 24.6% | 21.6% |

Table 4: **RMSE on semi-synthetic data based on the MIMIC-III extract.** Our IGC-Net consistently outperforms all baselines. We highlight the relative improvement over the best-performing baseline.

● **Semi-synthetic data:** Next, we study how our IGC-Net performs when (i) the covariate space is *high-dimensional* and when (ii) the *prediction windows τ become larger*. For this, we use semi-synthetic data, which, similar to the fully-synthetic dataset, allows us to access the ground-truth outcomes under an interventional sequence of treatments for benchmarking.

Our data-generating process is taken from (Melnychuk et al., 2022), which builds upon the MIMIC-extract (Wang et al., 2020) based on the MIMIC-III dataset (Johnson et al., 2016). In short, we use $d_x = 25$ different vital signs as time-varying covariates, and simulate observational outcomes for training, and interventional outcomes for testing, respectively. As the covariate space is high-dimensional, we thereby study how robust our IGC-Net is with respect to estimation variance. We further increase the prediction windows from $\tau = 2$ up to $\tau = 6$. We report details on the data-generating process in **Supplement G.1.**

Results: **Table 4** shows the average RMSE over five different runs. Again, we emphasize that our comparison is fair (see hyperparameter tuning in **Supplement I**). We make three observations:

(i) Our **IGC-Net** consistently outperforms all baselines by a large margin. The performance of IGC-Net is robust across all sample sizes $N$. Further, it is stable across different prediction windows. We observe that our IGC-Net has a better performance compared to the best baseline of up to $26.7\%$.

(ii) The ① baselines that do not perform proper adjustments (i.e., **CRN** (Bica et al., 2020), **CT** (Melnychuk et al., 2022)) tend to perform better than baselines with problematic adjustment strategies (i.e., RMSNs (Lim et al., 2018), G-Net (Li et al., 2021)). The reason is that the former baselines are (i) regression-based (ii) do not require IPW pseudo-outcomes. Hence, they can better handle the high-dimensional covariate space. They are, however, biased as they do not adjust for time-varying confounders and thus still perform significantly worse than our IGC-Net.

(iii) The baselines with ② problematic adjustment strategies (i.e., **RMSNs** (Lim et al., 2018), **G-Net** (Li et al., 2021), **G-transformer** (Xiong et al., 2024)) struggle with the high-dimensional covariate space and larger prediction windows $\tau$. This can be expected, as RMSNs suffer from overlap violations and thus produce unstable inverse propensity weights. Similarly, G-Net suffers from the curse of dimensionality, as it requires estimating a $(d_x + d_y) \times (\tau - 1)$-dimensional distribution.

● **Ablation studies** We now compare IGC-Net against two ablations: (i) an *IGC-LSTM ablation* and (ii) a *biased transformer ablation*. For the former, we substitute the multi-input transformer in our IGC-Net with a simple LSTM. For the latter, we use the same transformer backbone as in our main results but directly learn the G-computation heads on the factual data. Thereby, we omit the iterative generation and learning steps and do not perform proper adjustments for time-varying confounding.

**Figure 8** shows the performance of both ablations. (i) We can see that the IGC-LSTM ablation has competitive performance due to our approach to G-computation. Of note, the ICG-LSTM is proposed in this work and thus presents a key novelty by itself. (ii) The *biased transformer* clearly has poor performance due to the absence of proper adjustments, which further highlights the *importance of our iterative generation and learning algorithm.*

● **Real-world data:** We additionally evaluate on the MIMIC-III ICU dataset in **Table 5**, following the setup of Melnychuk et al. (2022), where the goal is to predict factual patient outcomes (e.g., effects of vasopressors and ventilation on diastolic

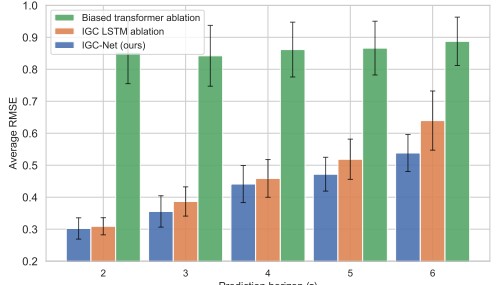

Figure 3: **Ablations.** *IGC-LSTM* has competitive performance, while the *biased transformer* without proper adjustments is inferior.

| Training samples | N = 1000 | | | | | N = 2000 | | | | | N = 3000 | | | | |
|---|---|---|---|---|---|---|---|---|---|---|---|---|---|---|---|
| Prediction window | $\tau = 2$ | $\tau = 3$ | $\tau = 4$ | $\tau = 5$ | $\tau = 6$ | $\tau = 2$ | $\tau = 3$ | $\tau = 4$ | $\tau = 5$ | $\tau = 6$ | $\tau = 2$ | $\tau = 3$ | $\tau = 4$ | $\tau = 5$ | $\tau = 6$ |
| CRN (Bica et al., 2020) | 9.76 ± 0.35 | 10.45 ± 0.41 | 10.82 ± 0.40 | 11.08 ± 0.41 | 11.28 ± 0.43 | 9.61 ± 0.30 | 10.26 ± 0.35 | 10.61 ± 0.31 | 10.90 ± 0.33 | 11.13 ± 0.36 | 9.28 ± 0.46 | 9.93 ± 0.49 | 10.28 ± 0.52 | 10.56 ± 0.51 | 10.78 ± 0.53 |
| TE-CDE (Seedat et al., 2022) | 11.52 ± 0.25 | 11.82 ± 0.29 | 12.05 ± 0.32 | 12.23 ± 0.33 | 12.36 ± 0.35 | 11.05 ± 0.38 | 11.35 ± 0.37 | 11.60 ± 0.39 | 11.77 ± 0.40 | 11.92 ± 0.41 | 10.82 ± 0.30 | 11.13 ± 0.32 | 11.39 ± 0.36 | 11.55 ± 0.39 | 11.70 ± 0.43 |
| CT (Melnychuk et al., 2022) | **9.32 ± 0.38** | **10.02 ± 0.41** | 10.44 ± 0.40 | 10.76 ± 0.43 | 11.00 ± 0.45 | **9.26 ± 0.30** | **9.87 ± 0.35** | 10.23 ± 0.36 | 10.53 ± 0.39 | 10.76 ± 0.41 | **9.05 ± 0.43** | **9.68 ± 0.45** | **10.04 ± 0.47** | 10.32 ± 0.49 | **10.54 ± 0.53** |
| RMSNs (Lim et al., 2018) | 11.32 ± 0.91 | 12.37 ± 0.96 | 13.09 ± 0.97 | 13.57 ± 0.96 | 13.94 ± 0.97 | 11.07 ± 0.98 | 12.21 ± 0.90 | 12.82 ± 0.92 | 12.71 ± 0.97 | 12.80 ± 0.96 | 11.38 ± 0.96 | 12.95 ± 0.95 | 13.40 ± 0.99 | 13.48 ± 0.83 | 13.59 ± 0.86 |
| G-transformer (Xiong et al., 2024) | 11.46 ± 0.43 | 12.77 ± 0.44 | 13.56 ± 0.46 | 14.07 ± 0.47 | 14.44 ± 0.50 | 11.55 ± 0.40 | 12.68 ± 0.41 | 13.32 ± 0.42 | 13.75 ± 0.43 | 14.16 ± 0.47 | 11.58 ± 0.40 | 12.78 ± 0.43 | 13.43 ± 0.44 | 13.82 ± 0.41 | 14.14 ± 0.44 |
| G-Net (Li et al., 2021) | 11.46 ± 0.38 | 12.94 ± 0.40 | 13.90 ± 0.42 | 14.53 ± 0.43 | 15.03 ± 0.45 | 11.87 ± 0.34 | 13.20 ± 0.35 | 14.02 ± 0.38 | 14.58 ± 0.39 | 15.12 ± 0.41 | 11.90 ± 0.38 | 13.17 ± 0.40 | 13.96 ± 0.43 | 14.54 ± 0.06 | 15.03 ± 0.05 |
| **IGC-Net** (ours) | 9.42 ± 0.49 | 10.03 ± 0.52 | 10.43 ± 0.53 | 10.71 ± 0.57 | 10.92 ± 0.59 | 9.32 ± 0.52 | 9.88 ± 0.56 | 10.20 ± 0.56 | 10.49 ± 0.61 | 10.73 ± 0.61 | 9.14 ± 0.53 | 9.76 ± 0.56 | 10.07 ± 0.58 | 10.30 ± 0.28 | 10.62 ± 0.38 |

Table 5: **RMSE on real-world data based on the MIMIC-III extract (best in bold, second-best underlined.** We conduct a *sanity check*, and evaluate all methods on real-world data for *factual outcome prediction*. Of note, methods that perform adjustments for time-varying confounding are **not** primarily tailored for factual outcome prediction, as there are **no** causal interventions. Predicting factuals only requires a simple history-adjustment as in CT, CRN and TE-CDE. Yet, our IGC-Net is **highly competitive**, and is the best-performing method along with CT.

blood pressure). Because no counterfactuals are observed, this experiment serves only as a *sanity check* rather than an evaluation of causal accuracy. Nevertheless, our method achieves state-of-the-art factual prediction performance, which demonstrates that our IGC-Net remains highly effective even when no time-varying adjustment is required, and is directly applicable to real-world clinical data.

• **Overlap sensitivity analysis:** We further evaluate the robustness to overlap violations using the synthetic tumor dataset in **Table 6**. Therein, we scale the treatment-assignment logits with a factor $\rho$. Smaller values of $\rho$ produce well-balanced treatment overlap, whereas larger values push the assignment probabilities toward 0 or 1, which induces increasingly severe overlap violations. Our IGC-Net outperforms all baselines, which demonstrates strong robustness even when overlap deteriorates.

| Overlap | $\rho = 0.5$ | $\rho = 0.6$ | $\rho = 0.7$ | $\rho = 0.8$ | $\rho = 0.9$ | $\rho = 1.0$ | $\rho = 1.1$ | 1.2 | $\rho = 1.3$ | $\rho = 1.4$ | $\rho = 1.5$ |
|---|---|---|---|---|---|---|---|---|---|---|---|
| CRN (Bica et al., 2020) | 2.99 ± 0.26 | 3.62 ± 0.87 | 3.87 ± 0.36 | 4.00 ± 0.52 | 5.34 ± 1.81 | 6.17 ± 1.27 | 5.85 ± 1.03 | 5.30 ± 0.36 | 5.24 ± 0.55 | 5.30 ± 1.51 | 5.49 ± 0.90 |
| TE-CDE (Seedat et al., 2022) | 2.99 ± 0.13 | 3.43 ± 0.22 | 3.75 ± 0.58 | 4.09 ± 0.43 | 4.10 ± 0.43 | 4.29 ± 0.39 | 4.37 ± 0.52 | 4.73 ± 0.33 | 4.78 ± 0.67 | 4.75 ± 0.68 | 4.72 ± 0.75 |
| CT (Melnychuk et al., 2022) | 2.60 ± 0.40 | **2.72 ± 0.17** | 3.33 ± 0.50 | 3.39 ± 0.74 | 4.03 ± 0.83 | 3.59 ± 0.59 | 4.46 ± 1.21 | 4.54 ± 1.70 | 4.91 ± 1.31 | 4.37 ± 0.80 | 4.34 ± 0.95 |
| RMSNs (Lim et al., 2018) | 2.55 ± 0.29 | 2.84 ± 0.31 | 3.35 ± 0.56 | 3.58 ± 0.27 | 3.81 ± 0.47 | 3.70 ± 0.29 | 3.74 ± 0.44 | 4.50 ± 0.51 | 4.09 ± 0.38 | 4.49 ± 0.75 | 4.37 ± 0.46 |
| G-transformer (Xiong et al., 2024) | 3.90 ± 1.32 | 4.74 ± 1.37 | 5.93 ± 1.88 | 4.91 ± 1.50 | 6.34 ± 1.80 | 5.32 ± 1.85 | 6.49 ± 1.87 | 5.64 ± 1.92 | 6.56 ± 2.21 | 5.64 ± 1.77 | 6.70 ± 2.31 |
| G-Net (Li et al., 2021) | 3.14 ± 0.27 | 3.26 ± 0.53 | 4.14 ± 0.74 | 4.03 ± 0.46 | 4.61 ± 0.58 | 4.35 ± 0.45 | 5.01 ± 0.69 | 4.50 ± 0.51 | 5.10 ± 0.67 | 4.67 ± 0.59 | 5.06 ± 0.44 |
| **IGC-Net** (ours) | **2.53 ± 0.14** | 2.78 ± 0.18 | **3.07 ± 0.20** | **3.16 ± 0.12** | **3.36 ± 0.27** | **3.24 ± 0.12** | **3.48 ± 0.32** | **3.39 ± 0.18** | **3.55 ± 0.32** | **3.48 ± 0.19** | **3.85 ± 0.29** |
| Rel. improvement | 0.9% | −2.1% | 7.7% | 6.8% | 11.8% | 10.0% | 7.0% | 15.1% | 13.0% | 20.4% | 11.2% |

Table 6: **RMSE with overlap violations.** Based on the tumor data with $\tau = 2$ and varying levels of **overlap**. *Lower values* of the overlap parameter $\rho$ indicate more *balanced overlap*, whereas *larger values* of $\rho$ skew the overlap towards extreme values close to 0 or 1. Our IGC-Net outperforms all baselines. We highlight the relative improvement over the best-performing baseline.

• **Unobserved confounding:** We perform robustness checks under *unobserved confounding* in **Table 7**. Hence, we introduce an unobserved confounder $U_i \sim \mathcal{N}(0, 1)$ for each patient $i$ in the tumor dataset. The confounder is omitted from the observed state, but influences both treatment assignment and tumor evolution. Treatment logits for chemotherapy and radiotherapy are shifted by an additive term $0.2 \times U_i$, and therefore systematic hidden bias is injected into the assignment policy.

| Parameter | $\omega = 0.000$ | $\omega = 0.005$ | $\omega = 0.010$ | $\omega = 0.015$ | $\omega = 0.020$ |
|---|---|---|---|---|---|
| CRN (Bica et al., 2020) | 4.42 ± 0.83 | 4.87 ± 0.45 | 5.51 ± 0.37 | 5.83 ± 2.02 | 4.65 ± 0.67 |
| TE-CDE (Seedat et al., 2022) | 4.41 ± 0.87 | 4.08 ± 0.26 | 4.01 ± 0.45 | 4.36 ± 0.50 | 4.20 ± 0.26 |
| CT (Melnychuk et al., 2022) | 4.20 ± 1.01 | 4.12 ± 0.65 | 4.29 ± 0.91 | 3.78 ± 0.53 | 4.23 ± 1.06 |
| RMSNs (Lim et al., 2018) | 3.65 ± 0.47 | 4.04 ± 0.61 | 3.77 ± 0.56 | 3.72 ± 0.65 | 4.11 ± 0.46 |
| G-transformer (Xiong et al., 2024) | 6.04 ± 1.72 | 5.93 ± 1.56 | 6.06 ± 1.37 | 5.90 ± 1.19 | 6.06 ± 1.32 |
| G-Net (Li et al., 2021) | 4.77 ± 0.73 | 4.72 ± 0.78 | 4.71 ± 0.73 | 4.78 ± 0.67 | 4.75 ± 0.68 |
| **IGC-Net** (ours) | **3.41 ± 0.27** | **3.52 ± 0.22** | **3.52 ± 0.43** | **3.65 ± 0.21** | **3.61 ± 0.10** |
| Rel. improvement | 6.5% | 12.8% | 6.6% | 1.8% | 12.3% |

Table 7: **RMSE under unobserved confounding.** Our IGC-Net maintains the best performance under increasing confounding $\omega$.

Additionally, the tumor growth is perturbed by adding $\omega \times U_i$ inside the multiplicative growth factor. The scalar coefficients $\omega$ controls the strength of the unobserved confounding. Of note, *none* of the baselines is designed to handle unobserved confounding. Our results show that our IGC-Net remains fairly **robust**, and performance does **not** deteriorate any worse than the baselines.

• **Additional results:** We report additional *ablation studies*, a *sensitivity analysis* of our pseudo-outcomes, *uncertainty quantification with MC dropout and kernel smoothing*, and the *coefficient of variation* in **Supplement F**.

**Conclusion:** In this paper, we propose the IGC-Net, a novel end-to-end method that adjusts for time-varying confounding. Therefore, we expect our IGC-Net to be an important step toward personalized medicine with machine learning.

## ACKNOWLEDGMENTS

This work has been supported by the German Federal Ministry of Education and Research (Grant: 01IS24082). This paper is supported by the DAAD program "Konrad Zuse Schools of Excellence in Artificial Intelligence", sponsored by the Federal Ministry of Education and Research.

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

## A    EXTENDED RELATED WORK

**Estimating CAPOs in the static setting:** Extensive work on estimating potential outcomes focuses on the *static* setting (e.g., Alaa & van der Schaar, 2017; Frauen et al., 2023b; Johansson et al., 2016; Louizos et al., 2017; Melnychuk et al., 2023; Yoon et al., 2018; Zhang et al., 2020)). However, observational data such as electronic health records (EHRs) in clinical settings are typically measured *over time* (Allam et al., 2021; Bica et al., 2021). Additionally, treatments are rarely applied all at once but rather sequentially over time (Apperloo et al., 2024). Therefore, the underlying assumption of these methods prohibitive and does not properly reflect medical reality. Hence, static methods are **not** tailored to accurately estimate potential outcomes when (i) time series data is observed and (ii) multiple treatments in the future are of interest.

**Additional literature on estimating CAPOs over time:** Recently, Frauen et al. (2025); Hess et al. (2026) proposed model-agnostic meta-learners for heterogeneous treatment effect estimation over time. Therein, they analyze theoretically the advantages and disadvantages of several adjustment strategies such as regression adjustment, IPW, and DR estimators. Our IGC-Net uses a similar identification regression-adjustment (RA) approach, in the sense that both rely on the G-formula and estimate the conditional mean of the next outcome given the observed history. However, the core innovation of our IGC-Net lies not in adopting RA as an identification strategy, but in developing a novel end-to-end learning algorithm that implements the full multi-step G-computation recursion within a single neural architecture. Instead of fitting a separate model at each time step, our method couples representation learning with iterative pseudo-outcome learning+generation, which enables joint optimization across all time-steps.

There are some non-parametric methods for this task (Schulam & Saria, 2017; Soleimani et al., 2017; Xu et al., 2016), yet these suffer from poor scalability and have limited flexibility regarding the outcome distribution, the dimension of the outcomes, and static covariate data; because of that, we do not explore non-parametric methods further but focus on neural methods instead.

Other works are orthogonal to ours. For example, (Hess et al., 2024; Vanderschueren et al., 2023) are approaches for informative sampling and uncertainty quantification, respectively. However, they do not focus on the causal structure, and are therefore *not* primarily designed for our task of interest. Further, Hess & Feuerriegel (2025) propose the first neural method for proper causal adjustments in continuous time, which is a different stream of literature, and is not tailored for our discrete time setting. Ma et al. (2025) propose a neural method that imposes parametric assumptions on the DGP.

Deng et al. (2025) do not introduce a new estimator for time-varying conditional treatment effects and CAPOs; instead, they take existing models as fixed bases and add approximate Bayesian uncertainty quantification layers (deep ensembles, variational dropout, SWAG) on top. Their focus is on improving calibration and decision-making by modeling parameter uncertainty, not on changing the underlying CAPO estimand or addressing bias/variance trade-offs of the point estimates themselves. As such, their work is orthogonal to ours and could be applied on top of our IGC-Net as well.

Further, Wang et al. (2025b) and Wu et al. (2024) try to estimate the entire distribution of CAPOs over time, which is a different task from ours: their estimand is the counterfactual density, not the conditional mean potential outcome (CAPO) that our paper targets. This task fundamentally differs from ours, as they require learning high-dimensional conditional distributions and sampling trajectories and are therefore unsuitable for our task.

Finally, Wang et al. (2025a) propose a state-space-model propose a framework with decorrelation regularization. However, its learning objective is different in two essential ways: It does not perform proper adjustments and therefore does not identify the CAPO under sequential ignorability. Instead, it regularizes correlations between hidden states and treatments, which does not correspond to any identified estimand in longitudinal causal inference. It directly predicts outcomes from hidden states learned under de-correlation penalties. As a result, its outputs are not guaranteed to estimate the CAPO.

**Survival analysis:** Some works in survival analysis (Andersen & Perme, 2010; Andersen et al., 2017; Su et al., 2022) employ pseudo-outcomes, which is similar to our approach. However, these works are different in that they are aimed at *survival outcomes* and **not** CAPOs for sequences of treatments. Further, they do **not** consider neural networks as estimators. Additionally, (Andersen et al., 2017) only considers a **single, static treatment**, and (Andersen & Perme, 2010) only uses

**linear** estimators. Finally, (Su et al., 2022) focuses on **average** causal effects and is therefore not applicable to personalized medicine.

**G-computation and Q-learning:** Q-learning (Murphy, 2003; Kallus & Uehara, 2019) from the reinforcement learning literature (Furuta et al., 2022; Jang et al., 2022; Kumar et al., 2019; Pashevich et al., 2021; Javurek et al., 2026) is closely related to G-computation, although both have a different purpose. They are similar in that they share a common goal of understanding the effect of treatments/actions, but operate in complementary domains: G-computation is grounded in causal inference for evaluating potential outcomes, whereas Q-learning is rooted in reinforcement learning to derive *policies that maximize long-term rewards*. We show more details on the two in the following:

G-computation can be written as the iterative update

$$g_{t+\delta}^{\bar{a}}(\bar{h}_{t+\delta}^t) = \mathbb{E}[G_{t+\delta+1}^{\bar{a}} \mid \bar{H}_{t+\delta}^t = \bar{h}_{t+\delta}^t, A_{t:t+\delta} = a_{t:t+\delta}], \tag{21}$$

In our setting, we aim to estimate $\mathbb{E}\left[Y_{t+\tau}[a_{t:t+\tau-1}] \mid \bar{H}_t = \bar{h}_t\right]$.

However, we could also consider the expected *cumulative rewards* $\mathbb{E}\left[\bar{Y}_{t+\tau}[a_{t:t+\tau-1}] \mid \bar{H}_t = \bar{h}_t\right]$, where we define $\bar{Y}_{t+\tau}[a_{t:t+\tau-1}] = \sum_{\ell=1}^{t+\tau} \gamma^\ell Y_{t+\ell}[a_{t:t+\ell-1}]$ and where $\gamma < 1$ is a so-called discount factor that weighs the importance of immediate and future rewards. One can show that the G-computation update becomes

$$g_{t+\delta}^{\bar{a}}(\bar{h}_{t+\delta}^t) = \mathbb{E}[Y_{t+\delta} + \gamma G_{t+\delta+1}^{\bar{a}} \mid \bar{H}_{t+\delta}^t = \bar{h}_{t+\delta}^t, A_{t:t+\delta} = a_{t:t+\delta}]. \tag{22}$$

If we only care about the *optimal* treatment sequence $a^*$ (i.e., the one that maximizes the cumulative reward), we can write

$$g_{t+\delta}^{a^*}(\bar{h}_{t+\delta}^t) = \mathbb{E}[Y_{t+\delta} + \gamma \max_{a_{t+\delta+1}^*} G_{t+\delta+1}^{a^*} \mid \bar{H}_{t+\delta}^t = \bar{h}_{t+\delta}^t, A_{t:t+\delta} = a_{t:t+\delta}^*]. \tag{23}$$

Eq. equation 23 is known as *Q-learning* in the literature on dynamic treatment regimes (Murphy, 2003; Kallus & Uehara, 2019) and can be used to compute an optimal dynamic policy.

In reinforcement learning, one often makes *additional* Markov and stationarity assumptions such that the history $\bar{h}_{t+\delta}^t$ simplifies to a single state $s_{t+\delta}$ and the function $g^{a_t^*}(s_t)$ is not dependent on time. These assumptions allow us to consider infinite time-horizons and break the so-called curse of horizon (Kallus & Uehara, 2022; Uehara et al., 2022). Then, Q-learning simplifies to

$$g^{a_t^*}(s_t) = \mathbb{E}[Y_t + \gamma \max_{a_{t+1}^*} G^{a^*} \mid S_t = s_t, A_t = a_t^*], \tag{24}$$

which is often called *fitted Q-iteration* in the RL literature (Kallus & Uehara, 2020; Uehara et al., 2022). In contrast, our work does not make these assumptions.

State-of-the-art neural instantiations such as (Chebotar et al., 2023) are *different* to our work in that they (i) serve the purpose of *learning long-term rewards*, and (ii) rely on *restrictive Markov* assumptions. In contrast, our IGC-Net is designed to estimate CAPOs for sequences of treatments, conditionally on the entire individual patient history.

# B DISCUSSION ON IDENTIFIABILITY ASSUMPTIONS AND CLINICAL RELEVANCE

In this work, we present a novel neural network, the iterative G-computation network, for estimating conditional average potential outcomes (CAPOs) from observational data such as electronic health records (EHRs). Our IGC-Net addresses a **crucial question in personalized medicine**: *"What would the outcome be for patient X if they were administered treatments A, B, and C sequentially over the next 5 days, given their unique clinical history?"* Unlike many existing methods that focus on static or single-point interventions (Alaa & van der Schaar, 2017; Johansson et al., 2016; Zhang et al., 2020), our method is specifically designed to *handle the sequential nature of treatments in medical practice* – a feature that is both realistic and necessary, as treatments are rarely applied all at once but rather sequentially over time (Apperloo et al., 2024). With the growing availability of large-scale observational data from EHRs (Allam et al., 2021; Feuerriegel et al., 2024; Bica et al., 2021) and wearable devices (Battalio et al., 2021), there is an increasing need for robust methods that estimate the effect of multiple treatments, given the *individual* patient history.

Our framework builds on three key assumptions: (i) consistency, (ii) positivity, and (iii) sequential ignorability (see Section 3). These assumptions are the *standard* assumptions for estimating CAPOs over time (Bica et al., 2020; Li et al., 2021; Melnychuk et al., 2022; Seedat et al., 2022). Notably, compared to other methods that rely on even *stricter* assumptions, such as additional Markov or independence assumptions (Özyurt et al., 2021), our assumptions are *less* restrictive. Furthermore, these assumptions are the *dynamic* analogues of the standard causal inference assumptions in *static* settings (Alaa & van der Schaar, 2017; Muandet et al., 2021; Johansson et al., 2016). Importantly, methods for the static setting implicitly impose *unrealistic assumption* that treatments occur only once and that covariates and outcomes remain static over time. Such limitations can introduce significant bias in sequential decision-making contexts. In contrast, our approach models the time-varying nature of clinical interventions and patient evolution, making it less restrictive and far more aligned with real-world medical scenarios.

Further, we argue that these assumptions are both plausible and practical in medical applications. First, consistency is generally satisfied as long as EHR data is accurately and systematically recorded. Second, positivity can be ensured through thoughtful data pre-processing, such as filtering observations or applying propensity clipping. Additionally, as the scale of observational datasets grows, this assumption becomes less restrictive. Third, the sequential ignorability assumption is a standard assumption in epidemiology (Little & Rubin, 2000), and studies in digital health interventions may satisfy this assumption by design. Furthermore, advances in sensitivity analysis (Frauen et al., 2023b; Oprescu et al., 2023) and partial identification (Duarte et al., 2023) offer complementary pathways to relax this assumption. That is, these literature streams are *orthogonal* to our work. In practice, our IGC-Net thus integrates into established workflows that include point estimation, uncertainty quantification, and sensitivity analysis.

From a practical perspective, our IGC-Net addresses key challenges in estimating CAPOs for sequences of treatments. Specifically, our IGC-Net provides a neural end-to-end solution that adjusts for time-varying confounding. On top, it neither relies on large-variance pseudo-outcomes (Proposition 3) nor on estimating high-dimensional probability distributions. Therefore, we are convinced that our IGC-Net is an important step towards reliable personalized medicine.

## C PROOFS

### C.1 UNBIASED ESTIMAND

**Proposition 1.** *Our regression-based iterative G-computation yields the CAPO in Equation 1.*

*Proof.* For the proof, we only need to apply the definition of the pseudo-outcomes $G_{t+\delta}^{\bar{a}}$:

$$\mathbb{E}[Y_{t+\tau}[a_{t:t+\tau-1}] \mid \bar{H}_t = \bar{h}_t] \tag{25}$$

$$=\mathbb{E}\bigg\{\mathbb{E}\bigg[\ldots\mathbb{E}\big\{\mathbb{E}[Y_{t+\tau} \mid \bar{H}_{t+\tau-1}^t, A_{t:t+\tau-1} = a_{t:t+\tau-1}] \mid \bar{H}_{t+\tau-2}^t, A_{t:t+\tau-2} = a_{t:t+\tau-2}\big\}$$

$$\ldots \bigg| \bar{H}_{t+1}^t, A_{t:t+1} = a_{t:t+1}\bigg] \bigg| \bar{H}_t = \bar{h}_t, A_t = a_t\bigg\} \tag{26}$$

$$=\mathbb{E}\bigg\{\mathbb{E}\bigg[\ldots\mathbb{E}\big\{\mathbb{E}[G_{t+\tau}^{\bar{a}} \mid \bar{H}_{t+\tau-1}^t, A_{t:t+\tau-1} = a_{t:t+\tau-1}] \mid \bar{H}_{t+\tau-2}^t, A_{t:t+\tau-2} = a_{t:t+\tau-2}\big\}$$

$$\ldots \bigg| \bar{H}_{t+1}^t, A_{t:t+1} = a_{t:t+1}\bigg] \bigg| \bar{H}_t = \bar{h}_t, A_t = a_t\bigg\} \tag{27}$$

$$=\mathbb{E}\bigg\{\mathbb{E}\bigg[\ldots\mathbb{E}\big\{g_{t+\tau-1}^{\bar{a}}(\bar{H}_{t+\tau-1}^t) \mid \bar{H}_{t+\tau-2}^t, A_{t:t+\tau-2} = a_{t:t+\tau-2}\big\}$$

$$\ldots \bigg| \bar{H}_{t+1}^t, A_{t:t+1} = a_{t:t+1}\bigg] \bigg| \bar{H}_t = \bar{h}_t, A_t = a_t\bigg\} \tag{28}$$

$$=\mathbb{E}\bigg\{\mathbb{E}\bigg[\ldots\mathbb{E}\big\{G_{t+\tau-1}^{\bar{a}} \mid \bar{H}_{t+\tau-2}^t, A_{t:t+\tau-2} = a_{t:t+\tau-2}\big\}$$

$$\ldots \bigg| \bar{H}_{t+1}^t, A_{t:t+1} = a_{t:t+1}\bigg] \bigg| \bar{H}_t = \bar{h}_t, A_t = a_t\bigg\} \tag{29}$$

$$=\mathbb{E}\bigg\{\mathbb{E}\bigg[\ldots g_{t+\tau-2}^{\bar{a}}(\bar{H}_{t+\tau-2}^t)\ldots \bigg| \bar{H}_{t+1}^t, A_{t:t+1} = a_{t:t+1}\bigg] \bigg| \bar{H}_t = \bar{h}_t, A_t = a_t\bigg\} \tag{30}$$

$$= \ldots \tag{31}$$

$$=\mathbb{E}\bigg\{G_{t+1}^{\bar{a}} \bigg| \bar{H}_t = \bar{h}_t, A_t = a_t\bigg\} \tag{32}$$

$$=g_t^{\bar{a}}(\bar{h}_t), \tag{33}$$

where Equation 26 holds due the G-computation formula (see Supplement D). □

## C.2 TARGET OF OUR IGC-NET

**Proposition 2.** *Our IGC-Net estimates the G-computation formula as in Equation 10 and, therefore, performs proper adjustments for time-varying confounding.*

*Proof.* For the proof, we perform the steps as in Supplement C.1:

$$\hat{\mathbb{E}}[Y_{t+\tau}[a_{t:t+\tau-1}] \mid \bar{H}_t = \bar{h}_t] \tag{34}$$

$$=\hat{\mathbb{E}}\Big\{\hat{\mathbb{E}}\Big[\ldots\hat{\mathbb{E}}\big\{\hat{\mathbb{E}}[Y_{t+\tau} \mid \bar{H}^t_{t+\tau-1}, A_{t:t+\tau-1} = a_{t:t+\tau-1}] \mid \bar{H}^t_{t+\tau-2}, A_{t:t+\tau-2} = a_{t:t+\tau-2}\big\}$$

$$\ldots\Big|\bar{H}^t_{t+1}, A_{t:t+1} = a_{t:t+1}\Big]\Big|\bar{H}_t = \bar{h}_t, A_t = a_t\Big\} \tag{35}$$

$$=\hat{\mathbb{E}}\Big\{\hat{\mathbb{E}}\Big[\ldots\hat{\mathbb{E}}\big\{\hat{\mathbb{E}}[\tilde{G}^{\bar{a}}_{t+\tau} \mid \bar{H}^t_{t+\tau-1}, A_{t:t+\tau-1} = a_{t:t+\tau-1}] \mid \bar{H}^t_{t+\tau-2}, A_{t:t+\tau-2} = a_{t:t+\tau-2}\big\}$$

$$\ldots\Big|\bar{H}^t_{t+1}, A_{t:t+1} = a_{t:t+1}\Big]\Big|\bar{H}_t = \bar{h}_t, A_t = a_t\Big\} \tag{36}$$

$$=\hat{\mathbb{E}}\Big\{\hat{\mathbb{E}}\Big[\ldots\hat{\mathbb{E}}\big\{g^{\phi}_{\tau-1}(a_{t+\tau-1}, z^{\phi}(\bar{H}_{t+\tau-1}, a_{t:t+\tau-2})) \mid \bar{H}^t_{t+\tau-2}, A_{t:t+\tau-2} = a_{t:t+\tau-2}\big\}$$

$$\ldots\Big|\bar{H}^t_{t+1}, A_{t:t+1} = a_{t:t+1}\Big]\Big|\bar{H}_t = \bar{h}_t, A_t = a_t\Big\} \tag{37}$$

$$=\hat{\mathbb{E}}\Big\{\hat{\mathbb{E}}\Big[\ldots\hat{\mathbb{E}}\big\{\tilde{G}^{\bar{a}}_{t+\tau-1} \mid \bar{H}^t_{t+\tau-2}, A_{t:t+\tau-2} = a_{t:t+\tau-2}\big\}$$

$$\ldots\Big|\bar{H}^t_{t+1}, A_{t:t+1} = a_{t:t+1}\Big]\Big|\bar{H}_t = \bar{h}_t, A_t = a_t\Big\} \tag{38}$$

$$=\hat{\mathbb{E}}\Big\{\hat{\mathbb{E}}\Big[\ldots g^{\phi}_{\tau-2}(a_{t+\tau-2}, z^{\phi}(\bar{H}_{t+\tau-2}, a_{t:t+\tau-3}))\ldots\Big|\bar{H}^t_{t+1}, A_{t:t+1} = a_{t:t+1}\Big]\Big|\bar{H}_t = \bar{h}_t, A_t = a_t\Big\} \tag{39}$$

$$=\ldots \tag{40}$$

$$=\hat{\mathbb{E}}\Big\{\tilde{G}^{\bar{a}}_{t+1}\Big|\bar{H}_t = \bar{h}_t, A_t = a_t\Big\} \tag{41}$$

$$=g^{\phi}_0(a_t, z^{\phi}(\bar{h}_t)). \tag{42}$$

$\square$

### C.3 Variance of inverse propensity weighting

In this section, we compare two possible approaches to adjust for time-varying confounders: G-computation and inverse propensity weighting (IPW) (Robins & Hernán, 2009; Robins et al., 2000), which is leveraged by the existing baselines, namely, the RMSNs (Lim et al., 2018), and continuous time versions as in (Hess & Feuerriegel, 2025).

For a fair comparison of G-computation and IPW, we compare the *variance of the ground-truth pseudo-outcomes* that each method relies on – that is, the $G_{t+\delta}^{\bar{a}}$ of our IGC-Net and the inverse propensity weighted outcomes of RMSNs. Importantly, a larger variance of the pseudo-outcomes will directly translate into a larger variance of the respective estimator. We find that IPW leads to a larger variance, which is why we prefer G-computation in our IGC-Net.

**Proposition 3.** *Pseudo-outcomes constructed via inverse propensity weighting have larger variance than pseudo-outcomes in our iterative G-computation network.*

*Proof.* To simplify notation, we consider the variance of the pseudo-outcomes in the *static setting*. The analog directly translates into the time-varying setting.

Let $Y$ be the outcome, $X$ the covariates, and $A$ the treatment. Without loss of generality, we consider the potential outcome for $A = 1$.

For G-computation, the variance of the pseudo-outcome $g^1(X)$ is given by

$$\text{Var}[g^1(X)] = \text{Var}[\mathbb{E}[Y \mid X, A = 1]] \tag{43}$$

$$= \mathbb{E}\Big[\mathbb{E}[Y \mid X, A = 1]^2\Big] - \mathbb{E}\Big[\mathbb{E}[Y \mid X, A = 1]\Big]^2 \tag{44}$$

$$= \mathbb{E}\Big[\mathbb{E}[Y \mid X, A = 1]^2\Big] - \mathbb{E}\Big[Y[1]\Big]^2. \tag{45}$$

For IPW, the variance of the pseudo-outcome is

$$\text{Var}\Big[\frac{YA}{\pi(X)}\Big] = \mathbb{E}\Big[\Big(\frac{YA}{\pi(X)}\Big)^2\Big] - \mathbb{E}\Big[\frac{YA}{\pi(X)}\Big]^2 \tag{46}$$

$$= \mathbb{E}\Big[\mathbb{E}\Big[\frac{Y^2 A}{\pi^2(X)} \mid X\Big]\Big] - \mathbb{E}\Big[Y[1]\Big]^2 \tag{47}$$

$$= \mathbb{E}\Big[\mathbb{E}\Big[\frac{Y^2 \pi(X)}{\pi^2(X)} \mid X, A = 1\Big]\Big] - \mathbb{E}\Big[Y[1]\Big]^2 \tag{48}$$

$$= \mathbb{E}\Big[\underbrace{\frac{1}{\pi(X)}}_{\geq 1} \mathbb{E}[Y^2 \mid X, A = 1]\Big] - \mathbb{E}\Big[Y[1]\Big]^2, \tag{49}$$

and, with

$$\mathbb{E}[Y \mid X, A = 1]^2 + \underbrace{\text{Var}[Y \mid X, A = 1]}_{\geq 0} = \mathbb{E}[Y^2 \mid X, A = 1], \tag{50}$$

we have that

$$\text{Var}\Big[\frac{YA}{\pi(X)}\Big] \geq \text{Var}[g^1(X)]. \tag{51}$$

Therefore, we conclude that G-computation leads to a lower variance than IPW, and, hence, our IGC-Net has a lower variance than RMSNs. □

**Remarks:**

- The inverse propensity weight is what really drives the difference in variance between the approaches. Note that, in the time-varying setting, IPW relies on *products of inverse propensities*, which can lead to even more extreme weights for multi-step ahead predictions. This is expected: propensity weights are typically small and close to zero, especially in time-series settings. Hence, by diving to a quantity close to zero, the variance can naturally explode.

- IPW is particularly problematic when there are overlap violations in the data, as this implies propensity scores close to zero and thus division by values that are close to zero. However, as the input history $\bar{H}_t$ in the time-varying setting is very high-dimensional (i.e., $t \times (d_x + d_y)$-dimensional), overlap violations are even more problematic. This is another advantage of our method.

# D DERIVATION OF G-COMPUTATION FOR CAPOS

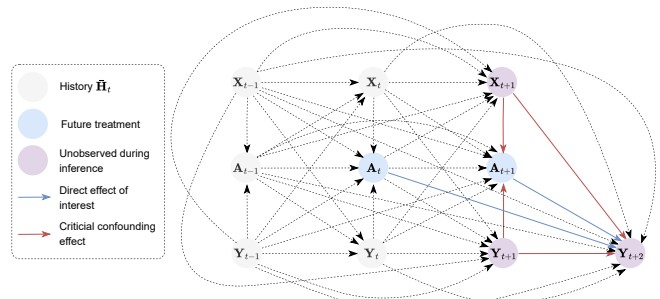

Figure 4: During inference, future time-varying confounders are *unobserved* (here: $(X_{t+1}, Y_{t+1})$). In order to estimate CAPOs for an interventional treatment sequence without **time-varying confounding bias**, proper causal adjustments such as G-computation are required.

In the following, we provide a derivation of the G-computation formula (Bang & Robins, 2005; Robins, 1999; Robins & Hernán, 2009) for CAPOs over time. Recall that G-computation for CAPOs is given by

$$
\mathbb{E}[Y_{t+\tau}[a_{t:t+\tau-1}] \mid \bar{H}_t = \bar{h}_t]
$$
$$
= \mathbb{E}\bigg\{\mathbb{E}\bigg[\dots\mathbb{E}\big\{\mathbb{E}[Y_{t+\tau} \mid \bar{H}^t_{t+\tau-1}, A_{t:t+\tau-1} = a_{t:t+\tau-1}] \mid \bar{H}^t_{t+\tau-2}, A_{t:t+\tau-2} = a_{t:t+\tau-2}\big\} \quad (52)
$$
$$
\dots \bigg|\bar{H}^t_{t+1}, A_{t:t+1} = a_{t:t+1}\bigg]\bigg|\bar{H}_t = \bar{h}_t, A_t = a_t\bigg\}.
$$

The following derivation follows the steps in (Frauen et al., 2023a) and extends them to CAPOs:

$$
\mathbb{E}[Y_{t+\tau}[a_{t:t+\tau-1}] \mid \bar{H}_t = \bar{h}_t]
$$
$$
= \mathbb{E}[Y_{t+\tau}[a_{t:t+\tau-1}] \mid \bar{H}_t = \bar{h}_t, A_t = a_t] \quad (53)
$$
$$
= \mathbb{E}[\mathbb{E}\{Y_{t+\tau}[a_{t:t+\tau-1}] \mid \bar{H}^t_{t+1}, A_t = a_t\} \quad (54)
$$
$$
\mid \bar{H}_t = \bar{h}_t, A_t = a_t]
$$
$$
= \mathbb{E}[\mathbb{E}\{Y_{t+\tau}[a_{t:t+\tau-1}] \mid \bar{H}^t_{t+1}, A_{t:t+1} = a_{t:t+1}\} \quad (55)
$$
$$
\mid \bar{H}_t = \bar{h}_t, A_t = a_t]
$$
$$
= \mathbb{E}[\mathbb{E}\{\mathbb{E}[Y_{t+\tau}[a_{t:t+\tau-1}] \mid \bar{H}^t_{t+2}, A_{t:t+1} = a_{t:t+1}] \quad (56)
$$
$$
\mid \bar{H}^t_{t+1}, A_{t:t+1} = a_{t:t+1}\}
$$
$$
\mid \bar{H}_t = \bar{h}_t, A_t = a_t]
$$
$$
= \mathbb{E}[\mathbb{E}\{\mathbb{E}[Y_{t+\tau}[a_{t:t+\tau-1}] \mid \bar{H}^t_{t+2}, A_{t:t+2} = a_{t:t+2}] \quad (57)
$$
$$
\mid \bar{H}^t_{t+1}, A_{t:t+1} = a_{t:t+1}\}
$$
$$
\mid \bar{H}_t = \bar{h}_t, A_t = a_t]
$$
$$
= \dots
$$
$$
= \mathbb{E}[\dots\mathbb{E}\{\mathbb{E}[Y_{t+\tau}[a_{t:t+\tau-1}] \mid \bar{H}^t_{t+\tau-1}, A_{t:t+\tau-1} = a_{t:t+\tau-1}] \quad (58)
$$
$$
\mid \bar{H}^t_{t+\tau-2}, A_{t:t+\tau-2} = a_{t:t+\tau-2}\}
$$
$$
\mid \dots
$$
$$
\mid \bar{H}_t = \bar{h}_t, A_t = a_t]
$$
$$
= \mathbb{E}[\dots\mathbb{E}\{\mathbb{E}[Y_{t+\tau} \mid \bar{H}^t_{t+\tau-1}, A_{t:t+\tau-1} = a_{t:t+\tau-1}] \quad (59)
$$
$$
\mid \bar{H}^t_{t+\tau-2}, A_{t:t+\tau-2} = a_{t:t+\tau-2}\}
$$
$$
\mid \dots
$$
$$
\mid \bar{H}_t = \bar{h}_t, A_t = a_t],
$$

where Equation 53 follows from the positivity and sequential ignorability assumptions, Equation 54 holds due to the law of total probability, Equation 55 again follows from the positivity and sequential ignorability assumptions, Equation 56 is the tower rule, Equation 57 is again due to the positivity and sequential ignorability assumptions, Equation 58 follows by iteratively repeating the previous steps, and Equation 59 follows from the consistency assumption.

# E    EXAMPLES

To illustrate how regression-based iterative G-computation works, we apply the procedure to two examples. First, we show the trivial case for $(\tau = 1)$-step-ahead predictions and, then, for $(\tau = 2)$-step-ahead predictions. Recall that the following only holds under our standard assumptions (i) *consistency*, (ii) *positivity*, and (iii) *sequential ignorability*.

• $(\tau = 1)$-step-ahead prediction:

This is the trivial case, as there is *no time-varying confounding*. Instead, all confounders are observed in the history. Therefore, we can simply condition on the observed history and resemble the *backdoor-adjustment* from the static setting. Importantly, this is **not** the focus of our work, but we show it for illustrative purposes:

$$\mathbb{E}\big[Y_{t+1}[a_t] \mid \bar{H}_t = \bar{h}_t\big] \tag{60}$$

$$\underbrace{=}_{\text{Ass. (ii)+(iii)}} \mathbb{E}\big[Y_{t+1}[a_t] \mid \bar{H}_t = \bar{h}_t, A_t = a_t\big] \tag{61}$$

$$\underbrace{=}_{\text{Ass. (i)}} \mathbb{E}\big[Y_{t+1} \mid \bar{H}_t = \bar{h}_t, A_t = a_t\big] \tag{62}$$

$$\underbrace{=}_{\text{Def. } G^{\bar{a}}_{t+1}} \mathbb{E}\big[G^{\bar{a}}_{t+1} \mid \bar{H}_t = \bar{h}_t, A_t = a_t\big] \tag{63}$$

$$\underbrace{=}_{\text{Def. } g^{\bar{a}}_t} g^{\bar{a}}_t(\bar{h}_t). \tag{64}$$

• $(\tau = 2)$-step-ahead prediction:

Importantly, $(\tau = 2)$-step-ahead predictions already incorporate all the difficulties that are present for multi-step ahead predictions. Here, we need to account for future time-varying confounders such as $(X_{t+1}, Y_{t+1})$ as in Figure 4:

$$\mathbb{E}\big[Y_{t+2}[a_{t:t+1}] \mid \bar{H}_t = \bar{h}_t\big] \tag{65}$$

$$\underbrace{=}_{\text{Ass. (ii)+(iii)}} \mathbb{E}\big[Y_{t+2}[a_{t:t+1}] \mid \bar{H}_t = \bar{h}_t, A_t = a_t\big] \tag{66}$$

$$\underbrace{=}_{\text{Law of total prob.}} \mathbb{E}\Big[\mathbb{E}\big[Y_{t+2}[a_{t:t+1}] \mid \bar{H}^t_{t+1}, A_t = a_t\big] \mid \bar{H}_t = \bar{h}_t, A_t = a_t\Big] \tag{67}$$

$$\underbrace{=}_{\text{Ass. (ii)+(iii)}} \mathbb{E}\Big[\mathbb{E}\big[Y_{t+2}[a_{t:t+1}] \mid \bar{H}^t_{t+1}, A_{t:t+1} = a_{t:t+1}\big] \mid \bar{H}_t = \bar{h}_t, A_t = a_t\Big] \tag{68}$$

$$\underbrace{=}_{\text{Ass. (i)}} \mathbb{E}\Big[\mathbb{E}\big[Y_{t+2} \mid \bar{H}^t_{t+1}, A_{t:t+1} = a_{t:t+1}\big] \mid \bar{H}_t = \bar{h}_t, A_t = a_t\Big] \tag{69}$$

$$\underbrace{=}_{\text{Def. } G^{\bar{a}}_{t+2}} \mathbb{E}\Big[\mathbb{E}\big[G^{\bar{a}}_{t+2} \mid \bar{H}^t_{t+1}, A_{t:t+1} = a_{t:t+1}\big] \mid \bar{H}_t = \bar{h}_t, A_t = a_t\Big] \tag{70}$$

$$\underbrace{=}_{\text{Def. } g^{\bar{a}}_{t+1}} \mathbb{E}\big[g^{\bar{a}}_{t+1}(\bar{H}^t_{t+1}) \mid \bar{H}_t = \bar{h}_t, A_t = a_t\big] \tag{71}$$

$$\underbrace{=}_{\text{Def. } G^{\bar{a}}_{t+1}} = \mathbb{E}\big[G^{\bar{a}}_{t+1} \mid \bar{H}_t = \bar{h}_t, A_t = a_t\big] \tag{72}$$

$$\underbrace{=}_{\text{Def. } g^{\bar{a}}_t} g^{\bar{a}}_t(\bar{h}_t). \tag{73}$$

# F  ADDITIONAL RESULTS

In the following, we report additional results:

1. We first compare our **ablations** from Section 5 to the baselines and report **additional prediction windows for semi-synthetic data** in Supplement F.1.

2. We perform a **sensitivity analysis** w.r.t. our generated pseudo-outcomes in Supplement F.2 and, thereby, confirm that our IGC-Net generates meaningful pseudo-outcomes in the iterative learning algorithm.

3. We finally report the **coefficient of variation** of our IGC-Net and the baselines in Supplement F.4, which further supports the stability of our IGC-Net.

## F.1  ADDITIONAL RESULTS AND ABLATIONS

In the following, we report the performance of two ablations: the **(A) IGC-LSTM** and the **(B) biased transformer (BT)**. For this, we show **(C) additional results** of our IGC-Net, the baselines, and the two ablations in **Figure 5** and **Figure 6**.

• **IGC-LSTM:** Our first ablation is the *IGC-LSTM*. For this, we replaced the transformer backbone $z^\phi(\cdot)$ of our IGC-Net by a simple LSTM network. We find that our **IGC-LSTM is highly effective**: it outperforms all baselines from the literature while our proposed IGC-transformer is still superior. This demonstrates that our novel method for iterative regression-based G-computation is both effective and general.

• **BT:** We implement a *biased transformer (BT)*. Here, we leverage the same transformer backbone $z^\phi(\cdot)$ as in our IGC-Net, but we directly train the output heads $g_\delta^\phi(\cdot)$ on the factual data. Thereby, the BT refrains from performing G-computation. We can thus isolate the contribution of the iterative G-computation to the overall performance. Our results show that the **BT suffers from significant bias** and, therefore, demonstrate that our proper adjustments for time-varying confounders are required for accurate estimates of the counterfactual outcomes.

We report additional results on

1. tumor growth data, where we report the performance of all methods for **lower levels of confounding** in **Figure 5**.

2. on MIMIC-III semi-synthetic data, where we report **additional prediction windows** up to $\tau = 12$ for $N = 1000$ in **Figure 6**.

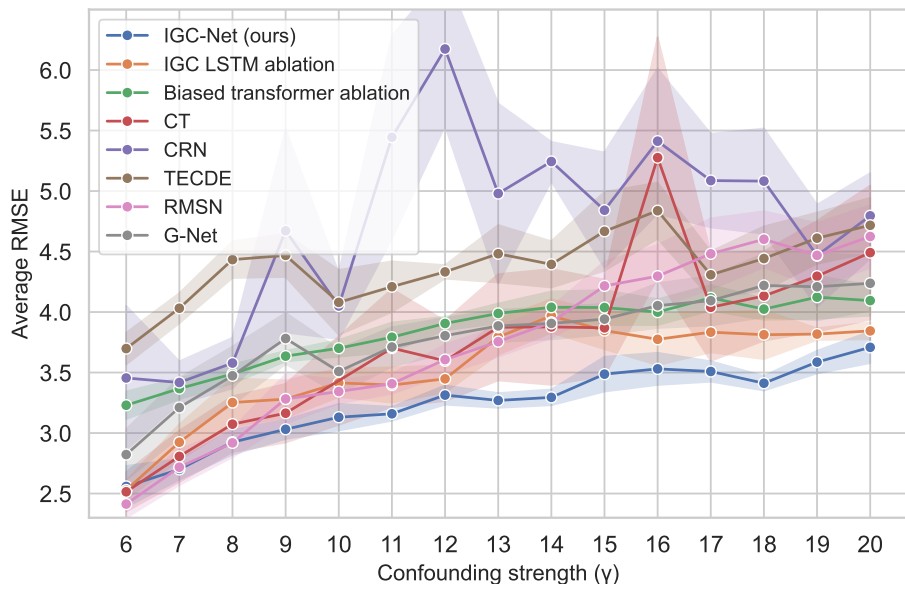

Figure 5: Tumor growth data: We report previous results of the baselines with the **new ablations: IGC-LSTM and BT**. → Notably, our IGC-LSTM has competitive performance, while BT suffers from significant bias. Our *IGC-transformer remains the strongest method*.

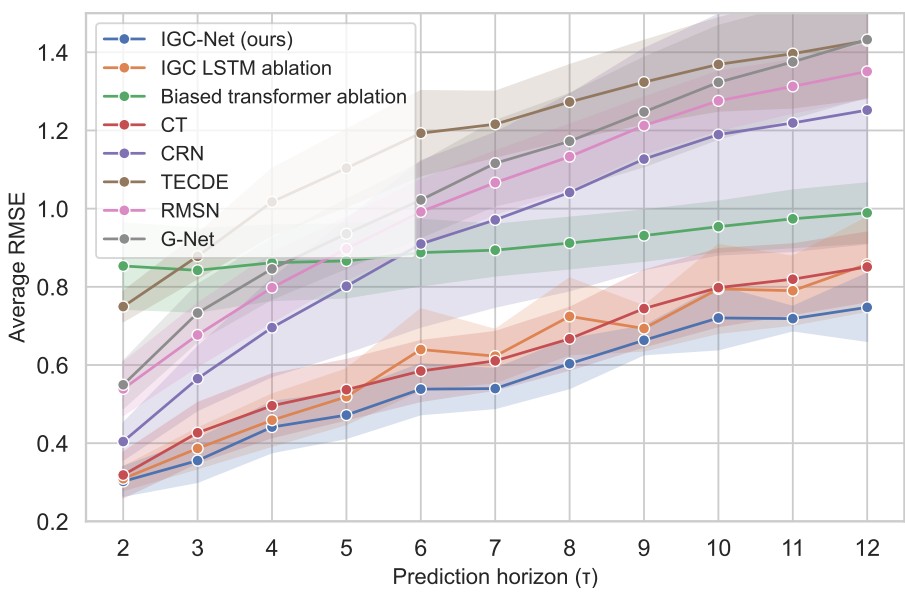

Figure 6: Semi-synthetic data: We **increase the prediction horizon** up to $\tau = 12$ for $N = 1000$ training samples. We further **implement two ablations**: our IGC-LSTM and the biased transformer (BT). → As in **Figure 5**, our IGC-LSTM almost consistently outperforms the baselines, while the BT has large errors. Our *IGC-transformer remains the best for all prediction windows*. This shows that our novel approach for G-computation leads to accurate predictions, irrespective of the neural backbone. Further, it shows that proper adjustments are important for counterfactual outcome estimation.

### F.2 SENSITIVITY TO NOISE IN PSEUDO-OUTCOMES

Finally, we provide more insights into the quality of the generated pseudo-outcomes $\tilde{G}^{\bar{a}}_{t+\delta}$ in Figure 7. Here, we added increasing levels of constant bias to the pseudo-outcomes during training. Our results show that these artificial corruptions indeed lead to a significant decrease in the overall performance of our IGC-Net. We therefore conclude that, without artificial corruption, our generated pseudo-outcomes are good estimates of the true nested expectations. Further, this shows that correct estimates of the pseudo-outcomes are indeed necessary for high-quality, unbiased estimates. Of note, the quality of the predicted pseudo-outcomes is also directly validated by the strong empirical performance in Section 5.

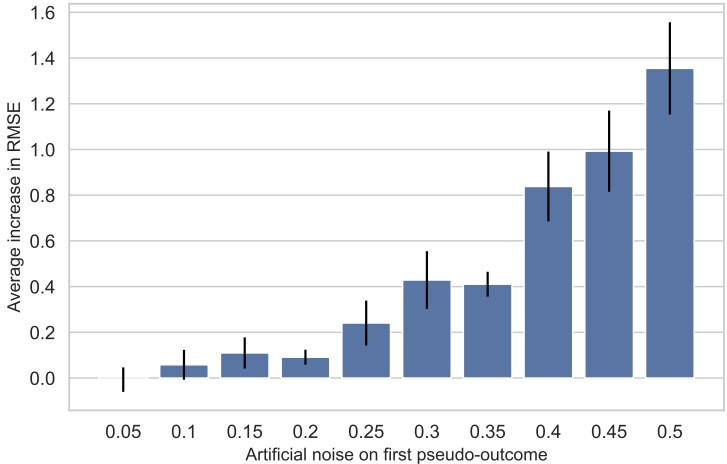

Figure 7: During training, we add **artificial levels of noise to the pseudo-outcomes** of our IGC-Net (prediction window $\tau = 2$, confounding strength $\gamma = 10$ on synthetic data). $\rightarrow$ We see that performance quickly deteriorates. This is expected, as it implies that the pseudo-outcomes generated by our IGC-Net are meaningful and important for accurate, unbiased predictions.

### F.3 UNCERTAINTY QUANTIFICATION

We can additionally equip our IGC-Net with uncertainty quantification, e.g., with dropout (Gal & Ghahramani, 2016), a standard and lightweight approach for predictive uncertainty estimation in neural networks. Because our model is fully differentiable and regression-based, MC dropout and similar approaches (Deng et al., 2025) can be applied without modifying the our IGC-Net, which makes uncertainty quantification straightforward to incorporate.

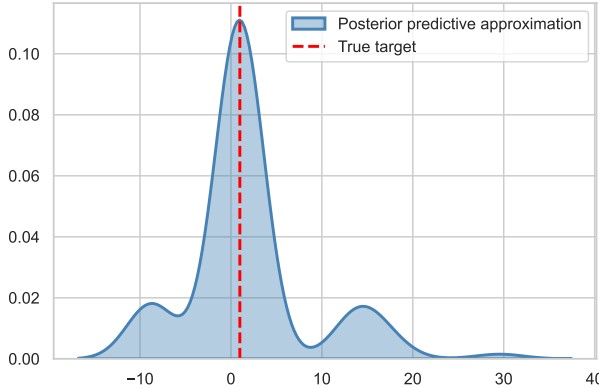

Figure 8: Our IGC-Net is easily compatible with *any* standard-procedure for uncertainty quantification. Reported is the posterior predictive using dropout and kernel smoothing.

### F.4 COEFFICIENT OF VARIATION

In the following, we additionally report the coefficient of variation of our main study on synthetic data in Section 5. Lower values in the coefficient of variation indicate more stable predictions. Table 8 shows the results. Clearly, our IGC-Net is superior to the baselines and has significantly more robust estimates of the CAPO.

| | $\gamma = 10$ | $\gamma = 11$ | $\gamma = 12$ | $\gamma = 13$ | $\gamma = 14$ | $\gamma = 15$ | $\gamma = 16$ | $\gamma = 17$ | $\gamma = 18$ | $\gamma = 19$ | $\gamma = 20$ |
|---|---|---|---|---|---|---|---|---|---|---|---|
| CRN (Bica et al., 2020) | 0.14 | 0.31 | 0.21 | 0.30 | 0.06 | 0.20 | 0.22 | 0.15 | 0.17 | 0.19 | 0.15 |
| TE-CDE (Seedat et al., 2022) | 0.13 | 0.10 | 0.03 | 0.10 | 0.09 | 0.14 | 0.10 | 0.09 | 0.12 | 0.09 | 0.10 |
| CT (Melnychuk et al., 2022) | 0.21 | 0.21 | 0.17 | 0.18 | 0.19 | 0.17 | 0.32 | 0.18 | 0.22 | 0.17 | 0.21 |
| RMSNs (Lim et al., 2018) | **0.06** | 0.05 | 0.07 | 0.07 | 0.07 | 0.09 | 0.12 | 0.13 | 0.10 | 0.12 | 0.11 |
| G-Net (Li et al., 2021) | 0.11 | 0.09 | 0.08 | 0.07 | 0.07 | **0.07** | 0.09 | 0.10 | 0.13 | 0.13 | 0.11 |
| **IGC-Net** (ours) | 0.07 | **0.04** | **0.06** | **0.04** | **0.03** | 0.09 | **0.07** | **0.07** | **0.09** | **0.06** | **0.07** |

Table 8: Coefficient of variation on synthetic data based on the tumor growth model with $\tau = 2$. Lower values indicate more stable predictions. Our IGC-Net clearly outperforms the baselines.

# G    DETAILS ON THE DATA GENERATING PROCESSES

## G.1    SYNTHETIC DATA

We use data based on the pharmacokinetic-pharmacodynamic tumor growth model (Geng et al., 2017), which is a standard dataset for benchmarking causal inference methods in the time-varying setting (Bica et al., 2020; Li et al., 2021; Lim et al., 2018; Melnychuk et al., 2022). Here, the outcome $Y_t$ is the volume of a tumor that evolves according to the stochastic process

$$Y_{t+1} = \left(1 + \underbrace{\rho \log\left(\frac{K}{Y_t}\right)}_{\text{Tumor growth}} - \underbrace{\alpha_c c_t}_{\text{Chemotherapy}} - \underbrace{(\alpha_r d_t + \beta_r d_t^2)}_{\text{Radiotherapy}} + \underbrace{\epsilon_t}_{\text{Noise}}\right) Y_t, \tag{74}$$

where $\alpha_c$, $\alpha_r$, and $\beta_r$ control the strength of chemo- and radiotherapy, respectively, and where $K$ corresponds to the carrying capacity, and where $\rho$ is the growth parameter. The radiation dosage $d_t$ and chemotherapy drug concentration $c_t$ are applied with probabilities

$$A_t^c, A_t^r \sim \text{Ber}\left(\sigma\left(\frac{\gamma}{D_{\max}}(\bar{D}_{15}(\bar{Y}_{t-1} - \bar{D}_{\max}/2)\right)\right), \tag{75}$$

where $D_{\max}$ is the maximum tumor volume, $\bar{D}_{15}$ the average tumor diameter of the last 15 time steps, and $\gamma$ controls the confounding strength. **We use the same parameterization as Melnychuk et al. (2022)**, and refer to their work for more details. For training, validation, and testing, we sample $N = 1000$ trajectories of lengths $T \leq 30$ each.

We are interested in the performance of our IGC-Net for increasing levels of confounding. We thus increase the confounding from $\gamma = 10$ to $\gamma = 20$. For each level of confounding, we fix an arbitrary intervention sequence and simulate the outcomes under this intervention for testing.

## G.2    SEMI-SYNTHETIC DATA

We build upon the MIMIC-extract (Wang et al., 2020), which is based on the MIMIC-III dataset (Johnson et al., 2016). Here, we use $d_x = 25$ different vital signs as time-varying covariates and as well as gender, ethnicity, and age as static covariates. Then, we simulate observational outcomes for training and validation, and interventional outcomes for testing, respectively. **Again, our data-generating process is taken from** (Melnychuk et al., 2022), which we refer to for more details. In summary, the data generation consists of three steps: (1) $d_y = 2$ untreated outcomes $\tilde{Y}_t^j$, $j = 1, 2$, are simulated according to

$$\tilde{Y}_t^j = \alpha_s^j \text{B-spline}(t) + \alpha_g^j g^j(t) + \alpha_f^j f_Y^j(X_t) + \epsilon_t, \tag{76}$$

where $\alpha_s^j$, $\alpha_g^j$ and $\alpha_f^j$ are weight parameters, B-spline$(t)$ is sampled from a mixture of three different cubic splines, and $f_Y^j(\cdot)$ is a random Fourier features approximation of a Gaussian process. (2) A total of $d_a = 3$ synthetic treatments $A_t^l$, $l = 1, 2, 3$, are applied with probability

$$A_t^l \sim \text{Ber}(p_t^l), \quad p_t^l = \sigma\left(\gamma_Y^l Y_{t-1}^{A,l} + \gamma_X^l f_Y^l(X_t) + b^l\right) \tag{77}$$

where $\gamma_Y^l$ and $\gamma_X^l$ are fixed parameters that control the confounding strength for treatment $A^l$, $Y_t^{A,l}$ is an averaged subset of the previous $l$ treated outcomes, $b^l$ is a bias term, and $f_Y^l(\cdot)$ is a random function that is sampled from an RFF (random Fourier features) approximation of a Gaussian process. (3) The treatments are applied to the untreated outcomes via

$$Y_t^j = \tilde{Y}_t^j + \sum_{i=t-\omega^l}^{t} \frac{\min_{l=1,\ldots,d_a} \mathbb{1}_{\{A_i^l=1\}} p_i^l \beta^{l,j}}{(\omega^l - i)^2}, \tag{78}$$

where $\omega^l$ is the effect window for treatment $A^l$ and $\beta^{l,j}$ controls the maximum effect of treatment $A^l$.

We run different experiments for training, testing, and validation sizes of $N = 1000$, $N = 2000$, and $N = 3000$, respectively, and set the time window to $30 \leq T \leq 50$. As the covariate space is high-dimensional, we thereby study how robust our IGC-Net is with respect to estimation variance. We further increase the prediction windows from $\tau = 2$ up to $\tau = 6$.

# H ARCHITECTURE OF ITERATIVE G-COMPUTATION NETWORK

In the following, we provide details on the architecture of our IGC-Net.

**Multi-input transformer:** The multi-input transformer as the backbone of our IGC-Net is motivated by (Melnychuk et al., 2022), which develops an architecture that is tailored for the types of data that are typically available in medical scenarios: (i) outcomes $\bar{Y}_t \in \mathbb{R}^{d_y \times t}$, covariates $\bar{X}_t \in \mathbb{R}^{d_x \times t}$, and treatments $\bar{A}_t \in \{0, 1\}^{d_a \times t}$. In particular, their proposed transformer model consists of three separate sub-transformers, where each sub-transformer performs *multi-headed self-attention mechanisms* on one particular data input. Further, these sub-transformers are connected with each other through *in-between cross-attention mechanisms*, ensuring that information is exchanged. Therefore, we build on this idea as the backbone of our IGC-Net, as we detail below.

Our multi-input transformer $z^\phi(\cdot)$ consists of three sub-transformer models $z^{\phi k}(\cdot)$, $k = 1, 2, 3$, where $z^{\phi k}(\cdot)$ focuses on one data input $\bar{U}_t^k \in \{\bar{Y}_t, \bar{X}_t, \bar{A}_{t-1}\}$, $k \in \{1, 2, 3\}$, respectively.

(1) Input transformations: First, the data $\bar{U}_t^k \in \mathbb{R}^{d_k \times t}$ is linearly transformed through

$$Z_t^{k,0} = (\bar{U}_t^k)^\top W^{k,0} + b^{k,0} \in \mathbb{R}^{t \times d_h} \tag{79}$$

where $W^{k,0} \in \mathbb{R}^{d_k \times d_h}$ and $b^{k,0} \in \mathbb{R}^{d_h}$ are the weight matrix and the bias, respectively, and $d_h$ is the number of transformer units.

(2) Transformer blocks: Next, we stack $j = 1, \ldots, J$ transformer blocks, where each transformer block $j$ receives the outputs $Z_t^{k,j-1}$ of the previous transformer block $j - 1$. For this, we combine (i) *multi-headed self- and cross-attentions*, and (ii) *feed-forward networks*.

(i) *Multi-headed self- and cross-attentions:* The output of block $j$ for sub-transformer $k$ is given by the *multi-headed cross-attention*

$$Z_t^{k,j} = \tilde{Q}_t^{k,j} + \sum_{l \neq k} \text{MHA}(\tilde{Q}_t^{k,j}, \tilde{K}_t^{l,j}, \tilde{V}_t^{l,j}), \tag{80}$$

where $\tilde{Q}_t^{k,j} = \tilde{K}_t^{k,j} = \tilde{V}_t^{k,j}$ are the outputs of the *multi-headed self-attentions*

$$\tilde{Q}_t^{k,j} = Z_t^{k,j-1} + \text{MHA}(Q_t^{k,j}, K_t^{k,j}, V_t^{k,j}). \tag{81}$$

Here, $\text{MHA}(\cdot)$ denotes the multi-headed attention mechanism as in (Vaswani et al., 2017) given by

$$\text{MHA}(q, k, v) = (\text{Attention}(q^1, k^1, v^1), \ldots, \text{Attention}(q^M, k^M, v^M)), \tag{82}$$

where

$$\text{Attention}(q^m, k^m, v^m) = \text{softmax}\left(\frac{q^m (k^m)^\top}{\sqrt{d_{qkv}}}\right) v^m \tag{83}$$

is the attention mechanism for $m = 1, \ldots, M$ attention heads. The queries, keys, and values $q^m, k^m, v^m \in \mathbb{R}^{t \times d_{qkv}}$ have dimension $d_{qkv}$, which is equal to the hidden size $d_h$ divided by the number of attention heads $M$, that is, $d_{qkv} = d_h/M$. For this, we compute the queries, keys, and values for the *cross-attentions* as

$$\tilde{Q}_t^{k,m,j} = \tilde{Q}_t^{k,j} \tilde{W}^{k,m,j} + \tilde{b}^{k,m,j} \in \mathbb{R}^{t \times d_{qkv}}, \tag{84}$$

$$\tilde{K}_t^{k,m,j} = \tilde{K}_t^{k,j} \tilde{W}^{k,m,j} + \tilde{b}^{k,m,j} \in \mathbb{R}^{t \times d_{qkv}}, \tag{85}$$

$$\tilde{V}_t^{k,m,j} = \tilde{V}_t^{k,j} \tilde{W}^{k,m,j} + \tilde{b}^{k,m,j} \in \mathbb{R}^{t \times d_{qkv}}, \tag{86}$$

and for the *self-attentions* as

$$Q_t^{k,m,j} = Z_t^{k,j-1} W^{k,m,j} + b^{k,m,j} \in \mathbb{R}^{t \times d_{qkv}}, \tag{87}$$

$$K_t^{k,m,j} = Z_t^{k,j-1} W^{k,m,j} + b^{k,m,j} \in \mathbb{R}^{t \times d_{qkv}}, \tag{88}$$

$$V_t^{k,m,j} = Z_t^{k,j-1} W^{k,m,j} + b^{k,m,j} \in \mathbb{R}^{t \times d_{qkv}}. \tag{89}$$

where $\tilde{W}^{k,m,j}, W^{k,m,j} \in \mathbb{R}^{d_h \times d_{qkv}}$ and $\tilde{b}^{k,m,j}, \tilde{b}^{k,m,j} \in \mathbb{R}^{d_{qkv}}$ are the trainable weights and biases for sub-transformers $k = 1, 2, 3$, transformer blocks $j = 1, \ldots, J$, and attention heads

$m = 1, \ldots, M$. Of note, each *self- and cross attention* uses relative positional encodings (Shaw et al., 2018) to preserve the order of the input sequence as in (Melnychuk et al., 2022).

(ii) *Feed-forward networks:* After the *multi-headed cross-attention* mechanism, our IGC-Net applies a feed-forward neural network on each $Z_t^{k,j}$, respectively. Further, we apply dropout and layer normalizations (Ba et al., 2016) as in (Melnychuk et al., 2022; Vaswani et al., 2017). That is, our IGC-Net transforms the output $Z_t^{k,j}$ for transformer block $j$ of sub-transformer $k$ through a sequence of transformations

$$\text{FF}^{k,j}(Z_t^{k,j}) = \text{LayerNorm} \circ \text{Dropout} \circ \text{Linear} \circ \text{Dropout} \circ \text{ReLU} \circ \text{Linear}(Z_t^{k,j}). \quad (90)$$

(3) Output transformation: Finally, after transformer block $J$, we apply a final transformation with dropout and average the outputs as

$$Z_t^{\bar{a}} = \text{ELU} \circ \text{Linear} \circ \text{Dropout}\left(\frac{1}{3}\sum_{k=1}^{3} Z_t^{k,J}\right), \quad (91)$$

such that $Z_t^{\bar{a}} \in \mathbb{R}^{d_z}$

**G-computation heads:** The *G-computation heads* $\{g_\delta^\phi(\cdot)\}_{\delta=0}^{\tau-1}$ receive the corresponding hidden state $Z_{t+\delta}^{\bar{A}}$ and the current treatment $A_{t+\delta}$ and transform it with another feed-forward network through

$$g_\delta^\phi(Z_{t+\delta}^{\bar{A}}, A_{t+\delta}) = \text{Linear} \circ \text{ELU} \circ \text{Linear}(Z_{t+\delta}^{\bar{A}}, A_{t+\delta}). \quad (92)$$

# I    IMPLEMENTATION DETAILS

In Supplements I.2 and I.3, we report details on the hyperparameter tuning. Here, we ensure that the total number of weights is comparable for each method and choose the grids accordingly. All methods are tuned on the validation datasets. As the validation sets only consist of *observational data* instead of interventional data, we tune all methods for $\tau = 1$-step ahead predictions as in (Melnychuk et al., 2022). All methods were optimized with Adam (Kingma & Ba, 2015). Further, we perform a random grid search as in (Melnychuk et al., 2022).

On average, training our IGC-Net on fully synthetic data took 13.7 minutes. Further, training on semi-synthetic data with $N = 1000/2000/3000$ samples took 1.1/2.1/3.0 hours. This is comparable to the baselines. All methods were trained on $1\times$ NVIDIA A100-PCIE-40GB. Overall, running our experiments took approximately 7 days (including hyperparameter tuning).

## I.1    COMPUTATIONAL COMPLEXITY

Let $N$ be the number of units, $t$ the observed window and $\tau$ the prediction horizon, $d$ the covariate dimension, $H$ the hidden size of the backbone, $L$ the number of backbone layers, and $K$ the number of Monte-Carlo samples used by G-Net (Li et al., 2021) and G-transformer (Xiong et al., 2024). We denote by $C_{\text{backbone}}(t, \tau)$ the cost of a single forward–backward pass of the sequence backbone over a prediction horizon $\tau$. For example, for an transformer or LSTM (for simplicity, assuming the same constant) this scales as

$$C_{\text{backbone}}(t, \tau) = O(\tau L(dH + H^2)), \tag{93}$$

(e.g., for a transformer it would include the usual $t$-dependent attention term). In both cases, the dependence on $\tau$ is contained inside $C_{\text{backbone}}(\tau)$.

Our method performs exactly one such backbone pass per unit, with a lightweight regression head on top. Thus, the total per-epoch cost is

$$C_{\text{IGC}} = O(N\,C_{\text{backbone}}(t, \tau)), \tag{94}$$

with no additional simulation or sampling loop.

G-Net and G-transformer share this backbone cost but **add** a Monte-Carlo simulation stage: for each unit and each time step they generate $K$ synthetic covariate updates using hold-out residuals. Each such update is $O(d)$; repeated for all time steps and all samples, this contributes

$$C_{\text{MC}} = O(N\tau Kd). \tag{95}$$

This term is additive because the Monte-Carlo simulation is an extra phase on top of the backbone training. Within this term, $K$ is **multiplicative** with $N$, $\tau$, and $d$: for every unit $N$ and every time step $\tau$, they perform $K$ residual draws, each touching all $d$ covariates. The total G-Net and G-transformer cost is therefore

$$C_{\text{G-Net/G-transformer}} = O(N\,C_{\text{backbone}}(t, \tau) + N\tau Kd), \tag{96}$$

which is strictly heavier than our regression-only IGC-Net whenever $K > 1$.

RMSNs have similar $O(\tau L(dH + H^2))$ order per network but must train both an outcome and a propensity model, effectively doubling the backbone term.

## I.2 HYPERPARAMETER TUNING: SYNTHETIC DATA

| Method | Component | Hyperparameter | Tuning range |
|---|---|---|---|
| CRN (Bica et al., 2020) | Encoder | LSTM layers ($J$) | 1 |
| | | Learning rate ($\eta$) | 0.01, 0.001, 0.0001 |
| | | Minibatch size | 64, 128, 256 |
| | | LSTM hidden units ($d_h$) | $0.5d_{yxa}, 1d_{yxa}, 2d_{yxa}, 3d_{yxa}, 4d_{yxa}$ |
| | | Balanced representation size ($d_z$) | $0.5d_{yxa}, 1d_{yxa}, 2d_{yxa}, 3d_{yxa}, 4d_{yxa}$ |
| | | FC hidden units ($n_{FC}$) | $0.5d_z, 1d_z, 2d_z, 3d_z, 4d_z$ |
| | | LSTM dropout rate ($p$) | 0.1, 0.2 |
| | | Number of epochs ($n_e$) | 50 |
| | Decoder | LSTM layers ($J$) | 1 |
| | | Learning rate ($\eta$) | 0.01, 0.001, 0.0001 |
| | | Minibatch size | 256, 512, 1024 |
| | | LSTM hidden units ($d_h$) | Balanced representation size of encoder |
| | | Balanced representation size ($d_z$) | $0.5d_{yxa}, 1d_{yxa}, 2d_{yxa}, 3d_{yxa}, 4d_{yxa}$ |
| | | FC hidden units ($n_{FF}$) | $0.5d_z, 1d_z, 2d_z, 3d_z, 4d_z$ |
| | | LSTM dropout rate ($p$) | 0.1, 0.2 |
| | | Number of epochs ($n_e$) | 50 |
| TE-CDE (Seedat et al., 2022) | Encoder | Neural CDE (Kidger et al., 2020) hidden layers ($J$) | 1 |
| | | Learning rate ($\eta$) | 0.01, 0.001, 0.0001 |
| | | Minibatch size | 64, 128, 256 |
| | | Neural CDE hidden units ($d_h$) | $0.5d_{yxa}, 1d_{yxa}, 2d_{yxa}, 3d_{yxa}, 4d_{yxa}$ |
| | | Balanced representation size ($d_z$) | $0.5d_{yxa}, 1d_{yxa}, 2d_{yxa}, 3d_{yxa}, 4d_{yxa}$ |
| | | Feed-forward hidden units ($n_{FF}$) | $0.5d_z, 1d_z, 2d_z, 3d_z, 4d_z$ |
| | | Neural CDE dropout rate ($p$) | 0.1, 0.2 |
| | | Number of epochs ($n_e$) | 50 |
| | Decoder | Neural CDE hidden layers ($J$) | 1 |
| | | Learning rate ($\eta$) | 0.01, 0.001, 0.0001 |
| | | Minibatch size | 256, 512, 1024 |
| | | Neural CDE hidden units ($d_h$) | Balanced representation size of encoder |
| | | Balanced representation size ($d_z$) | $0.5d_{yxa}, 1d_{yxa}, 2d_{yxa}, 3d_{yxa}, 4d_{yxa}$ |
| | | Feed-forward hidden units ($n_{FF}$) | $0.5d_z, 1d_z, 2d_z, 3d_z, 4d_z$ |
| | | Neural CDE dropout rate ($p$) | 0.1, 0.2 |
| | | Number of epochs ($n_e$) | 50 |
| CT (Melnychuk et al., 2022) | (end-to-end) | Transformer blocks ($J$) | 1,2 |
| | | Learning rate ($\eta$) | 0.01, 0.001, 0.0001 |
| | | Minibatch size | 64, 128, 256 |
| | | Attention heads ($n_h$) | 1 |
| | | Transformer units ($d_h$) | $1d_{yxa}, 2d_{yxa}, 3d_{yxa}, 4d_{yxa}$ |
| | | Balanced representation size ($d_z$) | $0.5d_{yxa}, 1d_{yxa}, 2d_{yxa}, 3d_{yxa}, 4d_{yxa}$ |
| | | Feed-forward hidden units ($n_{FF}$) | $0.5d_z, 1d_z, 2d_z, 3d_z, 4d_z$ |
| | | Sequential dropout rate ($p$) | 0.1, 0.2 |
| | | Max positional encoding ($l_{max}$) | 15 |
| | | Number of epochs ($n_e$) | 50 |
| RMSNs (Lim et al., 2018) | Propensity treatment network | LSTM layers ($J$) | 1 |
| | | Learning rate ($\eta$) | 0.01, 0.001, 0.0001 |
| | | Minibatch size | 64, 128, 256 |
| | | LSTM hidden units ($d_h$) | $0.5d_{yxa}, 1d_{yxa}, 2d_{yxa}, 3d_{yxa}, 4d_{yxa}$ |
| | | LSTM dropout rate ($p$) | 0.1, 0.2 |
| | | Max gradient norm | 0.5, 1.0, 2.0 |
| | | Number of epochs ($n_e$) | 50 |
| | Propensity history network / Encoder | LSTM layers ($J$) | 1 |
| | | Learning rate ($\eta$) | 0.01, 0.001, 0.0001 |
| | | Minibatch size | 64, 128, 256 |
| | | LSTM hidden units ($d_h$) | $0.5d_{yxa}, 1d_{yxa}, 2d_{yxa}, 3d_{yxa}, 4d_{yxa}$ |
| | | LSTM dropout rate ($p$) | 0.1, 0.2 |
| | | Max gradient norm | 0.5, 1.0, 2.0 |
| | | Number of epochs ($n_e$) | 50 |
| | Decoder | LSTM layers ($J$) | 1 |
| | | Learning rate ($\eta$) | 0.01, 0.001, 0.0001 |
| | | Minibatch size | 256, 512, 1024 |
| | | LSTM hidden units ($d_h$) | $1d_{yxa}, 2d_{yxa}, 4d_{yxa}, 8d_{yxa}, 16d_{yxa}$ |
| | | LSTM dropout rate ($p$) | 0.1, 0.2 |
| | | Max gradient norm | 0.5, 1.0, 2.0, 4.0 |
| | | Number of epochs ($n_e$) | 50 |
| G-Net (Li et al., 2021) | (end-to-end) | LSTM layers ($J$) | 1 |
| | | Learning rate ($\eta$) | 0.01, 0.001, 0.0001 |
| | | Minibatch size | 64, 128, 256 |
| | | LSTM hidden units ($d_h$) | $0.5d_{yxa}, 1d_{yxa}, 2d_{yxa}, 3d_{yxa}, 4d_{yxa}$ |
| | | LSTM output size ($d_z$) | $0.5d_{yxa}, 1d_{yxa}, 2d_{yxa}, 3d_{yxa}, 4d_{yxa}$ |
| | | Feed-forward hidden units ($n_{FF}$) | $0.5d_z, 1d_z, 2d_z, 3d_z, 4d_z$ |
| | | LSTM dropout rate ($p$) | 0.1, 0.2 |
| | | Number of epochs ($n_e$) | 50 |
| IGC-Net (ours) | (end-to-end) | Transformer blocks ($J$) | 1,2 |
| | | Learning rate ($\eta$) | 0.01, 0.001, 0.0001 |
| | | Minibatch size | 64, 128, 256 |
| | | Attention heads ($n_h$) | 1 |
| | | Transformer units ($d_h$) | $1d_{yxa}, 2d_{yxa}, 3d_{yxa}, 4d_{yxa}$ |
| | | Hidden representation size ($d_z$) | $0.5d_{yxa}, 1d_{yxa}, 2d_{yxa}, 3d_{yxa}, 4d_{yxa}$ |
| | | Feed-forward hidden units ($n_{FF}$) | $0.5d_z, 1d_z, 2d_z, 3d_z, 4d_z$ |
| | | Sequential dropout rate ($p$) | 0.1, 0.2 |
| | | Max positional encoding ($l_{max}$) | 15 |
| | | Number of epochs ($n_e$) | 50 |

Table 9: Hyperparameter tuning for all methods on fully synthetic tumor growth data. Here, $d_{yxa} = d_y + d_x + d_a$ is the overall input size. Further, $d_z$ denotes the hidden representation size of our IGC-Net, the balanced representation size of CRN (Bica et al., 2020), TE-CDE (Seedat et al., 2022) and CT (Melnychuk et al., 2022), and the LSTM (Hochreiter & Schmidhuber, 1997) output size of G-Net (Li et al., 2021). The hyperparameter grid follows (Melnychuk et al., 2022). Importantly, the tuning ranges for the different methods are comparable. Hence, the comparison of the methods in Section 5 is fair.

## I.3 HYPERPARAMETER TUNING: SEMI-SYNTHETIC DATA

| Method | Component | Hyperparameter | Tuning range |
|---|---|---|---|
| CRN (Bica et al., 2020) | Encoder | LSTM layers $(J)$ | 1,2 |
| | | Learning rate $(\eta)$ | 0.01, 0.001, 0.0001 |
| | | Minibatch size | 64, 128, 256 |
| | | LSTM hidden units $(d_h)$ | $0.5d_{yxa}, 1d_{yxa}, 2d_{yxa}$ |
| | | Balanced representation size $(d_z)$ | $0.5d_{yxa}, 1d_{yxa}, 2d_{yxa},$ |
| | | FF hidden units $(n_{FF})$ | $0.5d_z, 1d_z, 2d_z$ |
| | | LSTM dropout rate $(p)$ | 0.1, 0.2 |
| | | Number of epochs $(n_e)$ | 100 |
| | Decoder | LSTM layers $(J)$ | 1,2 |
| | | Learning rate $(\eta)$ | 0.01, 0.001, 0.0001 |
| | | Minibatch size | 256, 512, 1024 |
| | | LSTM hidden units $(d_h)$ | Balanced representation size of encoder |
| | | Balanced representation size $(d_z)$ | $0.5d_{yxa}, 1d_{yxa}, 2d_{yxa}$ |
| | | FC hidden units $(n_{FF})$ | $0.5d_z, 1d_z, 2d_z$ |
| | | LSTM dropout rate $(p)$ | 0.1, 0.2 |
| | | Number of epochs $(n_e)$ | 100 |
| TE-CDE (Seedat et al., 2022) | Encoder | Neural CDE hidden layers $(J)$ | 1 |
| | | Learning rate $(\eta)$ | 0.01, 0.001, 0.0001 |
| | | Minibatch size | 64, 128, 256 |
| | | LSTM hidden units $(d_h)$ | $0.5d_{yxa}, 1d_{yxa}, 2d_{yxa}$ |
| | | Balanced representation size $(d_z)$ | $0.5d_{yxa}, 1d_{yxa}, 2d_{yxa}$ |
| | | Feed-forward hidden units $(n_{FF})$ | $0.5d_z, 1d_z, 2d_z$ |
| | | Dropout rate $(p)$ | 0.1, 0.2 |
| | | Number of epochs $(n_e)$ | 100 |
| | Decoder | Neural CDE hidden layers $(J)$ | 1 |
| | | Learning rate $(\eta)$ | 0.01, 0.001, 0.0001 |
| | | Minibatch size | 256, 512, 1024 |
| | | LSTM hidden units $(d_h)$ | Balanced representation size of encoder |
| | | Balanced representation size $(d_z)$ | $0.5d_{yxa}, 1d_{yxa}, 2d_{yxa}$ |
| | | Feed-forward hidden units $(n_{FF})$ | $0.5d_z, 1d_z, 2d_z$ |
| | | LSTM dropout rate $(p)$ | 0.1, 0.2 |
| | | Number of epochs $(n_e)$ | 100 |
| CT (Melnychuk et al., 2022) | (end-to-end) | Transformer blocks $(J)$ | 1,2 |
| | | Learning rate $(\eta)$ | 0.01, 0.001, 0.0001 |
| | | Minibatch size | 32, 64 |
| | | Attention heads $(n_h)$ | 2,3 |
| | | Transformer units $(d_h)$ | $1d_{yxa}, 2d_{yxa}$ |
| | | Balanced representation size $(d_z)$ | $0.5d_{yxa}, 1d_{yxa}, 2d_{yxa}$ |
| | | Feed-forward hidden units $(n_{FF})$ | $0.5d_z, 1d_z, 2d_z$ |
| | | Sequential dropout rate $(p)$ | 0.1, 0.2 |
| | | Max positional encoding $(l_{max})$ | 30 |
| | | Number of epochs $(n_e)$ | 100 |
| RMSNs (Lim et al., 2018) | Propensity treatment network | LSTM layers $(J)$ | 1,2 |
| | | Learning rate $(\eta)$ | 0.01, 0.001, 0.0001 |
| | | Minibatch size | 64, 128, 256 |
| | | LSTM hidden units $(d_h)$ | $0.5d_{yxa}, 1d_{yxa}, 2d_{yxa}$ |
| | | LSTM dropout rate $(p)$ | 0.1, 0.2 |
| | | Max gradient norm | 0.5, 1.0, 2.0 |
| | | Number of epochs $(n_e)$ | 100 |
| | Propensity history network / Encoder | LSTM layers $(J)$ | 1 |
| | | Learning rate $(\eta)$ | 0.01, 0.001, 0.0001 |
| | | Minibatch size | 64, 128, 256 |
| | | LSTM hidden units $(d_h)$ | $0.5d_{yxa}, 1d_{yxa}, 2d_{yxa}$ |
| | | LSTM dropout rate $(p)$ | 0.1, 0.2 |
| | | Max gradient norm | 0.5, 1.0, 2.0 |
| | | Number of epochs $(n_e)$ | 100 |
| | Decoder | LSTM layers $(J)$ | 1 |
| | | Learning rate $(\eta)$ | 0.01, 0.001, 0.0001 |
| | | Minibatch size | 256, 512, 1024 |
| | | LSTM hidden units $(d_h)$ | $1d_{yxa}, 2d_{yxa}, 4d_{yxa}$ |
| | | LSTM dropout rate $(p)$ | 0.1, 0.2 |
| | | Max gradient norm | 0.5, 1.0, 2.0, 4.0 |
| | | Number of epochs $(n_e)$ | 100 |
| G-Net (Li et al., 2021) | (end-to-end) | LSTM layers $(J)$ | 1,2 |
| | | Learning rate $(\eta)$ | 0.01, 0.001, 0.0001 |
| | | Minibatch size | 64, 128, 256 |
| | | LSTM hidden units $(d_h)$ | $0.5d_{yxa}, 1d_{yxa}, 2d_{yxa}$ |
| | | LSTM output size $(d_z)$ | $0.5d_{yxa}, 1d_{yxa}, 2d_{yxa}$ |
| | | Feed-forward hidden units $(n_{FF})$ | $0.5d_z, 1d_z, 2d_z$ |
| | | LSTM dropout rate $(p)$ | 0.1, 0.2 |
| | | Number of epochs $(n_e)$ | 100 |
| IGC-Net (ours) | (end-to-end) | Transformer blocks $(J)$ | 1 |
| | | Learning rate $(\eta)$ | 0.001, 0.0001 |
| | | Minibatch size | 32, 64 |
| | | Attention heads $(n_h)$ | 2,3 |
| | | Transformer units $(d_h)$ | $1d_{yxa}, 2d_{yxa}$ |
| | | Balanced representation size $(d_z)$ | $0.5d_{yxa}, 1d_{yxa}, 2d_{yxa}$ |
| | | Feed-forward hidden units $(n_{FF})$ | $0.5d_z, 1d_z, 2d_z$ |
| | | Sequential dropout rate $(p)$ | 0.1, 0.2 |
| | | Max positional encoding $(l_{max})$ | 30 |
| | | Number of epochs $(n_e)$ | 100 |

Table 10: Hyperparameter tuning for all methods on semi-synthetic data. Here, $d_{yxa} = d_y + d_x + d_a$ is the overall input size. Further, $d_z$ is the hidden representation size of our IGC-Net, the balanced representation size of CRN (Bica et al., 2020), TE-CDE (Seedat et al., 2022) and CT (Melnychuk et al., 2022), and the LSTM (Hochreiter & Schmidhuber, 1997) output size of G-Net (Li et al., 2021). The hyperparameter grid follows (Melnychuk et al., 2022). Importantly, the tuning ranges for the different methods are comparable. Hence, the comparison of the methods in Section 5 is fair.

