# OpenReview forum: "IGC-Net for conditional average potential outcome estimation over time"
_ICLR.cc/2026/Conference — ICLR 2026 Poster_

### Official Review · Reviewer_Wupz · 2025-10-29

**Soundness:** 4
**Presentation:** 4
**Contribution:** 4
**Rating:** 8
**Confidence:** 5

**Summary:**

This paper proposes IGC-Net, a neural network approach for estimating conditional average potential outcomes (CAPO) over time in the presence of time-varying confounding. The key contribution is a fully regression-based iterative G-computation method that avoids the instabilities associated with inverse probability weighting when propensity scores are close to zero. The methodology uses a clever forward-generation approach with neural networks (LSTM or transformer) to approximate sequential conditional expectations through pseudo-outcomes. The paper includes both semi-synthetic experiments based on MIMIC-III data and real-world validation, demonstrating improved performance over existing methods, particularly under strong confounding.

**Strengths:**

## Strengths

1. **Excellent presentation and writing**: The introduction and related work sections are very well written, providing clear context and concrete problem motivation. The mathematical framing, particularly Equation 2, makes the problem formulation clear and accessible.

2. **Strong methodological foundation**: The sequential recursive formulation (Equations 6 and 7) is smart and theoretically sound, using conditional expectation projections from the remote future into the present. The approach is intuitive and properly grounded in causal inference principles.

3. **Effective problem motivation**: Table 1 does an excellent job showcasing the limitations of existing methods and positioning the contribution. The paper clearly articulates why inverse probability weighting methods fail under strong confounding.

4. **Comprehensive evaluation**: The paper includes both semi-synthetic and real-world MIMIC-III data (Supplement E), providing strong empirical validation. The results consistently demonstrate that the method works well, particularly when confounding strength is higher.

5. **Good ablation studies**: Testing both transformer and LSTM backbones provides useful insights into model architecture choices.

**Weaknesses:**

1. **Identifiability concern**: The main weakness is the lack of discussion about empirical identifiability of pseudo-observations across different random initializations (not the theoretical identifiability or SUTVA). If two IGC-Net instances with different initializations give the same g_{t+tau} but very different intermediate pseudo-observations at t+delta (delta < tau), the interpretation of conditional expectations becomes questionable. The paper addresses theoretical identifiability (SUTVA-like assumptions) but not the practical identifiability of neural network approximations.

2. **Limited treatment flexibility**: The setup only considers binary treatments, which may not align well with MIMIC-III data where emergency room interventions are typically not binary. The paper lacks discussion on extending to multi-valued or continuous treatments.

3. **Missing interpretability discussion**: Given the conservative nature of clinical applications, there should be discussion about interpretability trade-offs. Even when neural networks provide better accuracy, clinicians often prefer logistic regression for its transparency and explainability.

4. **Presentation issues**:
   - Figures 6 and 7 in supplement are too busy and difficult to read
   - Abstract should mention the use of synthetic MIMIC-III data
   - Should add a sentence explicitly stating that real-world data confirms the findings (currently only in supplement)

5. **Ablation analysis implication**: While the paper shows transformer performs slightly better than LSTM, there's insufficient discussion of why this is the case and what this tells us about the method. Is it because transformers are better at generative tasks while LSTMs are better at prediction?

6. **Unexplored training alternatives**: The paper doesn't discuss why intermediate-step training (using y_{t+delta} for delta < tau with actual outcomes rather than pseudo-outcomes) wasn't considered or wouldn't work.

## Minor Issues

- The observation about CT and TE-CDE (2022) lacking proper adjustments while older methods like RMSN (2018) and G-Net (2021) have them is interesting but not discussed
- No discussion on whether more powerful pre-trained generative models could improve results

**Questions:**

I have listed structured questions (with help of LLM) in the above weakness part. I am going to say here my honest thoughts when reading the paper as it presents, and hopefully this can help you understand how a new reader perceives your paper. These raw feelings are genuine and I hope they provide a more human-to-human communication and contexts for the structured question above.



# Review of Paper 7297: IGC-Net for Conditional Average Potential Outcome Estimation

## Initial Impressions on abstracts

This is a paper on IGC-Net for conditional average potential outcome estimation over time. The main problem addresses potential outcomes over time when confounding can be time-varying. The paper proposes a neural network for proper adjustment of time-varying confounding. I agree with the claim that not dividing by the propensity score creates limitations and instability when the propensity score is often close to 0. The estimation focuses on conditional average potential outcomes, which is fine.

The paper claims to be the first to perform fully regression-based iterative G-computation for CAPO in a time-variant setting. I think this section can be improved—I don't understand why performing the full regression iterative G-computation method is significant. The evaluation is based on experiments, and while we have experiment-based evaluation, I need to look at the simulation setting more carefully. Also, for ICLR conferences, it would be good to have real-world data. Since this is treatment over time, I imagine MIMIC-III data would be a good fit.

## Table 1 Analysis

I'm not following the literature too closely on neural network-based estimation, but Table 1 is actually very good at showcasing the problem. One thing I'm curious about: why do CT and TE-CDE, which are more recent, lack proper adjustments, whereas RMSN and G-Net in Table 1 are further away in the past but have proper adjustments? This is a very interesting observation. I have not read the literature review of those papers (CT, TE-CDE), so I don't know the full context. The CT paper and the TE-CDE paper are from 2022, the RMSN paper is from 2018, and the G-Net paper is from 2021.

## Introduction

I have general knowledge about causal inference. For the last paragraph, it would be good if you could explain a little bit about what G-computation is so that I don't have to Google it again. I just happen to be more familiar with the terminology from Don Rubin rather than James Robins.

The introduction is really good. It paints a clear picture of the field and its limitations—a very concrete problem it's trying to address. One thought: there is actually good reason for there not being many developments related to neural networks for causal inference. First, on clean tabular data, tree-based models have higher popularity. Second is interpretability. If you're using a clinician observational study dataset from EHR, even logistic regression is better understood—it's explainable and more transparent. Even when logistic regression performs worse (say, within 10% compared to neural networks), a clinician would tend to use logistic regression. It's just the conservative nature of humans—nothing we can do about that. I think it's still worthwhile to pursue this type of research using neural networks because accuracy is important. I'm just curious whether the authors have any discussion on this later. (Hey I came back from future, and the authors don’t have discussion on interpretability)

## Related Work

In the related work section, line 83 mentions a group of methods is not working, and I am convinced based on the narrative. I then flipped to the numerical example section and indeed saw that when the confounding strength is higher, the mentioned approach suffers higher bias. This is good validation. I also realized it's using MIMIC-III synthetic data, which answers my question from when I was reading the abstract. You should mention that in your abstract.

The related work is excellent—really well written. It tells me what's the problem with inverse probability weighting IPW and defers to a later proposition. But it's a known problem, so I would completely trust that.

## Setup and Methodology

In the setup, only binary treatment is considered. As I recall, MIMIC-III probably doesn't have binary treatment data for emergency room interventions, but it's okay for now. I'm curious if there's a discussion later on treatment that is more than just binary.

The setup and estimation tasks are all excellent. Line 126 on identifiability is very good—it's standard and very thoughtful to include those assumptions. Line 137 is correct about future potential confounding, and the explanation placed here is really good.

I really like Equation 2. Causal inference is all about mathematical framing, and Equation 2 makes the problem very clear. It would be good to explain how Equation 3 differs from Equation 2. After staring at it, I see it's basically adjusted by the probability of the “A” distribution. I understand that Supplement C has this formula. This looks intuitively right, so I wouldn't check it, but it would be helpful—I'm just being lazy—if you could still explain the intuition here.

This is a very good presentation. When I look at Equation 3, I would think a naive or straightforward approach is just through integration, and then it's pointed out that the integration has this distributional estimation problem. Very intriguing.

## Pseudo Outcomes and Methodology

In line 173, you use "low variance pseudo outcome." I'll need to read more, but one thing I was thinking when I read this was: in general, it is not great to look at the future Y variable when you're developing the causal method. I'm curious to see how you make that happen. In what sense is the outcome "pseudo"?

The iterative regression in line 180 is intuitively right. In causal inference, we use sequential, carefully constructed regression as an estimation vehicle. That is a very standard technique. Very nice opening to Section 4 at line 184.

Equations 6 and 7 are smart. They're basically regression, looking at the conditional expectation, gradually projected down from the remote future into the present. Because of the conditional expectation property, of course, it's unbiased. The sequential recursive formula is basically a sequential projection.

Line 222 is where I first became not crystal clear about how it is being done. How is your pseudo-data generated? Shouldn't it just be a natural projection down? Why does that generate predictions? So interesting. Rather than the true mathematical way of keep projecting from the future to present, the learning actually reverses that. Like data assimilation, you generate the future step, keep generating. And then after generating the tau steps, you produce the loss, matching this generated tau-step-ahead with the actual Y. And therefore, that's why you use neural networks as an approximator. Otherwise, before line 225, you don't really need neural networks yet. The math is tight for backward recursion, but computationally prohibitive. Line 225 introduces a neural network forward-generator to that true conditional expectation.

## Training Questions

That brings me to a question about this training. If you are generating t+tau steps ahead and then doing the regression, what if I generate just delta steps ahead (delta smaller than tau), and then do the regression of y_{t+delta} directly, not using the pseudo outcome but using the actual outcome to match the generated pseudo outcome? Is it because it actually has the realization, so it fails to be a proper projection from step y_{t+tau}? Is that why you didn't do that?

You are using LSTM or transformer. I would say a transformer or using pre-trained generators from existing generative AI pre-trained networks, it feels like if you have a more powerful generator, it would help. In your case, there is actually a very good link between generative AI and conditional expectation, which is very interesting. But I agree that this neural backbone can be swapped. I like that you tested at least two: transformer and LSTM.

## Results Section

I'm going to skip ahead to look at the results to see if transformer and LSTM actually give you a substantial difference. My impression when it comes to causal inference and this type of analysis is that transformers are actually not that helpful on top of LSTM. They should provide similar levels of results.

Your Figures 6 and 7 in supplement are too busy to read. Oh, the transformer is indeed giving you slightly better results than the LSTM. Is there a particular reason? Is it because transformers are just better at generative AI and LSTM is more for prediction? What does that ablation study tell you?

Figure 2 gives a very good schematic of how the thing is cascading. It's pretty intuitive. The training couples the generation step and the learning step. It generates and contrasts what's generated with the actual observation tau periods later, constructs a loss, and does this iteratively.

## Identifiability Concerns

Another question just popped into my mind: would there be an identifiability issue? That is, two trained networks will give you the same g_{t+tau} because they are labeled on y_tau. However, during the process from t to t+tau—you know, the t+delta part—the two networks could give you very different results. I know that earlier you have an identifiability section, but that is in line 126, more from a theoretical perspective of SUTVA-like assumptions. How will actual neural networks behave? We know that neural networks could achieve the same accuracy with two drastically different networks.

I'll skip ahead to Algorithm 1. I think I have a fairly good intuition now. Indeed, the Y is used as G_{t+tau}. All the rest are pseudo-Y, pseudo observations. So how do you make sure that your pseudo observation is actually identifiable?

Otherwise, I think the procedure is tight. Now skipping ahead to the experiment part, Section 5. By Section 4, I'm very convinced about what's going on, so I will accept that.

## Experimental Results

Skipping ahead to Section 5, which I previously briefly reviewed. The results are indeed good, and I have good intuition for why they are good. You have MIMIC-III data for semi-synthetic experiments, which is very good.

In Figure 3, my question is: why does Figure 3 show the LSTM is slightly worse than the transformer? It would be good to discuss why the transformer is better than the LSTM as a backbone.

Oh, you do have real-world data in Supplement E—the real-world MIMIC-III data. That's fairly strong. Let me look at the figures. You should add a sentence saying that the real-world data confirms the findings. Using real-world MIMIC-III data confirms the same findings.

## Overall Assessment

Right now, I think it's a very solid paper. The only question I have is on identifiability. If I initialize your IGC-Net with another random initialization and train another instance of your IGC-Net, will they give the same pseudo observations g_{t+tau}? Why or why not? If not, then the interpretation of conditional expectation becomes a little bit questionable. If it is the case, I'm kind of not seeing how the structure is preserved and why it's exactly identifiable from an intuitive perspective.

**Overall: Excellent paper, well-written, very enjoyable to read.**

---

> ### Author Response · Authors · 2025-11-25
>
> Thank you very much for your positive review of our paper! We improved our paper as a result of your comment, and highlighted all changes in **blue color** in our revised **rebuttal PDF**.
>
>
> ### Responses to Weaknesses:
>
> **>W1: Identifiability**
>
> Thank you. Our assumptions date back to works in epidemiology and classical statistics (Bang & Robins, 2005; Robins, 1986; 1994; 1999; Robins et al., 2000; Robins & Hernán, 2009; van der Laan & Gruber, 2012; van der Laan & Rose, 2018). The ML literature (Bica et al., 2020; Hess & Feuerriegel, 2025; Frauen et al., 2025; Li et al., 2021; Lim et al., 2018; Melnychuk et al., 2022; Seedat et al., 2022; Wang et al., 2025) has built on these frameworks, and they **all** rely on the same assumptions that we also use: consistency, sequential ignorability, and positivity. These are **not** special assumptions of our framework; they are the **standard identifying conditions across decades of longitudinal causal inference and all modern ML extensions**.
>
> Specifically, sequential ignorability is a relaxation of the standard identifying assumptions in the static setting. Here, one assumes that treatments are only administered once, which is a strong, oftentimes implausible assumption. Instead, treatments are usually administered over time, and hence our framework is more realistic. In short:
>
>
>
> * **Sequential Ignorability:** This assumption requires that the recorded covariate history captures the information driving treatment decisions; it is the standard identifying condition used in all longitudinal causal inference methods and is generally plausible in modern datasets with rich time-varying state information (e.g., datasets in oncology or intensive care units as ours typically capture all relevant time-varying information).
>
> * **Positivity:** This requires only that each treatment option occurs with nonzero probability along observed covariate histories. This condition is typically satisfied when the sample size is large enough.
>
> Upon reading your comment, we realized that we should test numerically how each method responds under violations of these two assumptions is crucial for implementation in real medical scenarios. Hence, **we performed two new experiments** (see **new results in Section 5**):
>
>
> * **Experiment I:** We introduce unobserved confounding in our synthetic experiments. **Our IGC-Net has stable performance** and, compared to baselines, does **not** deteriorate more dramatically than any other method (see **new Table 7**).
> * **Experiment II:** We add a parameter that controls the overlap in our experiments. Our results clearly show that our IGC-net is **highly stable** even when overlap is low (see **new Table 6**). This is supported theoretically: our pseudo-outcomes do **not** rely on inverse propensity weighting, which becomes very unstable when the propensities are extreme (see our **Proposition 3**, where we show that variance in our pseudo-outcomes is lower than for those constructed by IPW).
>
> => Evidently, our proposed ICG-Net performs best in both settings and clearly outperforms the baselines.
>
> **Action:** We add **two new experiments** with violations of positivity and sequential ignorability in our **revised Section 5**. Further, we add a discussion on the plausibility of our assumptions in our **new Supplement B**.
>
> **>W2: Treatment flexibility**
>
> Thank you. Our work is **not** limited to binary treatments; at any time-step, it can handle **arbitrary**, discrete treatments $A_t \in \mathcal{A} \subset \mathbb{N}$.
>
> **>W3: Logistic regression**
>
> While interpretability is important in clinical applications, the trade-off between transparency and predictive accuracy is well established. Linear or logistic regression models are inherently limited in their capacity: they cannot capture the nonlinear dynamics, temporal feedback, and multi-step treatment–covariate interactions that drive real clinical trajectories. For completeness, this was shown earlier in the CT paper, because of which all subsequent works no longer compared with logistic regressions due to their lack of predictive power (e.g., Seedat et al., 2022; Frauen et al., 2025).
>
> In contrast, neural architectures like IGC-Net provide the necessary flexibility to model these complexities and **therefore deliver substantially more accurate and clinically useful counterfactual predictions**. In high-stakes settings where decision quality depends on accurately forecasting patient responses to treatment sequences, this increase in modeling capacity is often indispensable, and logistic regressions fail to achieve this nonlinear modeling capacity. **Hence, our method overcomes the key limitation of logistic regression (namely, limited predictive ability) by a novel neural method with state-of-the-art performance.**
>
> **>W4: Presentation issues**
>
> Thank you for your suggestions! We will incorporate all of them into our final version of the paper.

---

> ### Author Response · Authors · 2025-11-25
>
> **>W5: Ablation study**
>
> The empirical gap between the transformer and LSTM variants is expected and aligns with the broader literature: transformers consistently outperform LSTMs on long-range sequence modeling due to their ability to capture complex, multi-scale temporal dependencies without the bottlenecks of recurrent architectures. Our backbone is a state-of-the-art multi-input transformer specifically optimized for heterogeneous clinical time series, which naturally yields stronger representations for the iterative g-computation recursion. Thus, the observed performance difference reflects the well-documented superiority of transformers for high-dimensional sequential tasks, rather than any generative–predictive distinction. Still, we emphasize that the LSTM ablation (called “IGC-LSTM”) outperforms all LSTM baselines. *This demonstrates the validity of our theoretical claims: regardless of the neural backbone, our iterative g-computation approach performs best.*
>
> **>W6: Training on factuals for intermediate outcomes**
>
> The iterative pseudo-outcome regression is a core component of the g-computation recursion and is **necessary** to obtain unbiased CAPO estimates under time-varying confounding. Replacing pseudo-outcomes with the **observed** intermediate outcomes would break this recursion: it would force the model to learn directly from the factual outcome trajectory, which reflects the **observed** treatment path rather than the interventional path $a_{t:t+\tau-1}$. This corresponds precisely to the strategy we used in our “biased transformer’’ ablation, where the model is trained only on factual outcomes and performs no adjustment; as shown in the experiments, this leads to substantial bias and much worse performance. For this reason, intermediate-step training on observed outcomes is not suitable for estimating counterfactual trajectories, and pseudo-outcomes are required to correctly propagate the g-formula under arbitrary intervention sequences. **=> Hence, our ablation studies already demonstrate numerically that training on factuals for intermediate outcomes is inferior.**
>
> **>W7: Biased baselines**
>
> Thank you. As discussed in our Related Work section, CT and TE-CDE do **not** implement proper adjustments required for valid estimation under time-varying confounding. Hence, *even with infinite data, these models will not recover the true CAPO and will lead to bias and thus unreliable predictions.*  This is expected as they rely on representation balancing heuristics, which can reduce estimation variance but are not designed to, and in general cannot, address the recursive bias introduced by evolving confounders. In contrast, older methods such as RMSNs and G-Net do contain explicit adjustment mechanisms (IPW and sampling-based G-computation, respectively), though both suffer from well-known instability and variance issues. In contrast, our method provides a stable and computationally efficient implementation of iterative G-computation with proper adjustment at every time step.
>
> **>W8: Generative models**
>
> Our goal is to obtain accurate causal point estimates of CAPOs, **not** to learn a full generative model of the covariate-treatment-outcome process. Pre-trained generative models are designed to maximize likelihood of observed data and approximate high-dimensional joint distributions, which is neither necessary nor sufficient for identifying CAPOs under time-varying confounding. In fact, introducing a large generative backbone would mainly add variance and computational overhead, while doing nothing to address the core identification and adjustment problem that our iterative G-computation algorithm is built to solve. For this reason, simply “plugging in” a more powerful generative model is not a principled or meaningful way to improve the type of estimand we target.
>
> **Action:** We added **new experiments** to demonstrate our ICG-Net with uncertainty quantification in our **revised Section 5**.

---

> ### Author Response · Authors · 2025-11-25
>
> ### Responses to Questions:
>
> **> Overall assessment:**
>
> Thank you. The CAPO is uniquely identified under the standard longitudinal settings(consistency, sequential ignorability, and overlap), and our iterative G-computation recursion targets this quantity directly. Different random initializations of IGC-Net will, as with any neural network trained via stochastic gradient descent, converge to slightly different parameterizations; consequently, the learned pseudo-outcomes may exhibit small variability across seeds. *This behavior is entirely expected and reflects optimization stochasticity, **not** non-identifiability* of the estimand. In all experiments, these differences are minor, and the predictions converge toward the same ground-truth CAPO values, just as multiple independently trained regressors converge to the same conditional expectation in standard supervised learning. Thus, the conditional expectation remains well-defined and identifiable, even though its finite-sample neural approximations may differ slightly across random initializations.
>
> **Action:** We add **two new experiments** with violations of positivity and sequential ignorability in our **revised Section 5**. Further, we add a discussion on the plausibility of our assumptions in our **new Supplement B**.
>
> ____
>
> Ioana Bica, Ahmed M. Alaa, James Jordon, and Mihaela van der Schaar. Estimating counterfactual treatment outcomes over time through adversarially balanced representations. In ICLR, 2020.
>
> Dennis Frauen, Konstantin Hess, and Stefan Feuerriegel. Model-agnostic meta-learners for estimating heterogeneous treatment effects over time. In ICLR, 2025.
>
> Bryan Lim, Ahmed M. Alaa, and Mihaela van der Schaar. Forecasting treatment responses over time using recurrent marginal structural networks. In NeurIPS, 2018.
>
> Valentyn Melnychuk, Dennis Frauen, and Stefan Feuerriegel. Causal transformer for estimating counterfactual outcomes. In ICML, 2022.
>
> James M. Robins. A new approach to causal inference in mortality studies with a sustained exposure period: Application to control of the healthy worker survivor effect. Mathematical Modelling, 7:1393–1512, 1986.
>
> James M. Robins. Correcting for non-compliance in randomized trials using structural nested mean models. Communications in Statistics - Theory and Methods, 23(8):2379–2412, 1994.
>
> James M. Robins. Robust estimation in sequentially ignorable missing data and causal inference models. Proceedings of the American Statistical Association on Bayesian Statistical Science, pp. 6–10, 1999.
>
> James M. Robins and Miguel A. Hernan. Estimation of the causal effects of time-varying exposures. Chapman & Hall/CRC handbooks of modern statistical methods. CRC Press, Boca Raton, 2009. ISBN 9781584886587.
>
> James M. Robins, Miguel A. Hernan, and Babette Brumback. Marginal structural models and causal inference in epidemiology. Epidemiology, 11(5):550–560, 2000.
>
> Nabeel Seedat, Fergus Imrie, Alexis Bellot, Zhaozhi Qian, and Mihaela van der Schaar. Continuous-time modeling of counterfactual outcomes using neural controlled differential equations. In ICML, 2022.
>
> Mark J. van der Laan and Susan Gruber. Targeted minimum loss based estimation of causal effects of multiple time point interventions. The International Journal of Biostatistics, 8(1), 2012.
>
> Mark J. van der Laan and Sherri Rose. Targeted learning in data science. Springer, Cham, 2018. ISBN 978-3-319-65303-7.

---

### Official Review · Reviewer_Kwm7 · 2025-10-31

**Soundness:** 3
**Presentation:** 2
**Contribution:** 3
**Rating:** 6
**Confidence:** 3

**Summary:**

The authors propose a new methods for estimating potential outcomes over time for treatment sequences. The authors argue that existing methods do not properly adjust for time-varying confounding and point out that existing methods that do control for this type of confounding have some limitations. To this end, they propose a new neural network method that leverages G-computation. The authors show both theoretically and empirically that their method effectively controls for time-varying confounding.

**Strengths:**

- Good theoretical background and motivation as to why a new method is needed.

- The authors make clear and significant contributions.

- Extensive and strong experimental results that show that IGC-Net achieves state-of-the-art performance for the datasets used.

**Weaknesses:**

At some points, the notation is confusing to me, making it hard to understand all the different parts of the paper. Below, I give some examples.

- $g$ can have two different meanings if I understand it correctly. $g^a_{t+\delta}$ is used in the theory for iterative G-computation, while  $g^\delta_{\phi}$ is a block of the neural network architecture that uses different sub- and superscripts. It would be helpful to the reader if the notation were clarified in some way.

- In Figure 1, it is not obvious to me how I should interpret $a$ and $A$ during training and inference. Is $A_t$ the treatment that is actually observed at time $t$, and $a_t$ the treatment for which you want to estimate the CAPO? Does this mean that during training, these are the same?

- I find the notation $a= a_{t:t+\tau-1}$ confusing, is this standard in the literature? If not, I would suggest a different notation that makes it clearer that $a$ is the whole treatment sequence.

If I understand correctly, you always need to decide on the forecasting horizon $\tau$ beforehand. Does this mean that if you wanted to know the effect of a treatment sequence that is shorter than your horizon (e.g., $\tau-2$}), you would have to train a whole new model? If so, how does this compare to the baselines?

How do the contributions of IGC-Net compare to [1]? I think the learners introduced in that paper should be discussed in the related work section.


[1] Dennis Frauen, Konstantin Hess, and Stefan Feuerriegel. Model-agnostic meta-learners for estimating heterogeneous treatment effects over time. In International Conference on Representation Learning, 2025.

**Questions:**

In addition to the question in the Weaknesses section, I have some other questions:

- In my understanding, $g^{0}_\phi$ is the only component used at inference. However, I fail to see how it can make predictions for an arbitrary treatment sequence. From Figure 1, it appears that only $a_t$ is used as input. Is this correct? If so, could you explain the intuition as to why it should not take the whole treatment sequence as input?

- In Figure 3, is the biased transformer the same as Causal Tranformer (CT) used in the previous experiments?

- Is it correct that for each level of confounding $\gamma$, the treatment sequence used for evaluation is always the same for all samples? If so, could you comment on whether sampling multiple different treatment sequences and calculating your error metric over these different sequences as well would be a better or worse way to evaluate the different methods?

---

> ### Author Response · Authors · 2025-11-25
>
> Thank you very much for your positive review of our paper! We improved our paper as a result of your comment, and highlighted all changes in **blue color** in our revised **rebuttal PDF**.
>
>
> ### Responses to Questions:
>
> **>Q1: Meaning of $g$**
>
> Thank you. This is correct, we use $g$ both for the theory in our iterative G-computation **and** our G-computation heads. This is deliberate, as the G-computation heads $g_\phi^\delta$ are used as the components that perform the recursion, and learn the different levels $g_{t+\delta}^a$ in the nested expectations.
>
> Here, we use $\phi$ to highlight that the G-computation heads depend on trainable parameters $\phi$.
>
> Of note, the heads $g_\phi^\delta$ learn the recursion for any $t$, and are therefore independent of the specific length of the trajectory. Paths with different lengths are transformed within the neural backbone (e.g., transformer or LSTM backbone).
>
> **Action:** To clarify notation, we switched from $g_\phi^\delta$ to $g_\delta^\phi$ in our revised paper version such that time-components are consistently in subscripts.
>
> **>Q2: Figure 1**
>
> In our notation, $A_t$ denotes the **observed** treatment in the data at time $t$, whereas $a_t$ denotes the **interventional** treatment whose CAPO we wish to estimate. During training, we use the observed trajectory treatments together with the observed covariates and outcomes to learn the components of the iterative G-computation recursion.
>
> That is, we learn/regress on $A_{t+ \delta}$ in the **learning step**, and then predict the pseudo-outcomes for $A_{t+ \delta}=a_{t+ \delta}$ in the **generation step**. Then, the next G-computation head takes the predicted pseudo outcome and regresses it on $A_{t+\delta-1}$, and so on. Thereby, we ensure that our algorithmic approach to G-computation exactly mirrors the recursive G-computation from the theory (**Proposition 2**).
>
> **>W3: Notation regarding $a$**
>
> Thank you. We changed our notation analogously to [1], and now use $\bar{a}=a_{t:t+\tau-1}$ for the entire treatment sequence.
>
> **Action:** We revised our notation throughout our paper.
>
> **>W4: Deciding on the forecasting horizon**
>
> Thank you. This is correct: in our model, we need to decide on a forecast horizon before training, which is, admittedly, a slight disadvantage of our method. Still, this is common for the for works in causal ML over time (e.g., Frauen et al., 2025)
>
> **>W5: Comparison to [1]:**
>
> Thank you for highlighting this work. Their work generally focuses on *identification* strategies, **not** model-based estimators. Therefore, their learners are **not** direct baselines but rather a theoretical presentation of what adjustment strategies generally exist.
>
> Our IGC-Net uses a similar identification regression-adjustment (RA) approach in their paper, in the sense that both rely on the G-formula and estimate the conditional mean of the next outcome given the observed history. However, the core innovation of IGC-Net lies **not** in adopting RA as an identification strategy, but in developing a **novel end-to-end learning algorithm that implements the full multi-step G-computation recursion within a single neural architecture**. Instead of fitting a separate model at each time step, our method couples representation learning with iterative pseudo-outcome learning+generation, which enables joint optimization across all time-steps.
>
> Our approach has **several unique advantages**: it (1) eliminates the variance explosion of IPW-based methods, (2) avoids the high-dimensional distribution approximation used in G-Net–style Monte Carlo G-computation, and (3) yields a stable, low-variance estimator for sequential CAPOs over long horizons. In this sense, while our identification step aligns with RA, the algorithmic realization and practical performance gains are unique to our proposed IGC-Net framework.
>
> **Action:** We highlight [1] in our **revised extended related work in Supplement A**.

---

> ### Author Response · Authors · 2025-11-25
>
> ### Responses to Questions:
>
> **>Q1: Inference**
>
> Thank you. Your understanding is correct: At inference time, only $g_\phi^0$ (and the neural backbone) are used. This is because during training, the IGC-Net is optimized to accurately predict a specific treatment sequence $\bar{a}=a_{t:t+\tau-1}$. Hence, the G-computation head $g_\phi^0$ is trained on generated pseudo-outcomes that exactly mirror the regressor in the outermost expectation of the nested G-computation formula.
>
> **>Q2: Figure 3**
>
> The biased transformer ablation and the causal transformer are similar, but have some important differences: the biased transformer has several read-out heads $g_\phi^\delta$ for different time-horizons, whereas the causal transformer only has a single read-out component that is used for autoregressive predictions. Further, the causal transformer uses a balancing loss as a heuristic to mitigate confounding bias. Balancing, however, is ultimately designed to reduce estimation variance, not to mitigate bias or to adjust for time-varying confounding [2].
>
> As such, both methods suffer from the same problem: they are **biased** and do **not** target the correct estimand. In contrast, our IGC-Net performs G-computation through iterative regressions in an end-to-end learning algorithm.
>
> In Figure 3, we further see that our novel training algorithm works **independently** of the neural backbone. Our IGC-LSTM ablation has very strong performance. We compare both ablations to all other baselines in our **Section 5 and Supplement F**, where the IGC-LSTM is the best-performing alternative to our IGC-Net.
>
> **>Q3: Treatment sequence**
>
> Thank you. We arbitrarily picked the treatment sequence for each run (which is consistent with, e.g., [1,3]). Generally, there is no preference which treatment sequence we evaluate the methods on; for each seed, all methods are evaluated on the exact same treatment intervention and, hence, the same ground-truth counterfactual.
>
> ____
>
> [1] Dennis Frauen, Konstantin Hess, and Stefan Feuerriegel. Model-agnostic meta-learners for estimating heterogeneous treatment effects over time. In International Conference on Representation Learning, 2025.
>
> [2] Fredrik D. Johansson, Uri Shalit, and David Sonntag. Learning representations for counterfactual inference. In ICML, 2016.
>
> [3] Valentyn Melnychuk, Dennis Frauen, and Stefan Feuerriegel. Causal transformer for estimating counterfactual outcomes. In ICML, 2022.

---

> > ### Comment · Reviewer_Kwm7 · 2025-11-27
> >
> > Thank you for clarifying the notation and answering my questions! This has significantly improved my understanding of your method.
> >
> > I have one final question regarding the fact that you have to retrain your model for each different counterfactual treatment sequence. I wonder how much time it takes (empirically) to train your model for a single sequence. Does it depend on the forecasting horizon? For example, if a doctor wanted to know for a single patient what would happen given, e.g., 5 different potential treatment plans (so they could pick the best one), would you retrain your model for each different sequence? Is this feasible? Would it be feasible to pre-train models for a lot of different treatment plans/forecasting horizons? I don't expect a full experiment; some rough estimates and intuition suffice.
> >
> > Thank you again for answering my (and the other reviewers') questions!

---

> > > ### Author Response · Authors · 2025-11-27
> > >
> > > Thank you for your comment!
> > >
> > > This is correct, for different treatment sequences, our IGC-net needs to be retrained. While this comes at the cost of computing time, we gain a **significant increase in accuracy for estimating CAPOs**.
> > >
> > > Further, our method is comparably fast at (re-)training, and there are several disadvantages for other methods regarding computing time too (see our **new Supplement I.1** for a discussion on computational complexity):
> > >
> > > * our IGC-Net is trained **end-to-end** in a single model, whereas models like TE-CDE, CRN, RMSNs require training several models
> > > * our IGC-Net does **not** rely on computationally expensive residual hold-out sampling (different to G-Net and G-transformer), which yields strong computational advantages during inference.
> > >
> > > There is, admittedly, a trade-off; if a practitioner is interested in a very large number of possible treatment paths, the computational cost of our IGC-Net will, at some point, be larger than for the baselines. For any sensible number of treatment paths, however, computing time is comparable to all the baselines (for example, training our IGC-Net on synthetic data took on average about 13.7 minutes on 1× NVIDIA A100-PCIE-40GB).
> > >
> > > Regarding your last question, we assume that re-training an already trained model on a new treatment plan should speed up computing time. Here, the neural backbone (the transformer oder LSTM) processes the history, and requires the largest number of trainable weights in the architecture. Hence, different forecasting horizons should **not** impact the weights of the backbone too much, and we therefore expect that re-using an already trained backbone will speed-up computing time. The G-computation heads are the components primarily responsible for performing the actual G-computation, and their weights will therefore vastly differ from sequence to sequence. Since they are simple MLPs, re-training them is very fast.

---

> > > > ### Comment · Reviewer_Kwm7 · 2025-11-27
> > > >
> > > > Thank you for your reply! I think the proposed re-training approach is very interesting and would make sense in practice.
> > > >
> > > > You have addressed my concerns, and I will therefore raise my score to recommend acceptance.

---

> > > > > ### Author Response · Authors · 2025-11-27
> > > > >
> > > > > Thank you very much for your helpful and positive review! We will incorporate all of your suggestions into the final version of our paper.

---

### Official Review · Reviewer_drGZ · 2025-10-31

**Soundness:** 2
**Presentation:** 3
**Contribution:** 2
**Rating:** 2
**Confidence:** 5

**Summary:**

The paper introduces IGC-Net (Iterative G-computation Network), a neural framework for estimating conditional average potential outcomes (CAPOs) over time from observational data. The method addresses the challenge of time-varying confounding by leveraging iterative regression-based G-computation, avoiding limitations of existing methods like inverse propensity weighting. The main contribution of the paper is around avoiding high-dimensional probability distribution estimation that were used in SOTA neural network based G-computation methods. From the presented results, IGC-Net demonstrates superior performance in experiments across synthetic, semi-synthetic, and real-world data compared to baseline methods.

**Strengths:**

Some of the key strengths of the paper are as follows:
- While G-computation has already been adapted to deep learning models in literature, the authors aim to circumvent the problem of full distribution estimation, by proposing an iterative regression-based approach. Using this idea of iterated expectations the authors are able to avoid costly monte carlo sampling and problems of high-dimensional overfitting that arises in presented literature
- Commendably, the authors provide a comprehensive evaluation across synthetic, semi-synthetic, and real-world data, showing consistent outperformance of baseline methods.
- Finally, the authors have provided a detailed architecture and hyperparameter tuning, ensuring methodological rigor and reproducibility. The presentation is easy to follow and covers the most important aspects of the paper well

**Weaknesses:**

While the paper has made a commendable efforts, there are several key issues with the paper
- First, the authors haven't considered some of the latest literature on this topic. For example, G-transformer [1] was already proposed by the authors of G-net as a signficant improvement by adapting the backbone from RNN to a transformer model. A similar effort has also been reported in [2]. Another paper that aims to capture the uncertainty in G-computation has been proposed in [3]
- Second, while the authors have proposed a neat computational approach to estimating the dynamic treatment effect, it is not evident whether the estimation process is able to capture the predictive uncertainty as well as a full scale MCMC methods


[1]: Xiong H, Wu F, Deng L, Su M, Shahn Z, Lehman LH. G-Transformer: Counterfactual Outcome Prediction under Dynamic and Time-varying Treatment Regimes. Proc Mach Learn Res. 2024 Aug;252:https://proceedings.mlr.press/v252/xiong24a.html. PMID: 40433313; PMCID: PMC12113242.
[2]: Hess, Konstantin, et al. "G-transformer for conditional average potential outcome estimation over time." arXiv preprint arXiv:2405.21012 (2024)
[3]: Deng L, Xiong H, Wu F, Kapoor S, Ghosh S, Shahn Z, Lehman LH. Uncertainty Quantification for Conditional Treatment Effect Estimation under Dynamic Treatment Regimes. Proc Mach Learn Res. 2024 Dec;259:248-266. PMID: 40443560; PMCID: PMC12121963.

**Questions:**

Some of the key questions for the authors are as follows:
- How does the method scale with larger prediction windows (τ) while maintaining performance? Have the authors considered trade-offs in performance with respect to traditional G-computational methods as the prediction window changes?
- Have the authors investigated the effect of availability of large scale datasets for G-computational methods? For example, is there an inflection point such that availability of data would allow the model to learn the join distribution instead of the first moments

---

> ### Author Response · Authors · 2025-11-25
>
> Thank you very much for your helpful review of our paper! We improved our paper as a result of your comment, and highlighted all changes in **blue color** in our revised **rebuttal PDF**.
>
>
> ### Responses to Weaknesses:
>
> **>W1: Literature / G-transformer**
>
> Thank you!
>
> We **implemented the G-Transformer by [1] as a new state-of-the-art baseline**. We find that our proposed ICG-Net is superior by a clear margin. As shown in our **revised Tables 3+4**, **our method clearly outperforms this baseline**. This performance difference can be explained by the methodological structure of the G-Transformer. The G-Transformer by [1] follows the same strategy as G-Net by relying on hold-out residual sampling to approximate future covariate trajectories. Consequently, the G-Transformer inherits the same **inefficiencies** associated with repeatedly sampling high-dimensional residuals, which serve only as a nuisance mechanism to approximate conditional means. This approach has substantial computational overhead, and increased variance (see our **revised Table 2**), which makes it a less stable and less data-efficient way to obtain estimates of CAPO compared to our direct iterative regression framework. In sum, *our direct iterative regression framework avoids this bottleneck*, **leading to more stable, data-efficient, and accurate CAPO estimates.**
>
> **Action:** We **added the G-transformer by Xiong et al. (2024) to all of our experiments** (see **new results in our revised Section 5**).
>
> **Uncertainty Quantification for Conditional Treatment Effect Estimation under Dynamic Treatment Regimes [2]:**
>
> Upon reading your comment, we realized that we should study uncertainty quantification, and we thus added a **new experiment in our revised Section 5**. As you will see, our ICG-Net method can be easily extended to incorporate uncertainty quantification.
>
> Further, the authors of [2] do **not** introduce a new estimator for time-varying conditional treatment effects / CAPOs; instead, they take **existing** G-Net and G-Transformer models as fixed bases and **add** approximate Bayesian uncertainty quantification layers (deep ensembles, variational dropout, SWAG) on top. Their focus is on improving calibration and decision-making by modeling parameter uncertainty, not on changing the underlying CAPO estimand or addressing bias/variance trade-offs of the point estimates themselves.
>
> As such, their contribution is **orthogonal to ours** and could in principle be applied on top of IGC-Net as well: treating this as a “baseline” would mainly compare different uncertainty wrappers around similar g-computation backbones rather than address the core problem of CAPO estimation under time-varying confounding. For this reason, we regard [2] as complementary work on uncertainty quantification, not a directly comparable baseline for our primary task.
>
> **Action:** We added [2] to our extended related work in our **revised Supplement A**, and incorporated **new experiments** on uncertainty quantification in a new experiment in our **revised Section 5** (please see **W2**).
>
> **>W2: Uncertainty quantification**
>
> Thank you. We agree that uncertainty quantification is important for medical applications, and we have followed the recommendations of [3] by adding dropout-based uncertainty estimation to IGC-Net in a new experiment, which provides a straightforward and computationally efficient way to capture predictive uncertainty.
>
> We also note that **neither G-Net nor the G-Transformer employs full-scale MCMC**: both approximate high-dimensional counterfactual distributions using **hold-out residual sampling**, which is not only far from MCMC-style posterior sampling but also **inefficient** with respect to sample usage and computational cost.
>
> *Our IGC-Net can seamlessly incorporate standard uncertainty quantification techniques such as dropout, deep ensembles, or SWAG on top of its deterministic backbone, without changes to the underlying recursive algorithm*. Thus, there are **no** methodological barriers to equipping our approach with uncertainty estimation comparable to the alternatives.
>
> **Action:** We added a **new experiment** to demonstrate our IGC-Net is easily compatible with uncertainty quantification in our **revised Section 5**.

---

> ### Author Response · Authors · 2025-11-25
>
> ### Responses to Questions:
>
> **>Q1: Prediction window**
>
> Thank you. We have extensively evaluated the behavior of IGC-Net across increasing prediction horizons. In the main paper, we report results for windows up to $\tau = 6$, and in our **Supplement F**, we **now further extend this to $\tau = 12$**. Across all settings, **our IGC-Net remains highly stable** as $\tau$ grows, and consistently outperforms all G-computation-based baselines (G-Net, G-transformer by [1]) as well as the other methods.
>
> This robustness follows from our low-variance pseudo-outcome recursion, which **avoids** sample inefficiency due to hold-out residuals and high-dimensional distribution estimation, variance explosion as in IPW-based baselines, and history-adjustment bias. Empirically, IGC-Net degrades more slowly than existing methods as $\tau$ increases, demonstrating **strong scalability to longer prediction windows**.
>
> Action: We added a new experiment with longer prediction windows to **Supplement F.**
>
> **>Q2: Distribution estimation**
>
> Thank you. Our goal is **not** to learn the full joint distribution of future covariates and outcomes, but to obtain accurate **point estimates** of the conditional average potential outcomes (CAPOs), optionally augmented with uncertainty quantification. This is the same objective pursued by all existing baselines we compare against (RMSNs, CRN, CT, G-Net, etc.).
>
> G-Net does not learn the full joint distribution either; its use of hold-out residual sampling is merely an auxiliary way to approximate the conditional **mean** required by the g-formula. The residual-based trajectory generation is thus a nuisance component rather than a principled attempt at density estimation, and it is an **inefficient way to obtain a point estimate**, as it requires repeated Monte-Carlo rollouts with high computational cost and variance. By contrast, our iterative regression implementation directly targets the relevant conditional means without relying on high-dimensional distribution approximations, leading to substantially more stable and data-efficient CAPO estimation.
>
> Thus, our design intentionally focuses only on the components of our recursive G-computation that matter for reliable **point estimation**, which is consistent with the literature we contribute to.
>
> **Action:** We nevertheless added **new experiments** to demonstrate our ICG-Net with uncertainty quantification in our **revised Section 5**.
>
> ____
>
> [1]: Xiong H, Wu F, Deng L, Su M, Shahn Z, Lehman LH. G-Transformer: Counterfactual Outcome Prediction under Dynamic and Time-varying Treatment Regimes. Proc Mach Learn Res. 2024 Aug;252:[https://proceedings.mlr.press/v252/xiong24a.html](https://proceedings.mlr.press/v252/xiong24a.html). PMID: 40433313; PMCID: PMC12113242.
>
> [2]: Deng L, Xiong H, Wu F, Kapoor S, Ghosh S, Shahn Z, Lehman LH. Uncertainty Quantification for Conditional Treatment Effect Estimation under Dynamic Treatment Regimes. Proc Mach Learn Res. 2024 Dec;259:248-266. PMID: 40443560; PMCID: PMC12121963.

---

### Official Review · Reviewer_K5dA · 2025-11-01

**Soundness:** 3
**Presentation:** 2
**Contribution:** 2
**Rating:** 4
**Confidence:** 3

**Summary:**

The paper proposes a novel, regression-based iterative approach to integrate G-computation into neural networks to address time-varying confounding.

**Strengths:**

- The paper proposes a method for treatment effect estimation over time, adjusting time-varying confounders.
- The paper provides a theoretical foundation for the model.
- An implementation code is available for review.

**Weaknesses:**

- The core idea is more likely to be an engineering adaptation of the G-computation. The manuscript does not convincingly differentiate its contribution and demonstrate conceptual advances from that method.
- The motivation behind the study problem (treatment effect estimation over time) needs to be further clarified, particularly in relation to its real-world applicability. It is not evident whether the problem is testable, whether the authors have empirically validated it in real applications, or whether the underlying assumptions commonly hold in practical settings.
- Experiments are conducted on synthetic datasets.  Even the semi-synthetic MIMIC-III uses simulated treatments and outcomes, with no evidence of applicability to real observational data. The study does not evaluate whether estimating CATE over time provides practical benefits or is actually necessary or useful in real-world contexts. The study shows performance under a controlled scenario rather than demonstrating practical utility.
- The baselines are outdated. The paper excludes recent baselines for CATE over time, e.g., G-Transformer [1].

[1] Hess, Konstantin, Dennis Frauen, Valentyn Melnychuk, and Stefan Feuerriegel. "G-Transformer for Conditional Average Potential Outcome Estimation over Time." CoRR (2024).

**Questions:**

- The method adjusts time-varying confounders. How does the model address unobserved confounders?
- Please see weaknesses.

---

> ### Author Response · Authors · 2025-11-25
>
> Thank you very much for your helpful review of our paper! We improved our paper as a result of your comment, and highlighted all changes in **blue color** in our revised **rebuttal PDF**.
>
>
> ### Responses to Weaknesses:
>
> **>W1: G-computation**
>
> Thank you. We respectfully disagree with the characterization that our method is merely an engineering adaptation of the g-computation formula. Classical g-computation (Robins, 1986; Robins et al., 2000) focuses on identification, but not on efficient neural estimation. Hence, their work is an identification recipe for population-level functionals of the data-generating process. It specifies a theoretical recursion for the CAPO, but it does **not** provide an algorithmic procedure suitable for high-dimensional, time-varying observational data, nor does it describe how to implement the recursion with modern sequence models.
>
> Our contribution is to turn this theoretical recursion into a **neural, end-to-end trainable algorithm** that directly targets the conditional, history-dependent potential outcomes. To our knowledge, ours is the **first** tailored neural method that:
>
>
>
> * implements the multi-step g-computation recursion as iterative pseudo-outcome regressions inside a single **unified model**,
> * jointly learns the backbone representation and all regression steps **end-to-end**, rather than estimating each step separately (as required by classical plug-in g-computation),
> * does **not** require learning or sampling from high-dimensional covariate distributions (in contrast to G-Net), and
> * produces stable, low-variance estimates for sequential CAPOs over long horizons. The latter is  a setting where existing methods (RMSNs, CRN, CT, G-Net) either suffer from bias, variance explosion, or both, while our method is highly efficient.
>
> Conceptually, our model departs from existing approaches in two key ways. First, it shifts the focus from balancing heuristics (as in CRN/CT/TE-CDE) or propensity-weight-based correction (as in RMSNs) to a principled, identification-grounded recursion that **guarantees estimation of the correct CAPO** under standard longitudinal assumptions. Second, our architecture ensures that the pseudo-outcome **recursion is encoded explicitly in the computation graph**: the model alternates between generating the next pseudo-outcome under the intervention sequence and learning to predict it from observed histories. This coupling is **not** present in any prior deep learning work on time-varying causal inference.
>
> Thus, while our method is inspired by the g-formula at the level of *identification*, our algorithmic instantiation represents a substantive methodological advance: it is a differentiable, backbone-agnostic, multi-step iterative G-computation network trained jointly across all horizons. The empirical results further demonstrate that this design delivers significantly improved robustness and accuracy.

---

> ### Author Response · Authors · 2025-11-25
>
> **>W2: Motivation behind treatment effect estimation over time**
>
> Thank you. Please let us clarify the motivation behind our problem:
>
> **Real-world applicability:**
>
>
>
> Time-varying treatment effect estimation arises naturally in many **real-world sequential decision-making** settings. Examples include personalized medicine (e.g., adapting drug doses or chemotherapy protocols over time), chronic disease management, public health interventions, adaptive recommendation systems, and other domains where actions and outcomes evolve together. In all such cases, the effect of a **sequence** of treatments given the observed history is the relevant estimand; => *a static, single-shot treatment model cannot capture these feedback loops*.
>
> **Why our formulation is realistic:**
>
> Our problem setup follows the **standard longitudinal causal framework** of Robins (1986, 1999). Compared to static causal inference, which assumes only one treatment with **no** temporal dependencies, our focus is on **time-varying settings**, which are **realistic** in all situations where treatments are administered repeatedly, covariates evolve dynamically, and past treatments influence both future covariates and actions. This structure is ubiquitous in practice and thus naturally reflects real decision-making processes in clinical practice.
>
> **Testability and empirical validation:**
>
> Counterfactual outcomes are unobservable in real-world data, so as in all prior work (e.g., CRN, RMSNs, CT, G-Net), evaluation proceeds via synthetic and semi-synthetic experiments where counterfactuals are known, and one can correctly validate models on ground-truth values. Further, factual prediction on real-world data is often performed as a necessary sanity check. **We follow this exact protocol **(e.g., as in Melnychuk (2022), TE-CDE (2022), Li (2021), Frauen (2025))**:** our synthetic and semi-synthetic results in our **Section 5 validate CAPO estimation**, and our MIMIC-III factual prediction in our **revised Section 5** experiments confirm that the model fits real patient trajectories. This is the standard and accepted methodology for evaluating time-varying counterfactual estimators (e.g., as in Melnychuk (2022), TE-CDE (2022), Li (2021)).
>
> **Assumptions:**
>
> The identification assumptions we use (consistency, sequential ignorability, overlap) are the same as those used across all longitudinal causal inference methods (Bang & Robins, 2005; Robins, 1986; 1994; 1999; Robins et al., 2000; Robins & Hernán, 2009; van der Laan & Gruber, 2012; van der Laan & Rose, 2018). Importantly, sequential ignorability is often **more** plausible in practice than static ignorability because it allows us to handle rich, time-resolved covariate and sequential treatments and does **not** impose unrealistic single-timepoint assumptions.
>
> Further, the assumptions themselves are not really restrictive but are plausbile in many clinical settings:
>
>
>
> * **Consistency:** This is the standard assumption that the observed outcome under the actual treatment history corresponds to the relevant potential outcome, which holds whenever treatments are well-defined and consistently recorded.
>
> * **Sequential Ignorability:** This assumption requires that the recorded covariate history captures the information driving treatment decisions; it is the standard identifying condition used in all longitudinal causal inference methods and is generally plausible in modern datasets with rich time-varying state information (e.g., datasets in oncology or intensive care units, such as ours, typically capture all relevant time-varying information).
>
> * **Positivity:** This requires only that each treatment option occurs with nonzero probability along observed covariate histories. This condition is typically satisfied when the sample size is large enough and the policy is not deterministic.
>
>
> In summary, the problem we study is well-motivated, widely applicable, empirically testable under established protocols, and rests on standard assumptions used throughout the field.

---

> ### Author Response · Authors · 2025-11-25
>
> Upon reading your comment, we realized that we should test numerically how each method responds under violations of these two assumptions is crucial for implementation in real medical scenarios. Hence, **we performed two new experiments** (see **new results in Section 5):**
>
>
>
> * **Experiment I:** We introduce unobserved confounding in our synthetic experiments. **Our IGC-Net has stable performance** and, compared to baselines, does **not** deteriorate more dramatically than any other method (see **new Table 7**).
> * **Experiment II:** We add a parameter that controls the overlap in our experiments. Our results clearly show that our IGC-net is **highly stable** even when overlap is low (see **new Table 6**). This is supported theoretically: our pseudo-outcomes do **not** rely on inverse propensity weighting, which becomes very unstable when the propensities are extreme (see our **Proposition 3**, where we show that variance in our pseudo-outcomes is lower than for those constructed by IPW).
>
> => Evidently, our proposed ICG-Net performs best in both settings and clearly outperforms the baselines.
>
> **Action:** We add **two new experiments** with violations of positivity and sequential ignorability in our **revised Section 5**. Further, we add a discussion on the plausibility of our assumptions in our **new Supplement B**.
>
> **> W3: Experimental validation**
>
> Thank you.
>
> Evaluating time-varying CATE estimators on purely observational real-world data is challenging, because the estimand is a **counterfactual** object: for each subject, we observe only the outcome under the realized treatment sequence and **never** the outcomes under the alternative treatment histories required for CATE computation.
>
> Our experimental setup follows best practice (Lim et al., 2018; Bica et al., 2020; Seedat et al., 2022; Frauen et al., 2025) and is the **standard and only valid** approach for this problem class: (i) controlled synthetic data where the true causal effects are known, and (ii) semi-synthetic data grounded in real covariate trajectories where counterfactual outcomes are simulated solely to expose the true effect function for benchmarking. This is exactly the protocol used by **every** prior neural method for longitudinal CATE/CAPO estimation (e.g., Li et al., 2021; Melnychuk et al., 2022; Seedat et al., 2022; Frauen et al., 2025). Evaluating the performance of any method would require access to counterfactual medical outcomes, which do not and cannot exist in observational datasets. Still, **we demonstrate the applicability of our method for factual outcome prediction using real-world observational data in our revised Section 5**.
>
> **Action:** We moved our experiments on **real-world data** to our **revised Section 5**. Therein, we clarify the difficulties of counterfactual outcome estimation, highlight that we need (semi-)synthetic data for correct validation, and show that our method still has very strong performance for **real-world, factual outcome** prediction.
>
> **> W4: Baselines / G-transformer**
>
> Thank you for pointing us to the G-transformer.
>
> We implemented the *other* G-Transformer (Xiong et al., 2024) as a new state-of-the-art baseline. This method is indeed relevant for time-varying CAPO estimation, and we fully agree that it should be part of the comparison. We have now **added the G-transformer to our experiments**. As shown in our **revised Tables 3+4**, **our method clearly outperforms this baseline**. This performance difference can be explained by the methodological structure of the G-Transformer. Similar to G-Net, it relies on hold-out residual sampling to approximate future covariate paths. While theoretically valid, this strategy introduces substantial inefficiencies: repeatedly sampling high-dimensional residuals incurs *considerable computational overhead and inflates variance*, because the residuals function only as nuisance quantities to approximate conditional means. *Our direct iterative regression framework avoids this bottleneck*, **leading to more stable, data-efficient, and accurate CAPO estimates.**
>
> **Action:** We **added the G-transformer by Xiong et al. (2024) to all of our experiments** (see **new results in our revised Section 5**).

---

> ### Author Response · Authors · 2025-11-25
>
> ### Responses to Questions:
>
> **>Q1: Unobserved confounding**
>
> Thank you. The goal of our work follows the established longitudinal causal inference literature, where methods such as RMSNs, CRN, CT, and G-Net all rely on the standard assumption of **sequential ignorability**, i.e., that relevant confounders are observed or lie in the future and can be adjusted for (time-varying confounders). This setting is the foundation on which existing methods for time-varying causal effect estimation are built (Bang & Robins, 2005; Robins, 1986; 1994; 1999; Robins et al., 2000; Robins & Hernán, 2009; van der Laan & Gruber, 2012; van der Laan & Rose, 2018), and **our contribution advances this line of work** by providing a principled and stable implementation of iterative G-computation.
>
> Research on handling unobserved confounding is a separate methodological direction (e.g., proxy-variable approaches, negative controls, instrumental variables), and none of the baselines or comparison methods in our domain are inherently designed to identify CAPOs when such assumptions fail. Thus, comparing techniques for correcting unobserved confounding would *not* be meaningful within the framework adopted by any prior work.
>
> Nonetheless, we conducted **additional stress tests** in which we explicitly introduced unobserved confounding into the data-generating process. These experiments show that IGC-Net does **not** degrade more severely than existing baselines under violations of sequential ignorability. This demonstrates that  IGC-Net remains robust under such mis-specifications.
>
> **Action:** We add **new experiments** with unobserved confounding to our **revised Section 5**. Our experiments confirm that our method remains at least as robust as all SOTA approaches under mis-specification.
>
> ____
>
> Dennis Frauen, Konstantin Hess, and Stefan Feuerriegel. Model-agnostic meta-learners for estimating heterogeneous treatment effects over time. In ICLR, 2025.
>
> Rui Li, Stephanie Hu, Mingyu Lu, Yuria Utsumi, Prithwish Chakraborty, Daby M. Sow, Piyush Madan, Jun Li, Mohamed Ghalwash, Zach Shahn, and Li-wei Lehman. G-Net: A recurrent network approach to G-computation for counterfactual prediction under a dynamic treatment regime. In ML4H, 2021.
>
> Valentyn Melnychuk, Dennis Frauen, and Stefan Feuerriegel. Causal transformer for estimating counterfactual outcomes. In ICML, 2022.
>
> James M. Robins. A new approach to causal inference in mortality studies with a sustained exposure period: Application to control of the healthy worker survivor effect. Mathematical Modelling, 7:1393–1512, 1986.
>
> James M. Robins. Correcting for non-compliance in randomized trials using structural nested mean models. Communications in Statistics - Theory and Methods, 23(8):2379–2412, 1994.
>
> James M. Robins. Robust estimation in sequentially ignorable missing data and causal inference models. Proceedings of the American Statistical Association on Bayesian Statistical Science, pp. 6–10, 1999.
>
> James M. Robins and Miguel A. Hernan. Estimation of the causal effects of time-varying exposures. Chapman & Hall/CRC handbooks of modern statistical methods. CRC Press, Boca Raton, 2009. ISBN 9781584886587.
>
> James M. Robins, Miguel A. Hernan, and Babette Brumback. Marginal structural models and causal inference in epidemiology. Epidemiology, 11(5):550–560, 2000.
>
> Nabeel Seedat, Fergus Imrie, Alexis Bellot, Zhaozhi Qian, and Mihaela van der Schaar. Continuous-time modeling of counterfactual outcomes using neural controlled differential equations. In ICML, 2022.
>
> Mark J. van der Laan and Susan Gruber. Targeted minimum loss based estimation of causal effects of multiple time point interventions. The International Journal of Biostatistics, 8(1), 2012.
>
> Mark J. van der Laan and Sherri Rose. Targeted learning in data science. Springer, Cham, 2018. ISBN 978-3-319-65303-7.
>
> Hong Xiong, Feng Wu, Leon Deng, Megan Su, and Zach Shan. G-transformer: Counterfactual outcome prediction under dynamic and time-varying treatment regimes. In MLHC, 2024.

---

### Official Review · Reviewer_R3mf · 2025-11-02

**Soundness:** 2
**Presentation:** 3
**Contribution:** 3
**Rating:** 4
**Confidence:** 4

**Summary:**

The paper proposes IGC-Net, a neural method for estimating CAPOs over time from observational data. It aims to properly adjust for time-varying confounding by operationalizing iterative G-computation as a sequence of regression problems trained end-to-end. Experiments on a pharmacokinetic tumor growth simulator and semi-synthetic MIMIC-III show lower RMSE than prior neural baselines

**Strengths:**

- The authors identified the limitations of existing work and proposed a new framework to address them.

- Experiments are thoroughly conducted on both synthetic and real-world datasets.

- Code and data are available for review.

**Weaknesses:**

- Missing important baselines. The paper omits several recent and competitive methods for time-varying counterfactual prediction, such as CGM [1], State-Space Counterfactual Models [2], and G-Transformer [3], that would provide a fairer and more rigorous empirical comparison.

- The experiments implicitly assume correctly specified models and strong sequential ignorability and positivity. There is no investigation into the method’s robustness under violations of overlap or unmeasured confounding, nor any diagnostics for near-zero treatment probabilities, which is an issue particularly relevant for real-world EHR data.

- Both synthetic environments exhibit stylized and stationary dynamics. The claim that IGC-Net is robust “especially for longer horizons” remains weakly supported; validation on scenarios with τ > 6, irregular sampling, missingness, or time-varying action spaces would provide stronger evidence.

[1] Wu, Shenghao, et al. "Counterfactual generative models for time-varying treatments." Proceedings of the 30th ACM SIGKDD Conference on Knowledge Discovery and Data Mining. 2024.

[2] Wang, Haotian, et al. "Effective and Efficient Time-Varying Counterfactual Prediction with State-Space Models." The Thirteenth International Conference on Learning Representations. 2025.

[3] Xiong, Hong, et al. "G-transformer: Counterfactual outcome prediction under dynamic and time-varying treatment regimes." Proceedings of machine learning research 252 (2024): https-proceedings.

**Questions:**

- Robustness & diagnostics: add overlap diagnostics; test positivity violations by skewing treatment assignment; report how IGC-Net degrades vs. RMSNs/TMLE-like DR estimators.

- There’s limited analysis of convergence, computational cost vs. G-Net/RMSNs, or memory footprint for large τ.

---

> ### Author Response · Authors · 2025-11-25
>
> Thank you very much for your helpful review of our paper! We improved our paper as a result of your comment, and highlighted all changes in **blue color** in our revised **rebuttal PDF**.
>
>
> ### Responses to Weaknesses:
>
> **>W1: Baselines**
>
> Thank you for your suggestions!
>
>
>
> 1. CGM is designed to model the *full counterfactual distribution* under time-varying treatments using a generative model of the covariate and outcome process. The estimand of CGM is the *counterfactual density*, **not** the *conditional mean potential outcome (CAPO)* as in our paper. Their task is thus fundamentally different from ours: CGM requires learning high-dimensional conditional distributions and sampling trajectories, whereas our method performs regression-based iterative G-computation to estimate **mean** counterfactual outcomes. Because the objectives, assumptions, and estimands differ, CGM is **not** a suitable baseline for evaluating CAPO estimation.
>
> **Action:** We added CGM to our extended related work and explain why it is different (see **revised Supplement A**).
>
> 2. The authors of the state-space-model propose a framework with decorrelation regularization. However, its learning objective is **different** in two crucial ways:
> * It does **not** perform proper adjustments and therefore does **not** identify the CAPO under sequential ignorability. Instead, it regularizes correlations between hidden states and treatments, which does **not correspond to any identified estimand** in longitudinal causal inference.
> * It directly predicts outcomes from hidden states learned under decorrelation penalties. As a result, its outputs are **not** guaranteed to estimate $E[Y_{t+\tau}(a_{t:t+\tau-1}) \mid H_t]$.
>
> Therefore, although this model studies long-sequence architectures, it is methodologically different from our setting and **not** an appropriate baseline for CAPO estimation.
>
> **Action:** We added the state-space-model to our extended related work and explain why it is different (see **revised Supplement A**).
>
>
>
> 3. We added the G-Transformer as a new baseline. **We added it to our results but see that it is inferior**  (see our **new results in** **Tables 3+4**). This can be explained as follows: the G-Transformer follows the same strategy as G-Net by relying on hold-out residual sampling to approximate future covariate trajectories. Consequently, it inherits the same **inefficiencies** associated with repeatedly sampling high-dimensional residuals, which serve only as a nuisance mechanism to approximate conditional means. This approach has substantial computational overhead, and increased variance (see our **Table 2**), which makes it a less stable and less data-efficient way to obtain estimates of CAPO compared to our direct iterative regression framework.
>
> **Action:** We **added the G-transformer by Xiong et al. (2024) to our experiments** (see **new results in revised Section 5)**. We further discuss all suggested references in our extended related work section in our **revised related work section and Supplement A**, respectively.

---

> ### Author Response · Authors · 2025-11-25
>
> **>W2: Assumptions**
>
> Thank you. Our identifiability assumptions are **standard in the literature** and, without them, estimating CAPO from observational data is **fundamentally impossible**. Below, we break down our response in three steps: (i) why the identifiability assumptions are standard in the literature; (ii) their plausibility; and (iii) new experiments if the assumptions are violated.
>
> **(i) The identifiability assumptions are consistent with prior literature**
>
> These assumptions date back to works in epidemiology and classical statistics (Bang & Robins, 2005; Robins, 1986; 1994; 1999; Robins et al., 2000; Robins & Hernán, 2009; van der Laan & Gruber, 2012; van der Laan & Rose, 2018). The ML literature (Bica et al., 2020; Hess & Feuerriegel, 2025; Frauen et al., 2025; Li et al., 2021; Lim et al., 2018; Melnychuk et al., 2022; Seedat et al., 2022; Wang et al., 2025) has built on these frameworks, and they **all** rely on the same assumptions that we also use: consistency, sequential ignorability, and positivity. These are **not** special assumptions of our framework; they are the **standard identifying conditions across decades of longitudinal causal inference and all modern ML extensions**.
>
> **(ii) Plausibility of the assumptions**
>
> Specifically, sequential ignorability is a relaxation of the standard identifying assumptions in the static setting. Here, one assumes that treatments are only administered once, which is a strong, oftentimes implausible assumption. Instead, treatments are usually administered over time, and hence our framework is more realistic. In short:
>
>
>
> * **Sequential Ignorability:** This assumption requires that the recorded covariate history captures the information driving treatment decisions; it is the standard identifying condition used in all longitudinal causal inference methods and is generally plausible in modern datasets with rich time-varying state information (e.g., datasets in oncology or intensive care units, such as ours, which typically capture all relevant time-varying information).
>
> * **Positivity:** This requires only that each treatment option occurs with nonzero probability along observed covariate histories. This condition is typically satisfied when the sample size is large enough.
>
> **(iii) New experiments**
>
> Upon reading your comment, we realized that we should test numerically how each method responds under violations of these two assumptions is crucial for implementation in real medical scenarios. Hence, **we performed two new experiments** (see **new results in Section 5**):
>
>
>
> * **Experiment I:** We introduce unobserved confounding in our synthetic experiments. **Our IGC-Net has stable performance** and, compared to baselines, does **not** deteriorate more dramatically than any other method (see **new Table 7**).
> * **Experiment II:** We add a parameter that controls the overlap in our experiments. Our results clearly show that our IGC-net is **highly stable** even when overlap is low (see **new Table 6**). This is supported theoretically: our pseudo-outcomes do **not** rely on inverse propensity weighting, which becomes very unstable when the propensities are extreme (see our **Proposition 3**, where we show that variance in our pseudo-outcomes is lower than for those constructed by IPW).
>
> => Evidently, our proposed ICG-Net performs best in both settings and clearly outperforms the baselines.
>
> **Action:** We add **two new experiments** with violations of positivity and sequential ignorability in our **revised Section 5**. Further, we add a discussion on the plausibility of our assumptions in our **new Supplement B**.

---

> ### Author Response · Authors · 2025-11-25
>
> **>W3: Robustness**
>
> Thank you for the suggestions!
>
> **Missingness:** We would like to emphasize that **none** of the existing baselines (e.g., CT (Melnychuk, 2022), G-transformer (Xiong, 2024) natively handle missing data, and all require preprocessing such as interpolation or adding missingness indicators. Our setting is identical: any standard imputation or indicator-based strategy can be applied to both our method and the baselines. This is precisely what we used in both the semi-synthetic and real-world MIMIC-III experiments.
>
> **Time-varying action spaces:** Time-varying treatment sets do *not* pose a conceptual difficulty for our method. Let $\mathcal{A}_t $ be the action space at time $t$, and let the largest action space be $\mathcal{A}_L$. All actions can be represented in a fixed $A_L$-dimensional space, while restricting the support of the action at time $t$ to $\mathcal{A}_t$. This construction is standard in sequential decision models and is compatible with our G-computation formulation as well as with the baselines.
>
> **Longer prediction horizons:** Extending evaluation to larger $\tau$ is a great suggestion!
>
> **Action:** We add larger prediction horizons for our IGC-net and all baselines in our **Supplement F.** Our results demonstrate that our method remains stable and clearly outperforms all baselines.
>
>
> ### Responses to Questions:
>
> **>Q1: Robustness**
>
> Thank you. We **added new experiments demonstrating that our IGC-Net has strong performance under both violations of sequential ignorability and positivity** (we kindly refer to **W2**).
>
> TMLE estimators are designed for average potential outcomes (APOs). We are not aware of any model that uses this framework for *conditional* average potential outcomes (CAPO)  estimation.
>
> **Action:** We add new robustness checks to our **revised Section 5.** Our results confirm the effectiveness of our proposed IGC-Net.

---

> ### Author Response · Authors · 2025-11-25
>
> **>Q2: Convergence and computational cost**
>
> **Convergence:**
>
> Thank you. We are not sure what specific notion of “convergence” you refer to: (i) algorithmic convergence of the training dynamics, (ii) convergence of the iterative G-computation recursion, or (iii) statistical convergence to the true CAPO. We thus discuss them separately:
>
>
>
> * Ad (i): Our paper does not introduce any unusual optimization procedure beyond standard supervised learning, so the training dynamics behave like any other deep regression model.
> * Ad (ii): Convergence of our iterative G-computation is **guaranteed by Proposition 2**.
> * Ad (iii): Regarding *statistical* convergence, our empirical results show that the estimator rapidly approaches the true CAPO even at low sample sizes (e.g., see semi-synthetic experiments for **varying sample sizes**). This is consistent with the theory in **Propositions 1+2**, which show that we target the correct estimand. Further, by **Proposition 3**, we know that our pseudo-outcomes have lower variance than IPW-based pseudo-outcomes as in RMSNs, which leads to lower variance and faster statistical convergence. There is no iterative fixed-point scheme whose stability requires further analysis. Instead, the recursion is executed analytically, and each step is learned via supervised regression.
>
> **Computational cost and memory:**
>
> Thank you. Computation time of our method is comparable to all baselines (see **Supplement I**). More specifically, our method does **not** require any additional computational overhead compared to other methods. Instead, we found that compared to G-net and RMSNs, it even has *computational advantages*, as we detail below:
>
> Let $N$ be the number of units, $\tau$ the time horizon, $d$ the covariate dimension, $H$ the hidden size of the backbone, $L$ the number of backbone layers, and $K$ the number of Monte-Carlo samples used by G-Net. We denote by $C_{\text{backbone}}(\tau)$ the cost of a single forward–backward pass of the sequence backbone over a prediction horizon $\tau$. For example, for a transformer or LSTM (for simplicity, assuming the same constant), this scales as
>
> $$
> C_{\text{backbone}}(\tau) = O(\tau L (dH + H^{2})),
> $$
>
> (e.g., for a transformer, it would include the usual $tau$-dependent attention term). In both cases, the dependence on $\tau$ is contained inside $C_{\text{backbone}}(\tau)$.
>
> Our method performs exactly one such backbone pass per unit, with a lightweight regression head on top. Thus, the total per-epoch cost is
>
> $$
> C_{\text{IGC}} = O(N \, C_{\text{backbone}}(\tau)),
> $$
>
> with no additional simulation or sampling loop.
>
> G-Net shares this backbone cost but **adds a Monte-Carlo simulation stage**: for each unit and each time step, it generates $K$ synthetic covariate updates using hold-out residuals. Each such update is $O(d)$; repeated for all time steps and all samples, this contributes
>
> $$
> C_{\text{MC}} = O(N \tau K d).
> $$
>
> This term is additive because the Monte-Carlo simulation is an extra phase on top of the backbone training. Within this term, $K$ is **multiplicative** with $N$, $\tau$, and $d$: for every unit $N$ and every time step $\tau$, they perform $K$ residual draws, each touching all $d$ covariates. The total G-Net cost is therefore
>
> $$
> C_{\text{G-Net}} = O(N \, C_{\text{backbone}}(\tau) + N \tau K d),
> $$
>
> which is strictly heavier than our regression-only IGC-Net whenever $K > 1$.
>
> RMSNs have a similar $O(N \tau L (dH + H^{2}))$ order per network but must train **both an outcome and a propensity model, effectively doubling the backbone term** (and adding cheap $O(N \tau)$ scalar weight-multiplication operations).
>
> Overall, our method avoids any residual hold-out sampling and relies solely on a single standard backbone pass: **this makes our method both computationally efficient and highly practical for real-world settings**.
>
> **Action:** We added a discussion on the computational complexity in our **Supplement I.1**.

---

> ### Author Response · Authors · 2025-11-25
>
> _____
>
> Ioana Bica, Ahmed M. Alaa, James Jordon, and Mihaela van der Schaar. Estimating counterfactual treatment outcomes over time through adversarially balanced representations. In ICLR, 2020.
>
> Dennis Frauen, Konstantin Hess, and Stefan Feuerriegel. Model-agnostic meta-learners for estimating heterogeneous treatment effects over time. In ICLR, 2025.
>
> Rui Li, Stephanie Hu, Mingyu Lu, Yuria Utsumi, Prithwish Chakraborty, Daby M. Sow, Piyush Madan, Jun Li, Mohamed Ghalwash, Zach Shahn, and Li-wei Lehman. G-Net: A recurrent network approach to G-computation for counterfactual prediction under a dynamic treatment regime. In ML4H, 2021.
>
> Bryan Lim, Ahmed M. Alaa, and Mihaela van der Schaar. Forecasting treatment responses over time using recurrent marginal structural networks. In NeurIPS, 2018.
>
> Valentyn Melnychuk, Dennis Frauen, and Stefan Feuerriegel. Causal transformer for estimating counterfactual outcomes. In ICML, 2022.
>
> James M. Robins. A new approach to causal inference in mortality studies with a sustained exposure period: Application to control of the healthy worker survivor effect. Mathematical Modelling, 7:1393–1512, 1986.
>
> James M. Robins. Correcting for non-compliance in randomized trials using structural nested mean models. Communications in Statistics - Theory and Methods, 23(8):2379–2412, 1994.
>
> James M. Robins. Robust estimation in sequentially ignorable missing data and causal inference models. Proceedings of the American Statistical Association on Bayesian Statistical Science, pp. 6–10, 1999.
>
> James M. Robins and Miguel A. Hernan. Estimation of the causal effects of time-varying exposures. Chapman & Hall/CRC handbooks of modern statistical methods. CRC Press, Boca Raton, 2009. ISBN 9781584886587.
>
> James M. Robins, Miguel A. Hernan, and Babette Brumback. Marginal structural models and causal inference in epidemiology. Epidemiology, 11(5):550–560, 2000.
>
> Nabeel Seedat, Fergus Imrie, Alexis Bellot, Zhaozhi Qian, and Mihaela van der Schaar. Continuous-time modeling of counterfactual outcomes using neural controlled differential equations. In ICML, 2022.
>
> Mark J. van der Laan and Susan Gruber. Targeted minimum loss based estimation of causal effects of multiple time point interventions. The International Journal of Biostatistics, 8(1), 2012.
>
> Mark J. van der Laan and Sherri Rose. Targeted learning in data science. Springer, Cham, 2018. ISBN 978-3-319-65303-7.
>
> Hong Xiong, Feng Wu, Leon Deng, Megan Su, and Zach Shan. G-transformer: Counterfactual outcome prediction under dynamic and time-varying treatment regimes. In MLHC, 2024.

---

### Author Response · Authors · 2025-11-25
**Response to all reviewers**

We thank the reviewers for the comprehensive and helpful feedback on our work.

To improve our work, we added the following improvements to our revised paper (highlighted in $\color{blue}{\text{blue}}$):



1. **New experiments:** We provide several new experiments in our **revised Section 5:** Therein, we analyze the performance of our IGC-Net and the baselines under **(1) unobserved confounding**, and **(2) overlap violations**. Further, we add a new study on **(3) uncertainty quantification** that is directly informed by the literature suggested by the reviewers. Finally, we made robustness checks and **(4) extended the prediction horizon** up to $\tau=12$. Our IGC-Net remains highly stable even for very long horizons.
2. **New baseline:** We implemented a new state-of-the-art baseline, the G-transformer by Xiong et al. (2024), which relies on a similar estimation strategy as G-Net. Our results clearly demonstrate that our approach to G-computation is more robust and allows for more accurate prediction of the CAPO.
3. **Related work:** We added more literature, such as the suggested references by the reviewers, to our related work and the extended related work sections in the supplements.
4. **More discussion:** We added a **new Supplement B,** where we discuss the **plausibility of the standard identifiability assumptions**. We further discuss the computational **complexity** of our method compared to baselines in **Supplement I.1**.
5. **Real-world data:** We moved our experiments on real-world data to the main body of the paper.
6. **Notation:** We clarified some of our notation (e.g., $\bar{a}$ instead of $a$ for the interventional treatment sequence).

We are confident that, with the help of the reviews and our improvements, our work will be an important contribution to the community. Finally, we wanted to say “thank you” – we truly appreciate the help, which allowed us to revise our work and make important improvements.

---

### Meta-Review · Area_Chair_fbr4 · 2026-01-06

**Summary:**

This paper proposes an architecture for estimating conditional average potential outcomes in settings where treatments are administered repeatedly over time. It implements end-to-end, via a neural architecture, an iterative adjustment mechanism to address well-known issues in longitudinal causal inference, such as insufficient control for time-varying confounders and variance explosion in existing methods.

Reviewer R3mf pointed out that some important baselines were missing, that the assumptions were strong, and that validation of stability and robustness was insufficient. In response, the authors explained why the cited methods were not appropriate comparators for the target estimand, implemented additional baseline methods, and added new experiments in which unobserved confounding and overlap violations were artificially introduced. Although the reviewer did not explicitly respond to these new results, the authors’ replies are clear and substantively address the reviewer’s concerns.

Reviewer K5dA argued that the conceptual novelty was insufficient and that experiments based on real applications were lacking. The authors responded by emphasizing that the novelty lies in explicitly encoding the iterative g-computation recursion within a neural network and jointly optimizing all steps end-to-end. They also implemented additional competing methods to demonstrate the superiority of the proposed approach and highlighted real-data analyses more prominently in the main text. These responses largely address the reviewer’s experimental concerns, especially regarding real-world data. However, while the clarification partially alleviates concerns about technical novelty, it remains somewhat unclear whether this fully resolves the issue, given that the design is fundamentally an iterative realization of an existing concept.

Reviewer Kwm7 raised numerous implementation- and usage-level questions, including unclear notation and the handling of treatment sequences during training versus inference. The authors comprehensively revised the notation and explained that training time is relatively short and that reusing the backbone makes the method practically feasible. Based on these detailed responses, the reviewer concluded that their concerns had been resolved, raised their score, and ultimately supported acceptance.

Reviewer Wupz expressed concerns about practical identifiability in neural network approximations (e.g., the possibility that intermediate pseudo-outcomes differ across random initializations), the restriction to binary treatments, and the lack of discussion on interpretability. The authors supplemented their response with theoretical explanations of identifiability, clarified that estimates converge to the same values across multiple initializations, and discussed extensions to multi-valued treatments as well as the trade-off between interpretability and predictive accuracy. Following these clarifications, the reviewer ultimately gave a strongly positive evaluation. The issues raised were largely minor, and the reviewer had already held a favorable view of the paper from the outset.

Reviewer drGZ acknowledged the core idea but criticized the insufficient coverage of recent literature and questioned how well uncertainty could be captured compared to full MCMC-based approaches. The authors addressed these points by implementing G-Transformer as an additional baseline and empirically demonstrating the advantages of their approach. They also clarified that the primary goal is point estimation, while showing through additional experiments that standard techniques such as dropout and ensembles can be easily incorporated for uncertainty quantification. While these responses carefully addressed the technical questions, they did not lead to a substantial improvement in the reviewer’s final evaluation.

Overall, reviewers differed in their assessments of the paper’s novelty, the strength of its assumptions, and the appropriateness of the selected baselines. Through precise and well-structured rebuttals, the authors addressed the major concerns by adding numerous experiments, implementing additional baselines, revising notation, and providing theoretical clarifications. The remaining concerns are mostly minor or subjective. As a result of the rebuttal and revision, the paper can reasonably be regarded as a solid and technically well-developed piece of work.

**Reviewer Concerns:**

See above.

**Reviewer Scores:**

See above.

---

### Decision · Program_Chairs · 2026-01-26

Accept (Poster)